# SPATIAL PATTERN OF ACCUMULATION AT TAYLOR DOME DURING MARINE ISOTOPE STAGE 4: STRATIGRAPHIC CONSTRAINTS FROM TAYLOR GLACIER

James A. Menking[1], Edward J. Brook[1], Sarah A. Shackleton[2], Jeffrey P. Severinghaus[2], Michael N. Dyonisius[3], Vasilii Petrenko[3], Joseph R. McConnell[4], Rachael H. Rhodes[5, 6], Thomas K. Bauska[5, 6], Daniel Baggenstos[7], Shaun Marcott[8], Stephen Barker[9]

[1]College of Earth, Ocean, and Atmospheric Sciences, Oregon State University, Corvallis, 97333, United States of America
[2]Scripps Institution of Oceanography, University of California San Diego, La Jolla, 92037, United States of America
[3]Department of Earth and Environmental Sciences, University of Rochester, Rochester, 14627, United States of America
[4]Division of Hydrological Sciences, Desert Research Institute, Reno, 89512, United States of America
[5]Department of Earth Sciences, University of Cambridge, Cambridge CB2 3EQ, United Kingdom
[6]Department of Geography and Environmental Sciences, Northumbria University, Newcastle upon Tyne NE1 8ST, United Kingdom
[7]Climate and Environmental Physics, University of Bern, Bern, 3012, Switzerland
[8]Department of Geoscience, University of Wisconsin-Madison, Madison, 53706, United States of America
[9]School of Earth and Ocean Sciences, Cardiff University, Cardiff, CF10 3AT, United Kingdom

*Correspondence to:* James A. Menking (menkingj@oregonstate.edu)

**Abstract.** New ice cores retrieved from the Taylor Glacier (Antarctica) blue ice area contain ice and air spanning the Marine Isotope Stage (MIS) 5/4 transition, a period of global cooling and ice sheet expansion. We determine chronologies for the ice and air bubbles in the new ice cores by visually matching variations in gas and ice phase tracers to preexisting ice core records. The chronologies reveal an ice age-gas age difference (Δage) approaching 10 ka during MIS 4, implying very low snow accumulation in the Taylor Glacier accumulation zone. A revised chronology for the analogous section of the Taylor Dome ice core (84 to 55 ka), located to the south of the Taylor Glacier accumulation zone, shows that Δage did not exceed 3 ka. The difference in Δage between the two records during MIS 4 is similar in magnitude but opposite in direction to what is observed at the last glacial maximum. This relationship implies that a spatial gradient in snow accumulation existed across the Taylor Dome region during MIS 4 that was oriented in the opposite direction of the accumulation gradient during the last glacial maximum.

## 1 Introduction

Trapped air in ice cores provides a direct record of the Earth's past atmospheric composition (e.g., Bauska et al., 2016; Petrenko et al., 2017; Schilt et al., 2014). Measurements of trace gas species, and particularly their isotopic composition create a demand for large-volume glacial ice core samples. Blue ice areas, where

a combination of glacier flow and high ablation rates bring old ice layers to the surface, offer relatively easy access to large samples and can supplement traditional ice cores (Bintanja, 1999; Sinisalo and Moore, 2010). Blue ice areas often have complex depth-age and distance-age relationships disrupted by folding and thinning of stratigraphic layers (e.g., Petrenko et al., 2006; Baggenstos et al., 2017). Taking full advantage of blue ice areas requires precise age control and critical examination of the glaciological context in which they form.

Effective techniques for dating ablation zone ice include matching of globally well-mixed atmospheric trace gas records (e.g., $CH_4$, $CO_2$, $\delta^{18}O_{atm}$, $N_2O$) and correlation of glaciochemical records (e.g., $\delta^{18}O_{ice}$, $Ca^{2+}$, insoluble particles) to existing ice core records with precise chronologies (Bauska et al., 2016; Schilt et al., 2014; Petrenko et al., 2008; Schaefer et al., 2009; Baggenstos et al., 2017; Petrenko et al., 2016; Aarons et al., 2017). Other useful techniques include $^{40}Ar_{atm}$ dating (Bender et al., 2008; Higgins et al., 2015), and radiometric $^{81}Kr$ dating (Buizert et al., 2014). Matching of gas and glaciochemical records can provide high precision with relatively small samples, and some measurements can be made in field settings. In contrast, $^{40}Ar_{atm}$ and $^{81}Kr$ require complex laboratory work and do not provide the level of age precision available from correlation methods, although these techniques do provide independent age information that can extend beyond the age range of existing records.

A number of blue ice areas have provided useful paleoclimate archives including Pakitsoq, Greenland for the Younger Dryas-Preboreal transition (Petrenko et al., 2006; Petrenko et al., 2009; Schaefer et al., 2009; Schaefer et al., 2006), Allan Hills, Victoria Land, Antarctica for ice 90-250 ka and > 1 Ma (Spaulding et al., 2013; Higgins et al., 2015), Mt. Moulton, Antarctica for the last interglacial (Korotkikh et al., 2011), the Patriot Hills, Horseshoe Valley, Antarctica, for ice from the last glacial termination (Fogwill et al., 2017), and Taylor Glacier, McMurdo Dry Valleys, Antarctica, for ice spanning the last glacial termination and MIS 3 (Bauska et al., 2016; Schilt et al., 2014; Baggenstos et al., 2017; Petrenko et al., 2017). Taylor Glacier is particularly well suited for paleoclimate reconstructions because of excellent preservation of near surface ice, large age span, and continuity of the record (Buizert et al., 2014; Baggenstos, 2015; Baggenstos et al., 2017). The proximity of the Taylor Dome ice core site to the probable deposition site for Taylor Glacier ice provides a useful point of comparison for the downstream blue ice area records (Figure 1).

This study extends the Taylor Glacier blue ice area archive by developing ice and gas chronologies spanning the MIS 5/4 transition (74-65 ka), a period of global cooling and ice sheet expansion. In 2014-2016 several ice cores were retrieved approximately 1 km down-glacier from the "Main Transect," the across-flow transect containing ice from Termination 1 through MIS 3 (Baggenstos et al., 2017) (Figure 1). This paper describes (1) dating the new ice cores via matching of variations in $CH_4$, $\delta^{18}O_{atm}$, dust, and $\delta^{18}O_{ice}$ to preexisting records, and (2) the description of a new climate record from Taylor Glacier across MIS 4, which was previously thought to be absent from the glacier (Baggenstos et al., 2017). New

measurements of $CH_4$ and $CO_2$ from the Taylor Dome ice core are used to revise the Taylor Dome chronology across the MIS 5/4 transition and MIS 4 to allow better comparison of the glaciological conditions at Taylor Dome with those at the accumulation region for Taylor Glacier. This comparison allows inferences about the climate history of the Taylor Dome region implied from the differences in the delta age ($\Delta$age = ice age – gas age) between the two sites.

## 2 Field site and methods

### 2.1 Field site

Taylor Glacier is an outlet glacier of the East Antarctic Ice Sheet that flows from Taylor Dome, and terminates in the McMurdo Dry Valleys (Figure 1). The Taylor Glacier deposition zone is on the northern flank of Taylor Dome, a peripheral ice dome of the East Antarctic Ice Sheet centered at 77.75 S, 159.00 E on the eastern margin of the Ross Sea (Figure 1). The Taylor Glacier deposition zone receives 3-5 cm ice equivalent accumulation annually in present-day climate conditions (Kavanaugh et al., 2009a; Morse et al., 1999). The glacier flows through Taylor Valley at a rate of ~ 10 m a$^{-1}$ and terminates near Lake Bonney, approximately 30 km from the Ross Sea (Kavanaugh et al., 2009b; Aciego et al., 2007). The ablation zone extends approximately 80 km from the terminus (Kavanaugh et al., 2009b). The close proximity to McMurdo Station provides excellent logistical access to the site (e.g., Fountain et al., 2014; Petrenko et al., 2017; Baggenstos et al., 2017; Marchant et al., 1994; Aarons et al., 2017).

A combination of relatively high sublimation rates (~ 10 cm a$^{-1}$) and relatively slow flow creates an ablation zone where ancient ice with a large range of ages is exposed at the surface of Taylor Glacier (Kavanaugh et al., 2009a; Kavanaugh et al., 2009b). An along-flow transect of water stable isotopes from just below the equilibrium line to the terminus revealed ice from the last glacial period outcropping at sporadic places along the transect (Aciego et al., 2007). The sporadic nature of the outcrops was later shown to be an artifact of sampling nearly parallel to isochrones such that they were occasionally crossed (Baggenstos et al., 2017). More recent across-flow profiles dated with stratigraphic matching of well-mixed atmospheric gases revealed ice that varies continuously in age from the Holocene to ~ 50 ka (Schilt et al., 2014; Bauska et al., 2016; Baggenstos et al., 2017), with ice of last interglacial or older age found near the terminus of the glacier (Baggenstos et al., 2017; Buizert et al., 2014). The most heavily sampled archive is a 500 m section called the 'Main Transect,' oriented perpendicularly to isochrones (Figure 1) across a syncline-anticline pair containing ice spanning ~ 50 ka before present (BP) to the mid Holocene (7 ka) (Baggenstos et al., 2017). Ice stratigraphy in the Main Transect dips approximately vertically so that it is possible to obtain large quantities of ice of the same age by drilling vertical or near-vertical ice cores (e.g., Baggenstos et al., 2017; Petrenko et al., 2017; Petrenko et al., 2016; Schilt et al., 2014; Bauska et al., 2016; Bauska et al., 2018). Ice containing the full MIS 5/4 transition was formerly considered to be missing from the glacier (Baggenstos, 2015; Baggenstos et al., 2017), but we show here that a new ice core near the Main Transect contains an intact record with ice dating from 76.5-60.6 ka and air dating from 74.0-57.7 ka.

## 2.2 Core retrieval

In the 2013-2014 season an exploratory core was drilled vertically using a "PICO" hand auger 380 m away ("-380 m" by convention) from a benchmark position (77.75891˚ S, 161.7178˚ E in Jan. 2014) along the Main Transect (Figure 1). In the 2014-2015 field season another exploratory core was drilled vertically using the "PICO" hand auger approximately 1 km down glacier from the Main Transect (77.7591˚ S, 161.7380˚ E in Dec. 2014) where older ice near the surface was suspected. This site is hereafter referred to as the MIS 5/4 site (Figure 1). An ice core was drilled directly adjacent to the PICO borehole at the MIS 5/4 site using the Blue Ice Drill (BID), a 24 cm diameter shallow coring device designed for retrieving large volume ice samples suitable for trace gas and isotope analysis (Kuhl et al., 2014). The section 9-17 m was sampled in the field for laboratory trace gas analyses at Oregon State University (OSU) and at the Scripps Institution of Oceanography (SIO).

In the 2015-2016 season a second large-volume core was drilled directly adjacent to the previous MIS 5/4 boreholes using the BID, and the sections 0-9 m and 17-19.8 m were sampled for trace gas analyses at OSU and at SIO. The entire 0-19.8 m of this core was sampled for continuous flow analysis (CFA) in the field and at the Desert Research Institute (DRI). Samples for all analyses were cut with a band saw on the glacier, stored in chest freezers at < -20˚ C in camp, and flown to McMurdo Station within 2 weeks of retrieval, where they were stored at < -20˚ C. Storage temperature was < -20˚ C for the remainder of their shipment to the USA and subsequent storage in laboratories.

## 2.3 Analytical methods

A field laboratory at the Taylor Glacier field camp permitted continuous measurements of $CH_4$ and particle count on ice core samples within days of drilling and recovery (Table 1). $CH_4$ concentration was measured using a Picarro laser spectrometer coupled to a continuous gas extraction line with a de-bubbler similar to that described in Rhodes et al. (2013). The continuous $CH_4$ data were calibrated by measuring standard air of known $CH_4$ concentration introduced into a stream of gas-free water to simulate a bubble/ liquid mixture similar to the melt stream from ice core samples. The tests indicated 3.5-5.5% loss of $CH_4$ due to dissolution in the melt stream. We adjusted the continuous $CH_4$ data upwards by 5% to account for the solubility effect, which resulted in a good agreement between our measurements and other Antarctic $CH_4$ records (e.g., Schilt et al., 2010). Insoluble particle abundance was also measured continuously in the field using an Abakus particle counter coupled to the continuous melt-water stream. In order to obtain exploratory gas age information and verify the continuous $CH_4$ data, discrete ice core samples were also measured for $CH_4$ concentration in the field using a Shimadzu gas chromatograph coupled to a custom melt-refreeze extraction line, a manually operated version similar to the automated system used at OSU (Mitchell et al., 2011; Mitchell et al., 2013).

Laboratory analyses on recovered samples and archived Taylor Dome samples included discrete $CH_4$ and $CO_2$ concentrations, $\delta^{15}N$ of atmospheric $N_2$, and $\delta^{18}O$ of atmospheric oxygen ($\delta^{18}O_{atm}$), continuous $CH_4$ concentration, $\delta^{18}O_{ice}$, major ion and elemental chemistry, and insoluble particle counts (Table 1). Continuous chemistry, dust, $\delta^{18}O_{ice}$, and $CH_4$ measurements were made at DRI by melting 3.5 cm x 3.5 cm ~ 1 m longitudinal samples of ice and routing the melt stream to in-line instruments (McConnell, 2002; Maselli et al., 2013). Insoluble particles were measured using an Abakus particle counter, water isotopes using a Picarro laser spectrometer (Maselli et al., 2013), and $CH_4$ using a Picarro laser spectrometer and air extraction system similar to that used in the field (Rhodes et al., 2013). Continuous $CH_4$ data measured at DRI were calibrated with air standards as described above. The upward adjustment to account for dissolution in the melt stream was 8% in this case. Discrete $CH_4$ and $CO_2$ measurements were made at OSU. $CH_4$ was measured using an Agilent gas chromatograph equipped with a flame ionization detector coupled to a custom melt-refreeze extraction system (Mitchell et al., 2011). $CO_2$ was measured (1) on an Agilent gas chromatograph equipped with a Ni catalyst and a flame ionization detector coupled to a custom dry extraction "cheese grater" system for carbon isotopic analyses (Bauska et al., 2014), and (2) on a similar Agilent gas chromatograph coupled to a dry extraction needle crusher system (Ahn et al., 2009). $\delta^{15}N-N_2$ and $\delta^{18}O_{atm}$ were measured at SIO using a Thermo Delta V mass spectrometer coupled to a custom gas extraction system (Severinghaus et al., 1998; Petrenko et al., 2006).

Discrete measurements of $CH_4$ and $CO_2$ were made at OSU on archived Taylor Dome ice core samples following the same procedures described above (Table 1).

### 2.4 Data uncertainties

The analytical uncertainties associated with new data presented in this manuscript are reported in Table 1. In addition to the uncertainties in concentration and isotopic measurements, we address uncertainties related to: (1) smoothing of gas records due to dispersion and mixing in the CFA system (Rhodes et al., 2013; Stowasser et al., 2012), (2) depth uncertainty in gas and ice samples, and (3) artifacts due to contamination of gas and dust in near-surface ice. The effect of analytical smoothing is negligible, as demonstrated by close agreement of continuous $CH_4$ with high-resolution discrete $CH_4$ data from 9-17 m in the 2014-2015 MIS 5/4 core (Figure S1). Depth uncertainties of up to 20 cm resulted from unaligned, angled core breaks of up to 10 cm in length as well as small depth logging errors. Contamination is only a concern in near-surface ice where thermal expansion and contraction causes abundant cracks on the surface of Taylor Glacier. The cracks rarely penetrate below 4 m and have never been observed deeper than 7 m (Baggenstos et al., 2017). Gas measurements may be sensitive to contamination from resealed cracks between 0-4 m depth, and dust measurements may be affected by local dust deposition between 0-40 cm depth (Baggenstos et al., 2017; Baggenstos et al., 2018). To minimize this problem we avoided analyses of ice with visible fractures.

**3 Age models for Taylor Glacier and Taylor Dome**

**3.1 Taylor Glacier MIS 5/4 cores**

For the new MIS 5/4 cores the sections retrieved during the 2014-2015 season (9-17 m) and 2015-2016 season (0-9 m and 17-20 m) are hereafter treated as one ice core record (unified depth and age scales), which is justified given the close proximity of the boreholes (< 2 m spacing at surface) and the minimal depth uncertainty between the cores (≤ ~ 20 cm). The depth uncertainty is the maximum offset due to angle breaks at the ends of cores, which never exceeded 10 cm. Observable depth offsets between replicate measurements also do not exceed 20 cm (discussed in more detail below and in Supplementary Information). No depth adjustments were made to the raw data from any of the ice cores.

A gas age model for the Taylor Glacier MIS 5/4 cores was constructed by matching variations in $CH_4$ and $\delta^{18}O_{atm}$ to preexisting ice core records synchronized to the Antarctic Ice Core Chronology (AICC) 2012 (Veres et al., 2013; Bazin et al., 2013) (Figure S1). This approach is valid for the gas age scale because $CH_4$ and $^{18}O_{atm}$ are globally well-mixed (Blunier et al., 2007; Blunier and Brook, 2001). Variations in $CH_4$ were tied to the EPICA Dronning Maud Land (EDML) record (Schilt et al., 2010), and $\delta^{18}O_{atm}$ was tied to the North Greenland Ice Coring Project (NGRIP) record (Landais et al., 2007). These datasets were chosen because they contain the highest-resolution $CH_4$ and $\delta^{18}O_{atm}$ data available on the AICC 2012 timescale for this time period. Tie points linking ages to depths were manually chosen (Figure S1 and Table 2). Ages between the tie points were interpolated linearly.

$CO_2$ data were not used to tie Taylor Glacier to AICC 2012. An offset between the Taylor Glacier data and the Antarctic composite record of Bereiter et al. (2015) during the MIS 4/3 $CO_2$ increase between 64 and 60 ka (Taylor Glacier lower by ~ 13 ppm at 61.5 ka, Figure 2) could bias our age model toward older ages. This offset may be real (e.g., Luthi et al., 2008), and we note that $CO_2$ offsets of even larger magnitude exist between Taylor Glacier and the composite record in the interval 68-64 ka (Figure 2).

Nonetheless, the general agreement with trends in preexisting $CO_2$ measurements supports the chosen tie points for the new gas age scale (Figure 2). The resemblance of the Taylor Glacier $\delta^{18}O_{atm}$ record to NGRIP $\delta^{18}O_{atm}$ between 72-63 ka also supports the gas age scale, because tie points younger than 72 ka were picked only from $CH_4$ data. This is particularly important because $CH_4$ variability is small between 70-60 ka, limiting potential tie point selections. Good agreement between $CH_4$ variability in the new MIS 5/4 cores and the independently dated $\delta^{18}O\text{-}CaCO_3$ from Hulu Cave speleothems (Wang et al., 2001) also suggests that the gas age scale is accurate (Figure S5). Agreement between atmospheric $CH_4$ concentration (a global signal) and Hulu Cave speleothem $\delta^{18}O\text{-}CaCO_3$ is expected because both parameters are sensitive to shifts in the latitudinal position of the Intertropical Convergence Zone and the delivery of moisture via the tropical rain belts (Rhodes et al., 2015; Buizert et al., 2015).

An ice chronology was constructed for the new Taylor Glacier MIS 5/4 cores by matching variations in $Ca^{2+}$, insoluble particle count, and $\delta^{18}O_{ice}$ to preexisting EPICA Dome C (EDC) dust (Lambert et al., 2008; Lambert et al., 2012) and $\delta^{18}O_{ice}$ records (Jouzel et al., 2007) synchronized to AICC 2012 (Figure S2). This approach has been used successfully at Taylor Glacier before (e.g., Baggenstos et al., 2018), and it is possible because to first order the temporal patterns of dust content and $\delta^{18}O_{ice}$ in Antarctic ice are highly correlated at different ice core locations across the continent (Mulvaney et al., 2000; Schupbach et al., 2013). Tie points were chosen manually (Figure S2 and Table 3), and ages were interpolated linearly between them. The synchronized records are displayed in Figure 2. A more detailed discussion and justification of tie point choices for the Taylor Glacier MIS 5/4 chronologies is provided in the supplementary information.

### 3.2 Taylor Glacier -380 m Main Transect core

To investigate continuity between the Taylor Glacier Main Transect and the new MIS 5/4 site, we constructed a gas age scale for the ice core at -380 m on the Main Transect collected during the 2013-2014 season (Figure 3). Gas ages were determined by matching $CH_4$ data to EDML on AICC 2012 (Table 4). The chronology of the -380 m core is more uncertain than for the MIS 5/4 cores because there are fewer features to match in the gas records, but the synchronous variability in $CH_4$, $CO_2$, and $\delta^{18}O_{atm}$ is unique to the late MIS 4 and MIS 4/3 transition. The observation of late MIS 4 air (but not the full MIS 5/4 transition) was the basis for moving our 2014-2015 ice reconnaissance efforts down-glacier from the Main Transect where older ice is closer to the surface.

### 3.3 Taylor Dome core

The early Taylor Dome chronologies (e.g., Steig et al., 1998; Steig et al., 2000) were recently revised by Baggenstos et al. (2018) from 0-60 ka in light of evidence that the original timescales were incorrect (e.g., Mulvaney et al., 2000; Morse et al., 2007). To investigate the new Taylor Glacier MIS 5/4 climate archive in the context of the glaciological history of the Taylor Dome region, we revised the Taylor Dome gas and ice age scales for the period 84-55 ka (504-455 m). We adopted the recently published age ties (Baggenstos et al., 2018) for the interval that overlaps with our new records (60-55 ka). We then extended the timescale to 84 ka using new and preexisting data. Gas tie points were chosen by manually value-matching variations in Taylor Dome $CH_4$ data to EDML $CH_4$ on AICC 2012. One of the new tie points matches variability observed in a preexisting $CH_4$ record from Taylor Dome (Brook et al., 2000) to the EDML $CH_4$ record (Supplementary Information), and three tie points adopted from Baggenstos et al. (2018) match variations observed in preexisting Taylor Dome $CO_2$ data (Indermuhle et al., 2000) to WD2014 (Buizert et al., 2015) (Figure S3 and Table 5). Ice tie points were chosen by manually matching variations in the Taylor Dome $Ca^{2+}$ record (i.e., Mayewski et al., 1996) to EDC dust (Lambert et al., 2012; Lambert et al., 2008) on AICC 2012 (Figure S4 and Table 6).

The general agreement between the Taylor Dome $CO_2$ record and preexisting data from other ice cores supports our revised gas age scale (Figure 4), but we did not use the $CO_2$ data in constructing the age scale apart from the points mentioned above. The general resemblance between Taylor Dome $\delta^{18}O_{atm}$ and NGRIP $\delta^{18}O_{atm}$ also supports the gas age scale, although the Taylor Dome $\delta^{18}O_{atm}$ are somewhat scattered due to lower measurement precision (Sucher, 1997). Taylor Dome $CH_4$ data on the new timescale also agree well with $\delta^{18}O$-$CaCO_3$ variability in Hulu Cave speleothems (Figure S5). The supplementary information provides further justification of tie point choices for our revised Taylor Dome chronology.

### 3.4 Age model uncertainties

There are two types of uncertainty associated with the new gas and ice age models: (1) absolute age uncertainty propagated from the reference age scale (AICC 2012), and (2) relative age uncertainty arising from depth offsets and the manual selection of tie points. The latter is a function of (a) choosing the correct features to tie, (b) the resolution of the data that define the tie point features, and (c) the measurement error. To estimate relative age uncertainty we assigned a maximum and minimum age to each chosen tie point (Figure 2, Figure 4, Tables 2-3, and Tables 5-6). The age ranges were determined by closely examining the matched features and estimating the maximum and minimum possible ages based on our judgment of factors (a)-(c) above. The resulting error ranges for our tie points are conservative. Maximum and minimum age scales were determined for the MIS 5/4 cores and the Taylor Dome ice core by interpolating linearly between the maximum and minimum age assigned to each tie point (Figures 5a and 5c).

Depth errors contribute additional uncertainty to the total relative uncertainty described above. Depth errors between the Taylor Glacier MIS 5/4 cores were estimated by observing the depth offsets in features resolved by the continuous versus discrete $CH_4$ measurements (Figure S1). The largest depth offset was at the $CH_4$ rise at ~ 16.0 m: there is a 10 cm offset between the continuous field $CH_4$ and the discrete laboratory $CH_4$, and a 20 cm offset between the continuous and discrete laboratory $CH_4$. Approximately 20 cm offsets are also apparent in the ice phase by comparing insoluble particle count data measured in the field versus in the laboratory (Figure S2). 20 cm equates to 420 years on the new gas age scale where gas age changes most rapidly with depth (65-60 ka, Figure 5a), and 360 years on the ice age scale where ice age changes most rapidly with depth (70-61 ka, Figure 5a). We adopted 420 years and 360 years as conservative estimates of the relative gas age error and ice age error, respectively, due to depth uncertainty. These errors were propagated into the calculations of maximum and minimum Taylor Glacier age scales. We are unaware of depth uncertainties in the archived Taylor Dome samples used in this study so no additional depth uncertainty was added to the age error estimates for Taylor Dome.

The mean of the estimated age errors along the cores provides a reasonable cumulative estimate of the relative uncertainty in the new Taylor Glacier MIS 5/4 and revised Taylor Dome chronologies. For Taylor Glacier the mean relative uncertainty is ± 0.9 ka for the gas age and + 1.3 ka/ - 1.2 ka for the ice age. For

Taylor Dome the mean relative uncertainty is + 0.7 ka/ - 0.5 ka for the gas age and ± 0.6 ka for the ice age. The relative uncertainty is larger in Taylor Glacier due to the depth errors described above.

We did not explicitly account for errors associated with interpolation. Given our conservative estimates of tie point error, we believe any additional uncertainty is minor relative to our conclusions. Tie points were not assigned to the end points of our records unless there was clearly a feature to match (with the exception of the last Taylor Glacier ice age tie point described in the supplementary information). Age models are extrapolated from the closest pair of tie points for the intervals 0-0.31 m for the ice age scale, and 0-1.74 m and 19.27-19.8 m for the gas age scale.

We suspect there are differences between Taylor Glacier and EDML due to gas transport in the firn layer, because the features resolved in the new Taylor Glacier $CH_4$ data are generally smoothed relative to the same features in EDML (Figures 2 and S1). However, we believe that the effect of firn smoothing on our tie point selections is within the estimated relative error for the chronology (Figure 5a). In contrast, $CH_4$

features in the Taylor Dome record appear less smoothed (Figures 4 and S3).

The absolute age uncertainty in the reference timescale (AICC 2012) is 2.5 ka for ice age and 1.5 ka for gas age (Veres et al., 2013). By nature, these errors are inherited by the Taylor Glacier 5/4 chronology and the revised Taylor Dome chronology, though the total error in our chronologies should be less than the total

propagated EDC and EDML 1 σ uncertainties because the uncertainties in gas age and ice age are correlated with depth. The close match of our gas age scales to the radiometrically dated Hulu Cave record (Wang et al., 2001) indicates that the absolute age uncertainties in our gas age scales are equal to or lower than the implied AICC 2012 error estimates (Figure S5). We estimate an upper absolute age uncertainty of 1.5 ka for our Taylor Glacier and Taylor Dome gas age scales based on the phasing of features in the $\delta^{18}O$-

$CaCO_3$ record from Hulu Cave and our $CH_4$ records.

### 4 Results

### 4.1 Data quality and initial observations

Preliminary observations of $CH_4$ variability in the MIS 5/4 PICO core revealed that the air likely contained

the full MIS 5/4 transition and the MIS 4/3 transition (Figure S1). The new Taylor Glacier MIS 5/4 ice cores provide a record of the atmospheric history spanning 74-57.7 ka including the ~ 40 ppm $CO_2$ concentration decrease at the MIS 5/4 transition and the ~ 30 ppm $CO_2$ concentration increase near the MIS 4/3 transition (Figure 2). The new ice cores also record millennial-scale variability in $CH_4$, $CO_2$, $\delta^{18}O_{atm}$, as well as $\delta^{18}O_{ice}$ and dust. Taylor Glacier $\delta^{18}O_{ice}$ is more variable than other Antarctic records, most likely

recording local-scale changes in post-depositional alteration (Baggenstos, 2015; Baggenstos et al., 2018; Neumann et al., 2005). We note that large features seen in other Antarctic stable isotope records are preserved (e.g. 2-3 ‰ changes at Antarctica Isotope Maximum (AIM) 19 and AIM 20).

Field measurements (continuous $CH_4$ and insoluble particles) were replicated in the laboratory at DRI (Figures S1 and S2). Replication allowed assessment of data quality and supports the original data acquired in the 2014-2015 and 2015-2016 field seasons. Offsets between laboratory and field measurements are

minor in the section 4-20 m and are due to the depth offsets described above (Figures S1 and S2). $CH_4$ offsets between field and DRI data in the section 0-4 m are much larger (Figure S1) and may be attributed to contamination of the gas signal due to resealed thermal cracks near the glacier surface (Baggenstos et al., 2017). We report these shallow $CH_4$ data for completeness. We assign two gas age tie points at 1.74 m (58.21 ka) and 3.15 m (59.10 ka) to offer a plausible gas age scale for the shallow ice, but the gas age scale

for 0-4 m is not interpreted further and does not influence the conclusions of this study. $CH_4$ data from the section 0-1 m were excluded due to very high amounts of contamination in both laboratory and field samples ($CH_4$ > 1000 ppb). Continuous laboratory $CH_4$ data were also excluded between 14.57-15.0 m and 17.55-17.95 m due to technical problems with instrumentation. Variations in $Ca^{2+}$ and insoluble particle counts generally agree with each other, suggesting both parameters are recorders of dust variability.

Particle count data measured at DRI were averaged every 1 cm, explaining why the record appears less noisy than insoluble particles counts measured in the field (Figure S2).

$CH_4$ variations in Taylor Glacier are smoother than in EDML. The largest difference appears at DO 18 (64.9 ka) where Taylor Glacier $CH_4$ is ~ 40 ppb lower than EDML (and Taylor Glacier $\delta^{18}O_{atm}$ is ~ 0.1 ‰

more enriched than NGRIP) (Figure 2). The $CH_4$ rise associated with DO 19 is less attenuated, ~ 20 ppb lower in Taylor Glacier relative to EDML (72.3 ka, Figure 2). Some of these differences may be due to higher analytical noise in the EDML record (mean of EDML $CH_4$ 1 $\sigma$ = 10.25 ppb between 74-60 ka). New Taylor Dome $CH_4$ data from OSU show little or no attenuation relative to the EDML record. Taylor Dome $CH_4$ at the onset of DO 19 (72.3 ka) is 14 ppb higher than in EDML and 10 ppb lower at the onset of DO

20 (75.9 ka) (Figure 4). These offsets are within the combined 1 $\sigma$ error of the measurements. The smoothing in the three ice cores reflects the firn conditions in which bubble trapping occurred, with smoother variations resulting from a thicker lock-in zone that traps bubbles with a larger age distribution. The new $CH_4$ data suggest Taylor Dome and EDML records are similarly smoothed by the firn while Taylor Glacier bubbles have a larger gas age distribution.

One clear observation from the new ice core is that the ice from MIS 4 is very thin at Taylor Glacier; indeed the entire MIS 4 period (70-60 ka) appears to be contained in ~ 6 m of ice (Figure 5a). This partially explains why the MIS 4 interval has been relatively difficult to locate. Thin ice could occur due to either low snow accumulation or mechanical thinning of ice layers due to glacier flow. The implications of thin

layers for the accumulation history are discussed in more detail below. Taylor Dome, in contrast, does not show such a steep age-depth relationship (Figure 5c).

Our new data also show that the ice at the MIS 5/4 site is stratigraphically linked to the Main Transect. The evidence for this is that the -380 m core contains air from late MIS 4 and the MIS 4/3 transition (Figure 3). The existence of MIS 4 ice on the Main Transect suggests continuity between the two archives, i.e. that both archives originated from the same accumulation zone. This is important because it means that it is possible to compare climate information from the new MIS 5/4 site to climate information from different intervals (e.g. the LGM) in ice from the Main Transect. More broadly speaking, it is important to note that geologic evidence from Taylor Valley suggests that Taylor Glacier has not changed dramatically in terms of its extent or its thickness in the last ~ 2.2 Ma, and that Taylor Dome has remained a peripheral dome of the East Antarctic Ice Sheet through the last ice age (Marchant et al., 1994; Brook et al., 1993). It is therefore unlikely that the location of the Taylor Glacier accumulation zone drastically changed during the intervals preserved in the Main Transect and the MIS 5/4 site (~ 77 to 7 ka).

A final observation is that the MIS 5/4 ice cores from Taylor Glacier have very low $\delta^{15}N$-$N_2$ (Figure 5b). The $\delta^{15}N$-$N_2$ enclosed in ice core air bubbles is controlled primarily by gravitational fractionation in the firn column (Sowers et al., 1992) (Supplementary Information). To first order the $\delta^{15}N$-$N_2$ records the height of the diffusive air column (Sowers et al., 1992), an estimate for total firn thickness. $\delta^{15}N$-$N_2$ is also influenced by convective mixing near the top of the firn (Kawamura et al., 2006; Severinghaus et al., 2010) and vertical gradients in firn temperature induced by rapid shifts in ambient temperature (Severinghaus et al., 1998). Low $\delta^{15}N$-$N_2$ (< 0.1 ‰) has been previously observed at Taylor Glacier (e.g., Main Transect position -125 m) and Taylor Dome (e.g., 380-390 m) and could result from thin firn and/or deep air convection (Baggenstos et al., 2018; Severinghaus et al., 2010; Sucher, 1997). The observation that $\delta^{15}N$-$N_2$ in the -380 m core is similarly low as $\delta^{15}N$-$N_2$ in the MIS 5/4 core supports our interpretation that the archives originated from the same deposition site (Figure 3).

**4.2 Gas Age-Ice Age Difference (Δage)**

Gas is trapped in air bubbles in firn at polar sites typically 50-120 m below the surface, and thus ice core air is younger than the ice matrix that encloses it (Schwander and Stauffer, 1984). The magnitude of the difference between ice age and gas age (Δage) depends primarily on temperature and accumulation rate, with accumulation having a stronger control (Herron and Langway, 1980; Parrenin et al., 2012; Capron et al., 2013). Δage ranges from 100-3000 years in polar ice cores under modern conditions (Schwander and Stauffer, 1984), with high-accumulation sites having the smallest Δage (e.g., Buizert et al., 2015; Etheridge et al., 1996) due to fast advection of firn to the lock-in depth where gases no longer mix with the overlying pore space. Extrema in Δage up to 6500 years (Vostok) and 12,000 years (Taylor Dome) have been documented for cold, low-accumulation sites at the last glacial maximum (e.g., Veres et al., 2013; Bender et al., 2006; Baggenstos et al., 2018), where slow grain metamorphism and slow advection of firn increase the lock-in time. Other important factors may include ice impurity content (Horhold et al., 2012; Freitag et al., 2013; Breant et al., 2017), surface wind stress, local summer insolation (Kawamura et al., 2007), and

firn thinning. These factors are of secondary importance for polar ice cores compared to the effects of temperature and accumulation rate (Buizert et al., 2015).

$\Delta$age was calculated for the new Taylor Glacier ice core by subtracting the gas age at a given depth from the independently determined ice age at the same depth ($\Delta$age = ice age – gas age). The $\Delta$age in the Taylor Glacier MIS 5/4 core approaches ~10 ka during late MIS 4 (Figure 5b), which exceeds $\Delta$age for typical modern polar ice core sites even where ice accumulates very slowly. This finding is unprecedented in ice from Taylor Glacier, as $\Delta$age in ice from the Main Transect does not exceed ~ 3 ka between 10-50 ka (Baggenstos et al., 2018). Our large $\Delta$age values imply that accumulation in the Taylor Glacier accumulation zone decreased significantly through MIS 4, which could have been caused by low precipitation and/ or high wind scouring. This interpretation is supported by the following lines of evidence: (1) The depth-age relationship suggests the ice during MIS 4 is very thin (Figure 5a). This is in contrast to ice from the last glacial maximum, which is found at the surface of Taylor Glacier in two thicker (layer thickness = ~ 50 m) outcrops that dip approximately vertically and strike along the glacier longitudinally (Baggenstos et al., 2017; Aciego et al., 2007). Thin MIS 4 layers could be due to mechanical thinning of the ice rather than low accumulation rates. However, we note that ice thinning does not alter $\Delta$age because $\Delta$age is fixed at the bottom of the firn when the ice matrix encloses bubbles (Parrenin et al., 2012). This is unlike $\Delta$depth, the depth difference between ice and gas of the same age, which evolves with thinning. So even if increased thinning caused the steep depth-age curve observed during MIS 4, one would still need to invoke an explanation for high $\Delta$age. (2) There is some degree of smoothing in the Taylor Glacier $CH_4$ data relative to EDML, which can result from the expected longer gas trapping duration in firn where accumulation rates are relatively low (Kohler et al., 2011; Fourteau et al., 2017; Spahni et al., 2003). (3) As $\Delta$age increased at the onset of MIS 4, the $\delta^{15}$N-$N_2$ progressively decreased (Figure 5b), which is consistent with thinning of the firn column in response to decreased net accumulation. Inspection of Figure 5b reveals that the change in $\delta^{15}$N-$N_2$ is not linear with $\Delta$age, potentially due to non-gravitational effects like thermal fractionation (Severinghaus et al., 1998) or convective mixing near the top of the firn (Kawamura et al., 2006). Very low accumulation rate is known to be associated with deep convective mixing in the firn (Severinghaus et al., 2010).

In contrast to Taylor Glacier, $\Delta$age at Taylor Dome reaches a maximum of 3 ka at ~ 56 ka and does not rise above 2.5 ka throughout MIS 4 (Figure 5d). The implication of the relatively "normal" $\Delta$age is that net accumulation at Taylor Dome did not dramatically change throughout MIS 4 while $\Delta$age in the Taylor Glacier accumulation region did.

$\Delta$age uncertainty was determined by propagating the error reported for the age models described above (Figures 5a and 5c). The maximum and minimum $\Delta$age curves were calculated by subtracting the oldest gas age scale from the youngest ice age scale and vice versa. The mean $\Delta$age uncertainty is ± 2.2 ka for the

Taylor Glacier MIS 5/4 cores and + 1.0 ka/ - 1.3 ka for the Taylor Dome core. The larger uncertainty for Taylor Glacier is due to the larger age uncertainties arising from the depth error. The uncertainties we estimate for Δage are of similar magnitude to the Δage uncertainty in other Taylor Glacier chronologies (Baggenstos et al., 2018).

### 4.3 Accumulation rate estimates

Given mean annual temperature and Δage, it is possible to use models of firn densification to estimate the accumulation rate at the Taylor Glacier accumulation zone. We used an empirical firn densification model (Herron and Langway, 1980) to compute firn density profiles for a range of temperatures and mean accumulation rates (Supplementary Information). Δage in the model is estimated by calculating the age of the firn when it has reached the close-off depth (when the density = 0.83 g cm$^{-3}$). The estimated accumulation rate comes from a simple lookup function that scans the full range of temperature and Δage and picks the corresponding accumulation rate (similar to work by Parrenin et al. (2012)). For Δage = 10 ka and temperature = -46 ℃ the estimated accumulation rate for the Taylor Glacier MIS 5/4 cores is 1.9 mm yr$^{-1}$ ice equivalent. The temperature -46 ℃ is derived from the average $\delta^{18}O_{ice}$ for the period of firn densification (70-60 ka) using the relationship $\Delta\delta^{18}O_{ice} = 0.5$ °C$^{-1}$ calibrated using modern $\delta^{18}O_{ice} = -41$ ‰ and modern temperature = -43 °C (Waddington and Morse, 1994; Steig et al., 2000), similar to Baggenstos et al. (2018). We used the average $\delta^{18}O_{ice}$ from the Taylor Dome record because it is less noisy and avoids the question of whether Taylor Glacier $\delta^{18}O_{ice}$ accurately records temperature (Baggenstos et al., 2018). Since the close-off depth is estimated from the modeled firn density profile (30 m), it is possible to estimate the expected $\delta^{15}N-N_2$ assuming that the close-off depth is an approximation of the height of the diffusive air column (Supplementary Information). Assuming a 3 m lock-in zone height and a 0 m convective zone height (see Supplementary Information), the predicted $\delta^{15}N-N_2$ (0.14 ‰) is enriched by a factor of 2 relative to measured values (~ 0.07 ‰ at 60 ka, Figure 5b). The difference in expected versus measured $\delta^{15}N-N_2$ may imply the influence of deep air convection in the Taylor Glacier firn column (Kawamura et al., 2006; Severinghaus et al., 2010). To bring the predicted $\delta^{15}N-N_2$ into closer agreement we introduced a convective zone height of 13.5 m (Figure S7). The apparent influence of air convection could be due to cracks that penetrate the surface of the firn (e.g., Severinghaus et al., 2010), which only occur in firn with a low mean accumulation rate.

A similar estimate was performed for the Taylor Dome core. Running the models with Δage = 2.3 ka (the Taylor Dome Δage at ~ 60 ka when Taylor Glacier Δage is maximum, Figure 5) and temperature = -46 ℃ yields an estimated mean accumulation rate of 1.6 cm yr$^{-1}$ ice equivalent, almost a factor of 10 larger than Taylor Glacier. The estimated diffusive column height (53 m) with a 3 m lock-in zone height and 0 m convective zone height predicts $\delta^{15}N-N_2$ of 0.26 ‰ (Figure S8), in somewhat better agreement with measured $\delta^{15}N-N_2$ (Figure 5d), implying less influence of deep air convection. The $\delta^{15}N-N_2$ data from Taylor Dome are lower-resolution and less precise than the new Taylor Glacier data; in fact there is not

actually a $\delta^{15}$N-N$_2$ measurement at 60 ka (Figure 5d). Still, we think the closer agreement between modeled $\delta^{15}$N-N$_2$ and the nearest measured $\delta^{15}$N-N$_2$ suggests a shallower convective zone, consistent with higher mean accumulation rate.

These accumulation rate and firn thickness calculations estimate how low the accumulation at Taylor Glacier may have been relative to Taylor Dome in late MIS 4. We caution that these estimates are uncertain given that we extrapolated below the empirical calibration range of the firn densification model (lowest accumulation = 2.4 cm yr$^{-1}$ ice equivalent at Vostok) (Herron and Langway, 1980). We are unaware of firn densification models that are specifically tailored to very low-accumulation sites. Another potential

uncertainty in our estimates is that we did not account for geothermal heat transfer through the firn, which is relatively close to bedrock at Taylor Dome (depth to bedrock = $\sim$ 550 m). The effect of excess geothermal heat would drive firn temperatures higher, decreasing $\Delta$age (Goujon et al., 2003). Higher firn temperatures could also cause lower $\delta^{15}$N-N$_2$, perhaps partially explaining low values of $\delta^{15}$N-N$_2$ observed at Taylor Glacier and Taylor Dome.

**5 Discussion**

Despite the model uncertainties, we conclude that the simplest explanation of the $\Delta$age patterns described above is markedly different accumulation rates in the Taylor Dome versus Taylor Glacier accumulation zones during MIS 4. Today the Taylor Glacier accumulation zone is on the northern flank of Taylor Dome,

whereas the Taylor Dome ice core site is on the south flank (Figure 1). The difference between the estimated accumulation rate at Taylor Glacier versus Taylor Dome implies a gradient in precipitation and/ or wind scouring between the two locations. This implication is perhaps not surprising given that a modern accumulation gradient is observed in the same direction, with accumulation decreasing from 14 cm yr$^{-1}$ to 2 cm yr$^{-1}$ going from south to north (Morse et al., 1999; Morse et al., 2007; Kavanaugh et al., 2009b).

Moisture delivery to Taylor Dome primarily occurs during storms that penetrate the Transantarctic Mountains south of the Royal Society Range and reach Taylor Dome from the south (Morse et al., 1998), therefore the modern-day accumulation rate decreases orographically from south to north. The Taylor Glacier accumulation zone is effectively situated on the lee side of Taylor Dome with respect to the modern prevailing storm tracks (Morse et al., 1999) (Figure 1). The difference between $\Delta$age at Taylor Glacier

versus Taylor Dome is too large to be explained by temperature contrasts between the two sites, which are on the order of 1-3 °C in the present day (Waddington and Morse, 1994).

A temporal change in the accumulation gradient across Taylor Dome (and hence between Taylor Dome and the Taylor Glacier accumulation zone) has already been suggested by other work, for the last glacial

maximum. Morse et al. (1998) calculated the accumulation rate history for the Taylor Dome ice core site using modern accumulation data, a calculated ice flow field, and an age scale determined by correlation of isotope and chemical data to Vostok ice core records (Figure 6). By mapping the Taylor Dome age scale to

ice layers resolved in radar stratigraphy, Morse et al. (1998) also inferred the accumulation rate history for a virtual ice core situated in the lee of the modern prevailing storm trajectory, ~7 km to the north of the Taylor Dome drill site and likely near the hypothesized Taylor Glacier accumulation zone (Figure 1).

The accumulation histories inferred from the layer thicknesses revealed differences for the two sites, but not in the direction expected from the modern south-to-north storm trajectory. The last glacial maximum accumulation histories were characterized by extremely low accumulation at the Taylor Dome ice core site relative to higher accumulation at the northern virtual ice core site. The possibility that different layer thicknesses (and inferred accumulation histories) were a result of differential ice flow was rejected because
deeper layers did not show the same effect (Morse et al., 1998). The reversed accumulation gradient inferred from ice layer thicknesses was qualitatively confirmed by independent Δage determinations on Taylor Glacier and Taylor Dome ice made by Baggenstos et al. (2018), which revealed a Taylor Glacier Δage = ~ 3000 years and Taylor Dome Δage = ~12,000 years at the last glacial maximum. Accumulation rate estimates from a firn densification model (Figure 6) confirmed that the orientation of the accumulation
gradient was north-to-south, in the opposite direction of the gradient observed today (Figure 1).

Our new Δage data and accumulation rate estimates indicate an accumulation gradient in the same direction as the modern, but opposite that of the last glacial maximum. The accumulation rate estimates by Morse et al. (1998) qualitatively agree with this pattern > 60 ka (Figure 6). It is hypothesized that the reversed
accumulation gradient at the last glacial maximum resulted from a shift in the trajectory of storm systems that delivered moisture to Taylor Dome, possibly in response to the extension of grounded ice far into the Ross Sea (Morse et al., 1998). If indeed the Antarctic ice sheet extended far enough into the Ross Sea to alter the atmospheric circulation during the last glacial maximum, the implication of our new data is that a similar situation did not exist during MIS 4. This hypothesis seems at odds with independent evidence that
the Southern Hemisphere experienced full glacial conditions during MIS 4 (Schaefer et al., 2015; Barker and Diz, 2014). A possible explanation is that the sea level minimum at MIS 4 was 25 m higher than during the last glacial maximum due to the lack of extensive Northern Hemisphere ice sheets (Shakun et al., 2015; Siddall et al., 2003; Cutler et al., 2003), which limited how far grounded ice from the West Antarctic Ice Sheet could extend into the Ross Embayment. This suggestion is consistent with (1) data suggesting the
maximum Ross Ice Shelf extent occurred during the last glacial termination (Hall et al., 2015; Denton and Hughes, 2000) rather than MIS 4, and (2) the notion that grounding line position in the Ross Sea is set by the balance between marine forcing (basal melting) and accumulation on the Antarctic ice sheets (Hall et al., 2015).

A second hypothesis arises from the notion that broad differences in regional atmospheric dynamics between MIS 4 and the last glacial maximum might occur, without invoking changes in the extent of the Ross Ice Shelf as a mechanism for disrupting the atmospheric circulation. The Amundsen Sea Low, a low-

pressure center that influences the Ross Sea and Amundsen Sea sectors of Antarctica, responds strongly to changes in tropical climate (Raphael et al., 2016; Turner et al., 2013) and exhibits cyclonic behavior that likely controls the path of storms that enter the Ross Embayment and reach Taylor Dome, as implied by Morse et al. (1998) and explored by Bertler et al., (2006). An intensified or shifted Amundsen Sea Low during MIS 4 relative to the last glacial maximum might result in strong meridional flow across Taylor Dome that maintained a south-to-north orographic precipitation gradient. Interestingly, variability in the Amundsen Sea Low has been linked to the extent of Northern Hemisphere ice sheets (Jones et al., 2018), which were smaller in extent at MIS 4 relative to the last glacial maximum. In summary, the anomalous accumulation gradients we document on Taylor dome in MIS 4 may have their origin in the modest Northern Hemisphere ice volume at that time.

## 6 Conclusions

We obtained the first ice core from the Taylor Glacier blue ice area that contains air with ages unambiguously spanning the MIS 5/4 transition and the MIS 4/3 transition (74.0-57.7 ka). The ice core also contains ice spanning the MIS 5/4 transition and MIS 4 (76.5-60.6 ka). The gas age-ice age difference ($\Delta$age) in the cores approaches 10,000 years during MIS 4 implying extremely arid conditions with very low net accumulation at the site of snow deposition. To the south of the Taylor Glacier accumulation zone, the Taylor Dome ice core exhibits lower $\Delta$age (1000-2500 years) during the same time interval. This implies a steep accumulation rate gradient across the Taylor Dome region with precipitation decreasing toward the north and/or extreme wind scouring affecting the northern flank. The direction of the gradient suggests that the trajectory of storms was south-to-north during MIS 4 and that storm paths were not disrupted by Antarctic ice protruding into the Ross Sea or by changes in the strength and/or position of the Amundsen Sea Low, as occurred at the last glacial maximum.

Data will be made available through the US Antarctic Program Data Center and the National Center for Environmental Information.

## Author contributions

JAM made measurements at OSU on Taylor Glacier samples, developed chronologies, and prepared the manuscript; JAM, EJB, and JRM made measurements in the field; TKB made measurements at OSU on the -380 m Taylor Glacier core; SB and SM made measurements at OSU on Taylor Dome samples; SAS made measurements on all new Taylor Glacier samples at SIO except the -380 m core, which were made by DB; JRM made measurements on Taylor Glacier samples at DRI; all authors provided valuable feedback and made helpful contributions to writing the manuscript.

The authors declare no conflicts of interest.

**Acknowledgements**

This work was supported by US NSF grant 1245821 to EJB, US NSF grant PLR-1245659 to VP, US NSF grant 1246148 to JS, and UK NERC grant 502625 to SB.

We thank Mike Jayred for maintaining and operating the blue ice drill and Kathy Schroeder and Chandra Llewellyn for managing the Taylor Glacier field camp; both tasks were Herculean. We thank Peter Sperlich, Isaac Vimont, Peter Neff, Heidi Roop, Bernhard Bereiter, Jake Ward, and Andrew M. Smith for help with field logistics and drilling, sampling, and packing ice cores. We thank Howard Conway and Ed Waddington for feedback on an early version of the manuscript, and Christo Buizert and Justin Wettstein

for helpful conversations about the Amundsen Sea Low. We thank Michael Kalk, Aron Buffen, and Michael Rebarchik for laboratory assistance at Oregon State University, and Monica Arienzo and Nathan Chellman for operation of the continuous melter and other instrumentation at Desert Research Institute. We thank the United States Antarctic Program with particular thanks to Science Cargo, the BFC, and Helicopter Operations. We also thank Brian Eisenstatt and Duncan May for ensuring the delivery of critical

supplies to the glacier at critical times during both Antarctic field seasons.

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

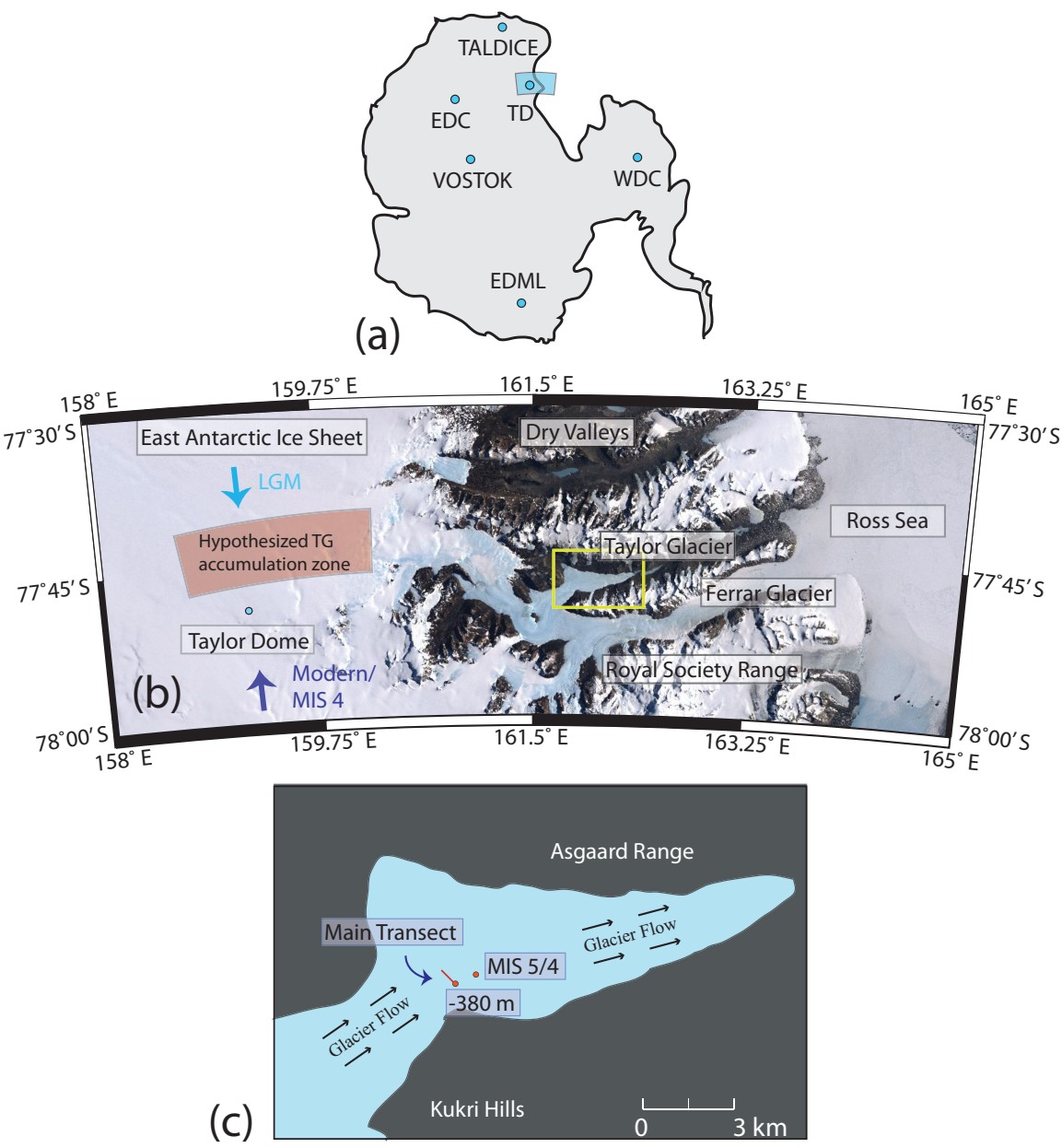

Figure 1 – (a) The locations of ice core sites discussed in this text are indicated with blue dots on the continent outline (EDC = EPICA Dome C, EDML = EPICA Dronning Maud Land, TALDICE = Talos Dome ice core, TD = Taylor Dome, WDC = West Antarctic Ice Sheet Divide core). (b) Landsat imagery of Taylor Valley (Bindschadler et al., 2008). Blue arrows conceptually show the modern storm trajectory as well as the hypothesized storm trajectories for the last glacial maximum (LGM) and the Marine Isotope Stage (MIS) 4 discussed later in the text. (c) Simplified map of Taylor Glacier showing main transect (red line) containing ice spanning the Holocene-MIS 3 time period and drill sites discussed in the text (red dots).

Table 1 – Summary of new datasets. Gas chromatograph (GC) and mass spectrometer (MS) measurements were made on discrete samples. Picarro, Abakus, and ICP-MS measurements were made by continuous-flow analysis. Analytical precision is from method reference or pooled standard deviation of replicate samples. OSU = Oregon State University, SIO = Scripps Institution of Oceanography, DRI = Desert Research Institute.

| Dataset | Drill Site | Ice Drill | Season Extracted | Approx. Depth Range | Location Measured | Instrume-ntation* | Analytical Precision (1 σ) |
|---|---|---|---|---|---|---|---|
| $CH_4$ | Taylor Dome | GISP2 | 1993-1994 | 455-505 m | OSU | GC[1] | 3.5 ppb |
| $CO_2$ | Taylor Dome | GISP2 | 1993-1994 | 455-505 m | OSU | GC[2] | 1.5 ppm |
| $CH_4$ | -380 m MT | PICO | 2013-2014 | 4-15 m | OSU | GC[1] | 3.5 ppb |
| $CO_2$ | -380 m MT | PICO | 2013-2014 | 4-15 m | OSU | GC[2] | 1.5 ppm |
| $\delta^{18}O_{atm}$ | -380 m MT | PICO | 2013-2014 | 4-15 m | SIO | MS[3] | 0.011 ‰ |
| $\delta^{15}N$ | -380 m MT | PICO | 2013-2014 | 4-15 m | SIO | MS[3] | 0.0028 ‰ |
| $CH_4$ | MIS 5/4 | PICO | 2014-2015 | 2-17 m | Field | GC[1] | 10 ppb |
| $CH_4$ | MIS 5/4 | BID | 2014-2015 | 9-17 m | OSU | GC[1] | 3.5 ppb |
| $CO_2$ | MIS 5/4 | BID | 2014-2015 | 9-17 m | OSU | GC[2] | 1.5 ppm |
| $CO_2$ | MIS 5/4 | BID | 2014-2015 | 9-17 m | OSU | MS[4] | 1.5 ppm |
| $\delta^{18}O_{atm}$ | MIS 5/4 | BID | 2014-2015 | 9-17 m | SIO | MS[3] | 0.011 ‰ |
| $\delta^{15}N$ | MIS 5/4 | BID | 2014-2015 | 9-17 m | SIO | MS[3] | 0.0028 ‰ |
| $CH_4$ | MIS 5/4 | BID | 2015-2016 | 0-20 m | Field | Picarro[5] | 2.8 ppb |
| Insoluble Particles | MIS 5/4 | BID | 2015-2016 | 0-20 m | Field | Abakus[7] | |
| $CO_2$ | MIS 5/4 | BID | 2015-2016 | 4-9 m, 17-20 m | OSU | MS[4] | 1.5 ppm |
| $\delta^{18}O_{atm}$ | MIS 5/4 | BID | 2015-2016 | 4-9 m, 17-20 m | SIO | MS[3] | 0.011 ‰ |
| $\delta^{15}N$ | MIS 5/4 | BID | 2015-2016 | 4-9 m, 17-20 m | SIO | MS[3] | 0.0028 ‰ |
| $CH_4$ | MIS 5/4 | BID | 2015-2016 | 0-20 m | DRI | Picarro[5] | 2.8 ppb |
| $\delta^{18}O_{ice}$ | MIS 5/4 | BID | 2015-2016 | 0-20 m | DRI | Picarro[6] | |
| Insoluble Particles | MIS 5/4 | BID | 2015-2016 | 0-20 m | DRI | Abakus[7] | |
| $Ca^{2+}$ | MIS 5/4 | BID | 2015-2016 | 0-20 m | DRI | ICP-MS[7] | ± 3 % |

*Superscripts denote references for analytical procedures: [1] (Mitchell et al., 2013; Mitchell et al., 2011); [2] (Ahn et al., 2009); [3] (Severinghaus et al., 1998; Petrenko et al., 2006); [4] (Bauska et al., 2014); [5] (Rhodes et al., 2013); [6] (Maselli et al., 2013); [7] (McConnell, 2002).

Table 2 – Tie points relating Taylor Glacier depth to gas age on the AICC 2012 timescale. Gray shading indicates tie points < 4 m depth where abundant cracks in shallow ice may cause contamination of gas records (see text). "DO" refers to Dansgaard-Oeschger event.

| Depth (m) | Gas Age (ka) | Age Range (ka) | Data | Data Source | Feature Description | Reference Record | Tie Point Source |
|---|---|---|---|---|---|---|---|
| 1.74 | 58.21 | 57.30-59.00 | $CH_4$ | This study | Peak during DO 16/17 | EDML $CH_4$ | This study |
| 3.15 | 59.10 | 58.21-59.60 | $CH_4$ | This study | Peak during DO 6/17 | EDML $CH_4$ | This study |
| 4.19 | 59.66 | 59.60-59.70 | $CH_4$ | This study | Midpoint transition DO 16/17 | EDML $CH_4$ | This study |
| 5.40 | 59.94 | 59.71-60.78 | $CH_4$ | This study | Low before DO 16/17 | EDML $CH_4$ | This study |
| 7.79 | 64.90 | 64.30-65.40 | $CH_4$ | This study | Peak during DO 18 | EDML $CH_4$ | This study |
| 11.24 | 69.92 | 69.00-70.36 | $CH_4$ | This study | Small peak between DO 19 and DO 18 | EDML $CH_4$ | This study |
| 12.43 | 70.62 | 70.25-71.10 | $CH_4$ | This study | Low after DO 19 | EDML $CH_4$ | This study |
| 13.25 | 71.21 | 70.94-71.42 | $CH_4$ | This study | High before transition late DO 19 | EDML $CH_4$ | This study |
| 16.20 | 72.27 | 72.10-72.45 | $CH_4$ | This study | Midpoint transition DO 19 | EDML $CH_4$ | This study |
| 17.40 | 72.70 | 72.20-73.30 | $\delta^{18}O_{atm}$ | This study | Midpoint transition | NGRIP $\delta^{18}O_{atm}$ | This study |
| 19.27 | 73.74 | 73.35-74.50 | $\delta^{18}O_{atm}$ | This study | Low before transition | NGRIP $\delta^{18}O_{atm}$ | This study |

Table 3 - Tie points relating Taylor Glacier depth to ice age on the AICC 2012 timescale. Gray shading indicates tie points < 0.4 m depth where abundant cracks in shallow ice may cause contamination of dust measurements (see text). Ice phase parameters (dust and $\delta^{18}O_{ice}$) are unaffected by surface cracks below 0.4 m depth. "AIM" refers to Antarctic Isotope Maximum event, and "MIS" refers to Marine Isotope Stage.

| Depth (m) | Ice Age (ka) | Age Range (ka) | Data | Data Source | Feature Description | Reference Record | Tie Point Source |
|---|---|---|---|---|---|---|---|
| 0.34 | 61.47 | 59.50-63.93 | Insoluble particles | This study | Peak near end of MIS 4 | EDC laser dust | This study |
| 1.25 | 63.93 | 63.00-64.70 | $nssCa^{2+}$ | This study | Peak late MIS 4 | EDC $nssCa^{2+}$ | This study |
| 1.80 | 64.91 | 64.00-65.65 | Insoluble particles | This study | Peak late MIS 4 | EDC laser dust | This study |
| 2.45 | 65.65 | 65.00-66.30 | Insoluble particles | This study | Peak mid MIS 4 | EDC laser dust | This study |
| 3.10 | 66.73 | 66.10-67.40 | $nssCa^{2+}$ | This study | Peak mid MIS 4 | EDC $nssCa^{2+}$ | This study |
| 4.47 | 68.63 | 67.86-69.60 | $nssCa^{2+}$ | This study | Peak mid MIS 4 | EDC $nssCa^{2+}$ | This study |
| 4.94 | 69.72 | 69.30-70.10 | $nssCa^{2+}$ | This study | Low early MIS 4 | EDC $nssCa^{2+}$ | This study |
| 5.60 | 70.20 | 69.70-70.65 | $nssCa^{2+}$ | This study | Peak early MIS 4 | EDC $nssCa^{2+}$ | This study |
| 7.75 | 71.95 | 71.00-73.00 | $\delta^{18}O_{ice}$ | This study | Peak AIM 19 | EDC $\delta^{18}O_{ice}$ | This study |
| 12.20 | 73.62 | 73.00-74.50 | $\delta^{18}O_{ice}$ | This study | Low between AIM 19 and AIM 20 | EDC $\delta^{18}O_{ice}$ | This study |
| 16.62 | 75.75 | 74.60-76.75 | $\delta^{18}O_{ice}$ | This study | Peak AIM 20 | EDC $\delta^{18}O_{ice}$ | This study |
| 19.76 | 76.50 | 75.75-77.00 | $nssCa^{2+}$ | This study | End of record, loosely constrained | EDC $nssCa^{2+}$ | This study |

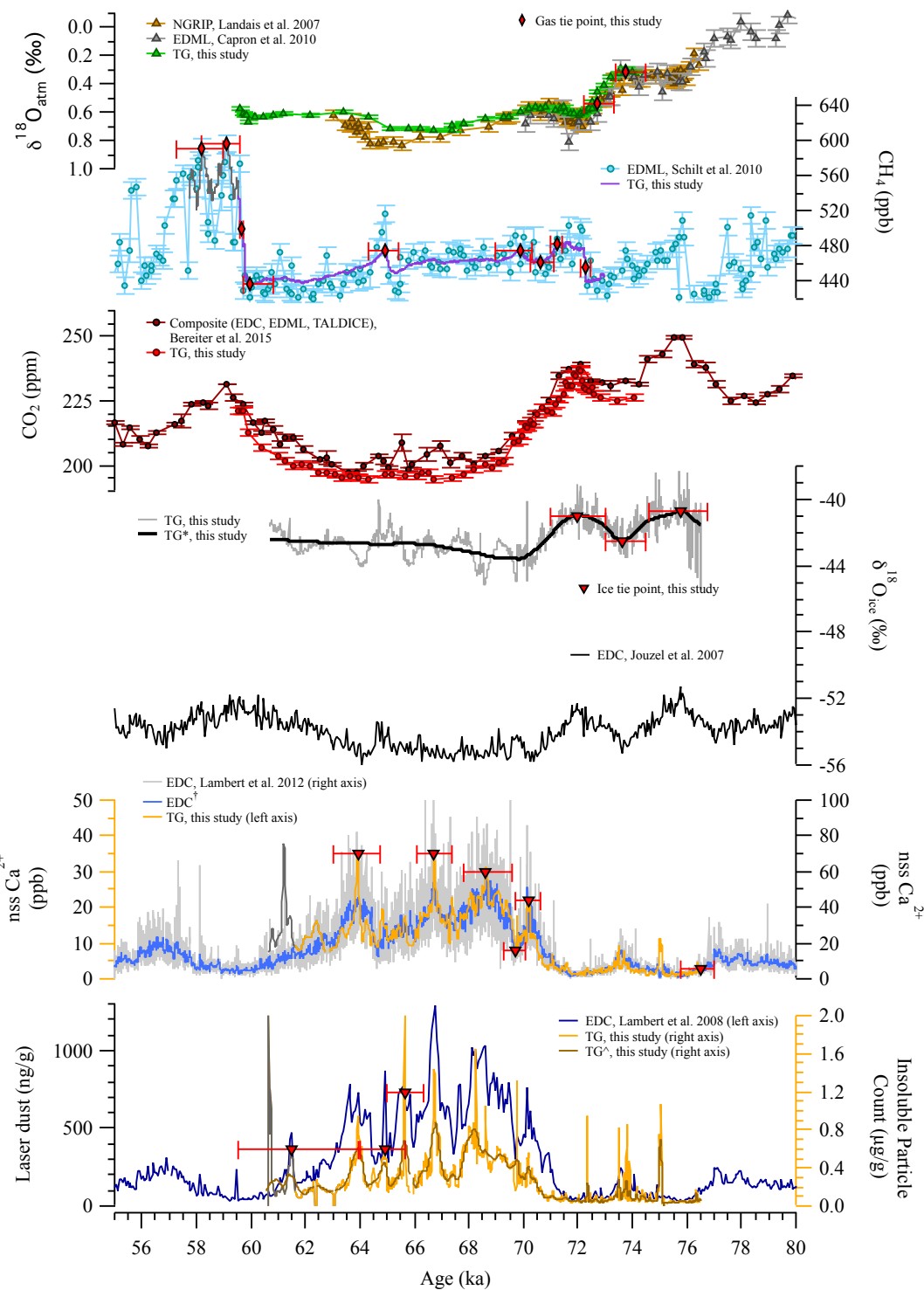

Figure 2 – Measurements of trace gases ($CH_4$ and $CO_2$), stable isotopes (ice and $O_2$), insoluble particles, and nss-$Ca^{2+}$ from the Taylor Glacier ice core on new gas and ice age scales. All ice core data are synchronized to AICC 2012. $CH_4$ data from < 4 m depth and dust data from < 40 cm depth are colored dark gray to denote potential contamination by surface cracks. NGRIP = North Greenland Ice Coring Project, TG = Taylor Glacier MIS 5/4 BID cores, EDML = EPICA Dronning Maud Land, EDC = EPICA Dome C, TALDICE = Talos Dome. *, †, and ^ denote smoothing with 5000 point, 100 point, and 50 point LOESS algorithms, respectively.

Table 4 – Tie points relating -380 m Main Transect core depth to gas age on the AICC 2012 timescale.

| Depth (m) | Gas Age (ka) | Data | Data Source | Feature Description | Reference Record | Tie Point Source |
|---|---|---|---|---|---|---|
| 3.751 | 59.53 | $CH_4$ | This study | High value at start of DO 16/17 | EDML $CH_4$ | This study |
| 5.301 | 59.83 | $CH_4$ | This study | Low before DO 16/17 | EDML $CH_4$ | This study |
| 9.929 | 64.40 | $CH_4$ | This study | Low after DO 18 | EDML $CH_4$ | This study |
| 14.849 | 66.00 | $CH_4$ | This study | Low before DO 18 | EDML $CH_4$ | This study |

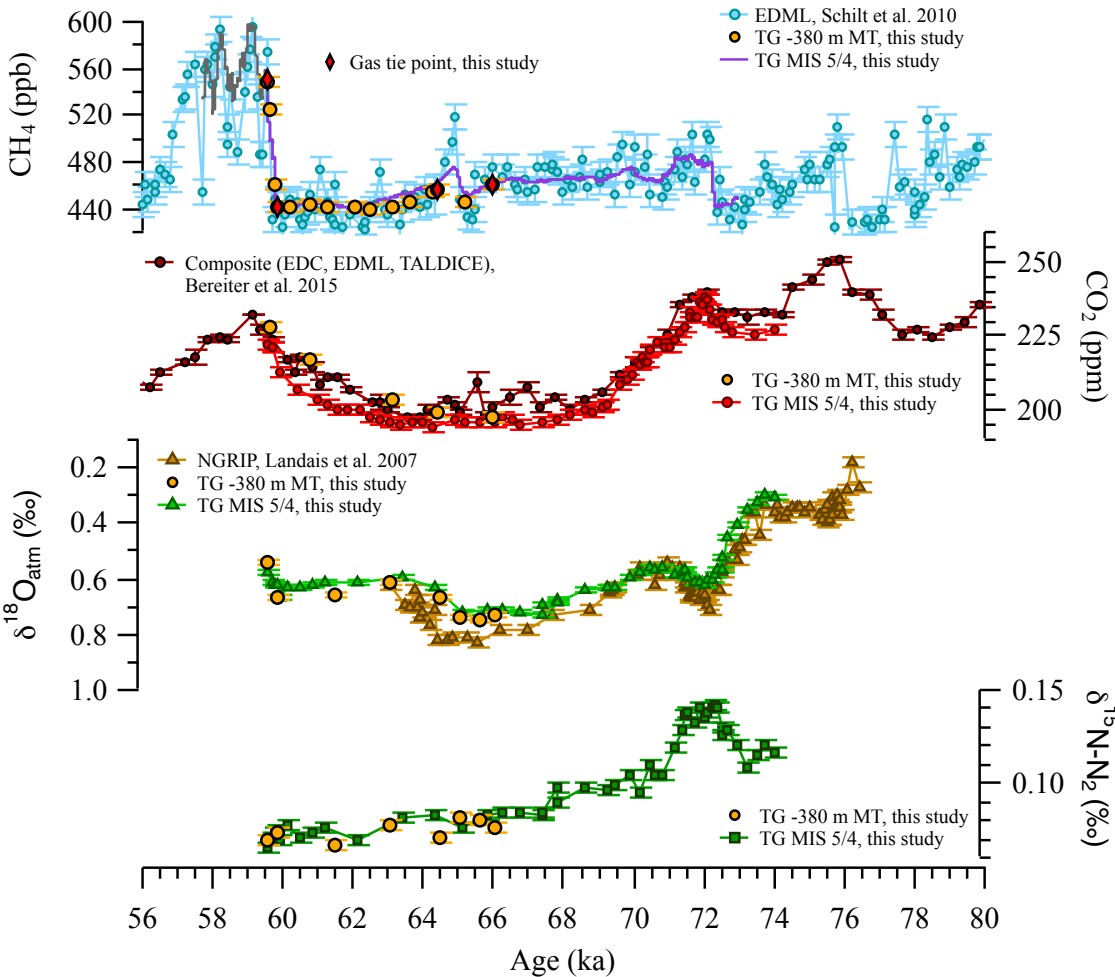

Figure 3 - Measurements of trace gases ($CH_4$ and $CO_2$), and stable isotopes ($O_2$ and $N_2$) from the -380 m Main Transect Taylor Glacier ice core and MIS 5/4 ice cores on new gas age scales. All ice core data are synchronized to AICC 2012. $CH_4$ data from < 4 m depth are colored gray to denote potential contamination by surface cracks. NGRIP = North Greenland Ice Coring Project, TG = Taylor Glacier, EDML = EPICA Dronning Maud Land, EDC = EPICA Dome C, TALDICE = Talos Dome.

Table 5 – Tie points relating Taylor Dome depth to gas age on the AICC 2012 timescale.

| Depth (m) | Gas Age (ka) | Age Range (ka) | Data | Data Source | Feature Description | Reference Record | Tie Point Source |
|---|---|---|---|---|---|---|---|
| 455.95 | 54.667 | 54.167-55.167 | $CO_2$ | Indermühle et al. 2000 | Midpoint transition A3 | WAIS $CO_2$ | Baggenstos et al. 2018 |
| 460.90 | 57.913 | 57.413-58.413 | $CO_2$ | Indermühle et al. 2000 | Midpoint transition A4 | WAIS $CO_2$ | Baggenstos et al. 2018 |
| 464.62 | 59.99 | 59.70-60.50 | $CH_4$ | Brook et al. 2000 | Low before DO 16/17 | EDML $CH_4$ | This study |
| 467.10 | 62.303 | 61.803-62.803 | $CO_2$ | Indermühle et al. 2000 | Midpoint transition A4 | WAIS $CO_2$ | Baggenstos et al. 2018 |
| 474.95 | 65.50 | 65.00-66.80 | $CH_4$ | This study | Low before DO 18 | EDML $CH_4$ | This study |
| 483.10 | 70.40 | 69.70-71.20 | $CH_4$ | This study | Low $CH_4$ after DO 19 | EDML $CH_4$ | This study |
| 486.95 | 72.27 | 72.00-72.70 | $CH_4$ | This study | Midpoint transition DO 19 | EDML $CH_4$ | This study |
| 493.50 | 76.05 | 75.75-76.30 | $CH_4$ | This study | Midpoint transition DO 20 | EDML $CH_4$ | This study |
| 503.90 | 83.90 | 83.65-84.10 | $CH_4$ | This study | High at DO 21 onset | EDML $CH_4$ | This study |

Table 6 - Tie points relating Taylor Dome depth to ice age on the AICC 2012 timescale.

| Depth (m) | Ice Age (ka) | Age Range (ka) | Data | Data Source | Feature Description | Reference Record | Tie Point Source |
|---|---|---|---|---|---|---|---|
| 455.10 | 55.80 | 54.25-57.00 | $Ca^{2+}$ | Mayewski et al. 1996 | See original work | WAIS $Ca^{2+}$ | Baggenstos et al. 2018 |
| 457.60 | 58.85 | 57.50-60.10 | $Ca^{2+}$ | Mayewski et al. 1996 | See original work | WAIS $Ca^{2+}$ | Baggenstos et al. 2018 |
| 463.30 | 61.47 | 61.00-62.00 | $Ca^{2+}$ | Mayewski et al. 1996 | Peak late MIS 4 | EDC laser dust | This study |
| 466.40 | 63.50 | 62.80-63.75 | $Ca^{2+}$ | Mayewski et al. 1996 | See original work | WAIS $Ca^{2+}$ | Baggenstos et al. 2018 |
| 467.80 | 64.30 | 63.90-64.80 | $Ca^{2+}$ | Mayewski et al. 1996 | See original work | WAIS $Ca^{2+}$ | Baggenstos et al. 2018 |
| 468.10 | 64.66 | 64.20-65.40 | $Ca^{2+}$ | Mayewski et al. 1996 | Low late MIS 4 | EDC $nssCa^{2+}$ | This study |
| 471.37 | 65.57 | 65.00-66.10 | $Ca^{2+}$ | Mayewski et al. 1996 | Peak mid MIS 4 | EDC laser dust | This study |
| 472.70 | 66.71 | 66.00-67.25 | $Ca^{2+}$ | Mayewski et al. 1996 | Peak mid MIS 4 | EDC $nssCa^{2+}$ | This study |
| 475.12 | 67.47 | 67.00-68.00 | $Ca^{2+}$ | Mayewski et al. 1996 | Low mid MIS 4 | EDC $nssCa^{2+}$ | This study |
| 476.90 | 68.63 | 67.75-69.40 | $Ca^{2+}$ | Mayewski et al. 1996 | Peak early MIS 4 | EDC $nssCa^{2+}$ | This study |
| 478.70 | 69.70 | 69.25-70.10 | $Ca^{2+}$ | Mayewski et al. 1996 | Low early MIS 4 | EDC $nssCa^{2+}$ | This study |
| 479.90 | 70.15 | 69.70-70.60 | $Ca^{2+}$ | Mayewski et al. 1996 | Peak early MIS 4 | EDC $nssCa^{2+}$ | This study |
| 484.30 | 71.95 | 71.60-72.30 | $\delta^{18}O_{ice}$ | Steig et al. 1998 | Peak AIM 19 | EDC $\delta^{18}O_{ice}$ | This study |
| 487.40 | 73.62 | 73.30-74.00 | $\delta^{18}O_{ice}$ | Steig et al. 1998 | Low between AIM 19 and AIM 20 | EDC $\delta^{18}O_{ice}$ | This study |
| 490.80 | 75.75 | 75.00-76.10 | $\delta^{18}O_{ice}$ | Steig et al. 1998 | Peak AIM 20 | EDC $\delta^{18}O_{ice}$ | This study |
| 493.40 | 77.08 | 76.65-77.50 | $\delta^{18}O_{ice}$ | Steig et al. 1998 | Low before AIM 20 | EDC $\delta^{18}O_{ice}$ | This study |
| 502.75 | 83.9 | 83.00-84.90 | $\delta^{18}O_{ice}$ | Steig et al. 1998 | Peak AIM 21 | EDC $\delta^{18}O_{ice}$ | This study |

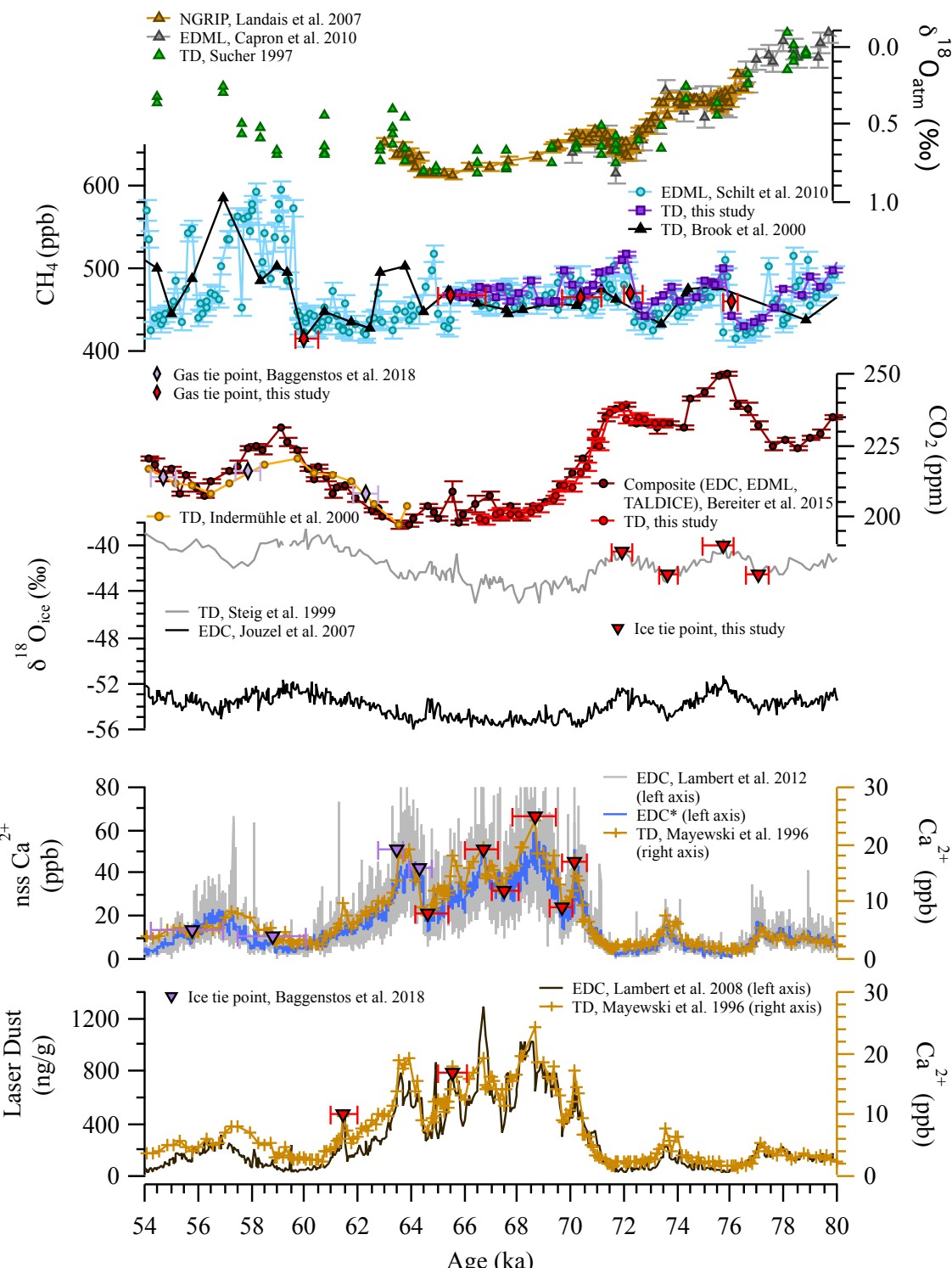

Figure 4 - Measurements of trace gases ($CH_4$ and $CO_2$), stable isotopes (ice and $O_2$), and $Ca^{2+}$ from the Taylor Dome ice core on new gas age and ice age scales. All ice core data are synchronized to AICC 2012. NGRIP = North Greenland Ice Coring Project, TD = Taylor Dome, EDML = EPICA Dronning Maud Land, EDC = EPICA Dome C, TALDICE = Talos Dome. * denotes smoothing with 100 point LOESS algorithm.

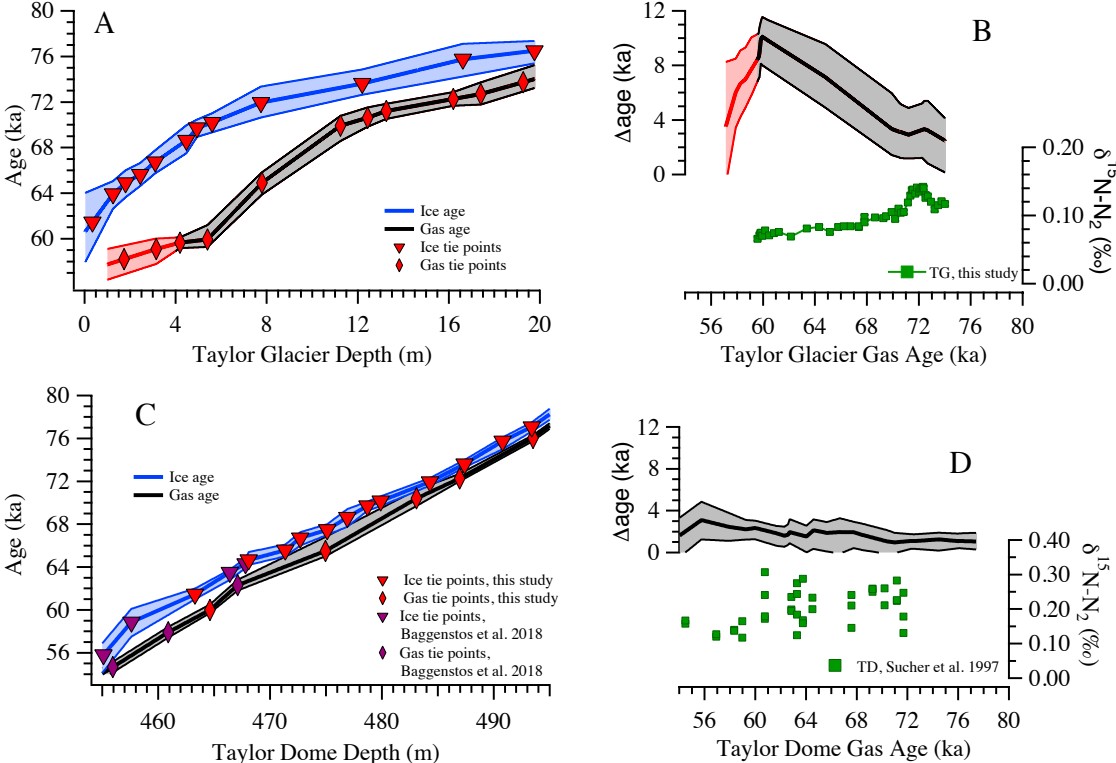

Figure 5 – (A) New Taylor Glacier MIS 5/4 gas and ice age models, and (B) Taylor Glacier Δage and $\delta^{15}$N-$N_2$. Where age data and Δage are plotted in red denote that gas data are from the top 4 m where contamination from surface cracks is possible. (C) Revised Taylor Dome gas and ice age models, and (D) Taylor Dome Δage and $\delta^{15}$N-$N_2$. Δage data are plotted on the gas age scale.

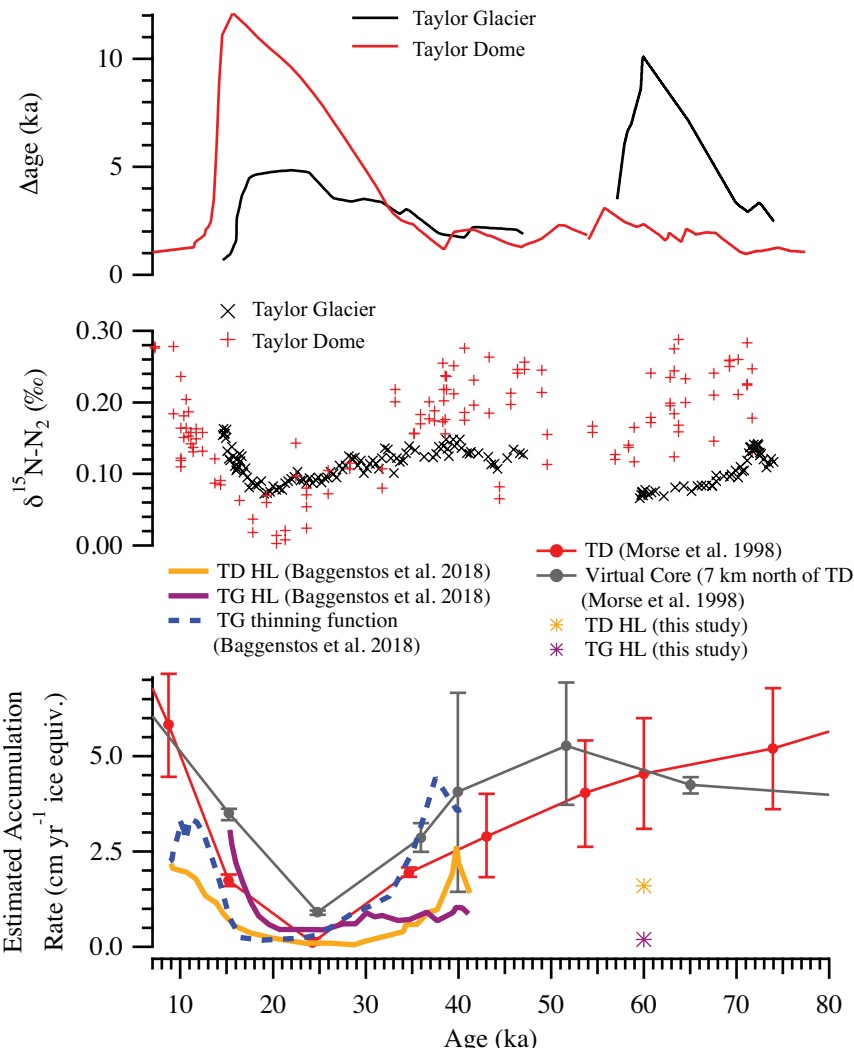

Figure 6 – Δage, δ[15]N-N₂, and estimated accumulation rate for Taylor Glacier and Taylor Dome from 75-7 ka. Δage and δ[15]N-N₂ data between 55-7 ka are from Baggenstos et al. (2018) and 80-55 ka are from this study, except all Taylor Dome δ[15]N-N₂ are from Sucher (1997). Δage data are plotted on the gas age scale. TD = Taylor Dome, TG = Taylor Glacier, HL = Herron and Langway (1980).