# Peer review of "SPATIAL PATTERN OF ACCUMULATION AT TAYLOR DOME DURING MARINE ISOTOPE STAGE 4: STRATIGRAPHIC CONSTRAINTS FROM TAYLOR GLACIER"

_Climate of the Past, 2018_

## Referee Comment (RC1) · Anonymous Referee #1 · 5 Jun 2018

The article "Spatial patterns of accumulation at Taylor Dome during the last glacial inception: stratigraphic constraints from Taylor Glacier" presents new data from Taylor Glacier obtained from blue ice covering the last glacial inception (74-65 ka). This period was primarily thought not to be recorded within the Taylor Glacier area. The comparison of data obtained on site and later in the lab permits to give an idea of the analytical uncertainty and robustness of the Taylor Glacier records of CH4 concentration and particle count. Using these records, they construct chronologies for the Taylor Glacier ice core through alignment with the EDML CH4 and NGRIP d18Oatm records for the gas phase, and the EDC dust record for the ice phase, all on the AICC2012 chronology. They moreover revise the chronology of Taylor Dome ice core using the

same method. Based on their new chronologies they calculate the △age, the age difference between ice and gas at the same depth, for both Taylor Glacier and Taylor Dome ice cores. While the △age remain nearly constant at Taylor Dome during the last glacial inception, at Taylor Glacier the △age progressively increased during MIS 4. The authors interpret the increasing △age gradient through MIS4 as variations in the snow accumulation rates between Taylor Dome and the supposed accumulation area of Taylor Glacier.

This paper present interesting new data obtained from blue ice of Taylor Glacier. I appreciate the efforts made to present the chronology construction, however the chosen figures do not permit to assess the robustness of the method. The choice of tie-point is subjective and when looking at your figures one could argue your choices, which weaken your article. Even if the authors tried to quantify some uncertainties, they did not finalise the uncertainty propagation for the final chronology, limiting the reader in the evaluation of the validity of their work. However, giving estimation of the minimum and maximum values of △age variations for both Taylor glacier and Taylor Dome ice cores is a good idea. I am not convinced about the authors interpretation of evolving gradient in △age between the two sites solely in terms of accumulation changes between Taylor Dome and the supposed accumulation area of Taylor Glacier. The authors need to give more proof for their preferred interpretation and need to justify why they completely disregard variations of thinning between the sites. This article is well within the scope of Climate of the Past and will be of value for the paleoclimate community. I suggest that this article should be accepted for publication after major revisions. You will find my general and specific/technical comments in the supplement.

Please also note the supplement to this comment:
https://www.clim-past-discuss.net/cp-2018-53/cp-2018-53-RC1-supplement.pdf

**Supplement:**

**General comments :**

I would strongly advise to reorganise the paper in separated sections for more clarity. The way it is now, you continuously go back and forth between the sites and methods.
I would suggest the following organization:
Introduction, Field sites and analytical methods (–> presentation of your sites and the measured data in the field and in the lab + analytical uncertainties), Age models (-> choice of tie-points and chronological uncertainty propagation for both TG and TD), Results -> Δage and Discussion. Your manuscript would gain in clarity and would guide the reader toward your results and interpretations.
You should avoid the listing of sites and data in the text and instead propose tables summarizing the data/sites information you need. This is particularly true for the blue ice sites you cite in the text and the different measurements performed on your cores.

**Specific comments & Technical corrections :**

ABSTRACT:
- line 24: "Taylor Glacier (Antarctica)"
- line 27: "low SNOW accumulation WITHIN the Taylor…"
- line 31: replace "Taylor Dome" (already used in the sentence) by "this area"

INTRODUCTION:
Page 1:
-line 36: missing references for past atmospheric composition and a list of trace gases
-lines 40-41: This statement is not true, close to bedrock folding can happen, disrupting the order of ice/gas layers, as seen for the bottom part of NEEM ice core in Greenland for example.
-line 41 "precise distance-age"-> from which reference is the distance measured?

Page 2:
-line 12: remove "with fast access to age information"
-line 13: as precise as what? The previous method?. Replace "have" by "present"
-paragraph 3: it would be easier for the reader if you summarize all in a table (site, location, period covered, references) and refer to it in the main text. Such a listing is difficult to follow with too many commas.
-line 29: replace "expands" by "extends" and replace "by developing ice and gas chronologies spanning" by "back to"
-Line 31-32: remove "the across-flow transect"
-Line 34-36: "paleoarchive FROM TAYLOR GLACIER, where it was previously thought to be absent". Remove "larger context of". Replace "into" by "within". Replace "at Taylor Dome" by "of this region"

FIELD SITE AND ANALYTICAL METHODS:
Page 3:
-line 5: if you are not citing an acronym, ice sheet is written without capital letters
-line 6: "northERN"
-line 7: "ice EQUIVALENT accumulation"

-line 15: "80 km LONG ablation zone", and you already said it in the previous paragraph

-line 16: need rewording. I suggest the following: "Water stable isotopes obtained from an along-flow transect just below the equilibrium line"… "revealed uncontinuous ice covering the last glacial period"

-line 20: "revealed continuous records of ice from the Holocene to the last ice age, with ice of the last interglacial and older found…" references for this statement?

-line 22: "the most COMMONLY USED archive" instead of utilized

-line 27: Reference for the previous ice core study. Where was taken this new ice core compared to the previous study? Need more precision. What was the sampling problem with the previous record?

-lines 30-33: need a reference

-line 36: replace "in" by "of"…"CH4 variations similar to those ASSOCIATED WITH  DO19" or "corresponding to"

-line 37: "CH4 CONCENTRATION increase"

Page 4:

-line 3: replace "work" by "analysis", replace "spanning" by "section"

-line 4: need rewording, proposition: "…CH4 and CO2 concentrations, which confirmed the MIS 4/5 transition record in the gas phase"

-line 6: spanning not properly used

-lines 7-11: it would help to make a table for all the analyses performed on the different cores, with specification of the proxy measures, where, the time coverage of samples (or portion of core) and the method used for measurements, analytical uncertainty…

-line 24: "resulted in a good agreement of our measurements with other…"

Page 5:

-line 19: "… on archived Taylor Dome ICE CORE samples…"

-line 22: "(~10g OF ICE, …)"

RESULTS AND DISCUSSION

3.1 AGE MODEL

Page 5:

-line 28: "synchronized to" not correct, more likely "presented on"

-line 29-33: need rewording, not clear

-line 33: need more precision, here a proposition: "We constructed our gas age scale based on the alignment of our CH4 and d18Oatm data with the EDML CH4 and NGRIP d18Oatm records on the AICC2012 chronology."

Page 6:

-line 10: 'CO2 CONCENTRATION decrease", again later

-line 11: remove "and"

-line 13: value of the offset?

-line 15: need rewording, a proposition: "…younger ages. Therefore, we refrain from further align the CO2 rises together for better consistency."

-line 17: why not use the d18Oatm of Vostok or TALDICE instead of NGRIP? You would have a complete record over your period of interest on AICC2012, but potentially with a lower resolution.

-line 23: replace "has" by presents"

-paragraph 3: I am not very much convinced by value matching for dating. We do not really understand the usefulness of the -380 core data until the idea of similar firn conditions. This and the following argument are important for your interpretation later. This paragraph needs rewording.

Page 7:
-paragraph 2: not useful, could be removed.

3.2 ANALYTICAL AND AGE MODEL UNCERTAINTIES
Page 7:
-line 18: "is likely"
-lines 19-22: not clear. You say that you consider the 2015-2016 data as uncontaminated, but as the same record differ from the lab, in the end you do not interpret the data... but you did later in the text...
Moreover, you did not discuss the reasons that could explain why the records are so different. I would possibly keep the tuning, but associate it with a much larger uncertainty than the other points due to the mismatch with the lab data. Then only use the CH4 data in grey area for dating purposes and no more.
The discussion about the analytical uncertainty should be following the presentation of the analytical methods.

-paragraph 4: I am not convinced about your argument for the confirmation of data. From the looks of the data presented on Figure 2, I would say that your choice of markers is not convincing, I would have chosen differently... From your Figure 3, I understand that your choices were made in order to align together the records you cite as confirming your alignment (e.g. nssCa). I would recommend to change the way you presented your figure 2 to make the reader see by himself why you choose these tie-points and not others.
You should focus more on this aspect, which is the base of your discussion later, it would strengthen your work. Not necessarily in the main text, it could be an appendix.

Page 8:
-lines 3-4: "20 cm = 300 years", based on what? Which chronology?
Lines 5-8: This is not a proper argument. If you say that both CH4 data from TG and EDML are similarly smoothed in the firn column, you are implying that they have similar firn conditions (i.e. accumulation rates, firn depth...). Is it the case?
-lines8-9: Analytical noise... why is that? What is the measurement uncertainty of your method?
-lines 9-12: Please, when using a chronology as reference, make sure of the uncertainty values you cite... What you wrote is not correct. The AICC2012 chronology uncertainty over your period of interest (i.e. ~65-74 ka) at EDML is ranging between 1500 years and 1400 years (cf. supplementary material of Veres et al., 2013). The values you have indicated correspond to the uncertainty of the ice and gas chronology at the orbital scale, prior to the last interglacial.

-Following all this discussion of uncertainties, what are the uncertainties associated with your ice and gas chronologies for TG? You never gave a value and I do not see them on your figures. The same for your revised TD chronology.

3.3 ΔAGE AND COMPARISON TO TAYLOR DOME
Page 8:
-line 7: Temperature and accumulation are not the only factors influencing Δage. All factors acting on the firnification process do as they impact on the firn depth variability. What about insolation of wind stress affecting the snow metamorphism into ice?
-line 19: remove "on the order of hundreds of years", it is given by the lower limits just before. Change "smaller" in "smallest"
-line 21: replace "at" by "for"
-lines 26-27: ok for the two sources of uncertainties, but you forgot to take into account the absolute uncertainty of the ice and gas chronologies. You have ~1500 years uncertainty from the AICC2012 age scale, consequently the uncertainty of your new chronology should be around ~1600 years for the gas age, and ~1530 years for the ice age (I took one random range from your choice of tie-points).
Then your maximum and minimum Δage should be obtained from the (ice age - 1sigma)-(gas age + 1sigma) and (ice age + 1sigma)-(gas age - 1sigma). You should give an approximate value of the Δage uncertainty for the reader to have an idea of the significance of your Δage values later.
-line 30: "10 ka"+/- ??? uncertainty needed.
-line 33: then why is it so different? Replace "high" by "large"
-line 34: now you talk of the influence of wind, but not before…

Page 9:
-paragraph 1: I do not think that the last sentence is necessary, you should delete it.
-paragraph 2: You should gather together in one section the chronology construction for your two sites, with the proper calculation of their respective uncertainties.
-line 13: "in the same manner AS described"
-lines 17-19: You should then directly give a 0 value. Note then the uncertainty associated to the Δage is then not guassian..
-line 21: Δage of 2.5 ka, but p8 line 20 you cited an extrema value of 12 ka with reference to Baggentos et al., in review… why are the values so different?
-lines 22-25: I disagree with this statement. It comes too soon. For TG, not located on a dome, ice thinning and ice flow are very important factors that could affect the depth-age relationship. For TG you cannot interpret directly your variations on Δage in terms of accumulation. To distinguish between the major influences of thinning and accumulation, you need an ice flow model. If your ice flow model indicate that there are no significant thinning variations, then and only then you can interpret it in terms of accumulation. Moreover, you give absolutely no justification for your favour toward accumulation changes, and you do not explain why you disregarded the thinning influence.
-last paragraph: you should give the modern values of accumulation measured at these two sites. It would give an idea of how much your prior assumption of all differences are due to accumulation changes is valid for modern times.

Page 10:

-lines 14-18: give values for the LGM reconstructed accumulation at both TD and the virtual sites. This gradient is reverse from yours. Why do you use it then? The useful result from this study to you is only "the opposite accumulation gradient (decreasing from south to north) for ice older than 60ka".
-lines 18-26: bring nothing more, just show support for the LGM gradient that is different from yours. I would advise to remove these sentences.
-last paragraph: remove the first two sentences, you are only rewording your results.

Page 11:
-line 4: need a reference for this statement.
-paragraph 2: the MIS 4 gradient is similar to modern conditions. Are modern conditions in agreement with your proposed hypothesis?

FIGURES & TABLES:
-Figure 1: I would advise to change the organization: a-Antarctica map, b-landsat imagery, c-simplified map of TG.

-Figure 2: The way the data are presented now, one can strongly argue your chosen tuning points. The scales are two small to see the consistency between the associated variability. I am not at all convinced about your tie-point between the d18Oice of EDC and TG, records present different variability. I would advise to remove from the legend the last two sentences.

-Tables 1&2: You should add some indications on your figure 2, on the reference records, to directly make the link between the tables and your chosen points (e.g. DO19…). In Table 2 legend, remove the sentence "Ice phase…"

Figure 3: I would say that there is absolutely no point in plotting together records that were tuned together, or if you really want to, it should be in an appendix. You already use some other untunned records to validate your chronologies. I would leave here only 1 gas, 1 ice records, and then the (b) part of the figure. You should extend the lines for the identification of MIS limits to the bottom of the figure for more clarity. In the legend your last sentence is not necessary, you could delete it.

Figure 4: Same comments as for Figure 3. Your should keep consistent the colours of curves from one figure to another. Why didn't you remove the three points in questions and simply state it in the measurement section?

---

## Referee Comment (RC2) · Anonymous Referee #2 · 8 Jun 2018

1- SUMMARY AND GENERAL COMMENTS:

The study by J. Menking and collaborators presents three new ice cores from the Taylor Glacier Blue ice area that they combine to provide the first "composite" ice core record from this location that covers the transition between Marine isotopic Stage (MIS) 5 and MIS 4 ($\sim$74 to 65 ka). The chronology for the air trapped in the ice is defined based on the analysis of the global atmospheric tracers CH4 and atmospheric $\delta$18O of O2 ($\delta$18Oatm) and their synchronisation with well-dated CH4 and $\delta$18Oatm records from other Antarctic ice cores. The ice age scale is defined mostly based on the ice dust content synchronisation, again with other well-dated Antarctic dust profiles. From these

two ice and gas age scales, they infer the evolution of the age difference between ice and gas at the same depth – the so-called Δage – though this MIS5-MIS4 climatic transition. Substantial Δage changes are observed through time over this time interval i.e. with values from ∼2000-3000 years at ∼74 ka and approaching ∼10 000 years at ∼60 ka. The authors also provide a new evaluation of the Δage evolution throughout the same period in the Taylor Dome ice core (located south of the glacier) which suggests no significant Δage changes for this site. The authors attribute these contrasting Δage evolutions between the two sites to a steep accumulation gradient across Taylor Dome that intensified across the transition from MIS 5 to MIS 4.

This paper presents a study that will be of great interest for the ice core community and to the extended paleoclimate community. It is thus well within the scope of Climate of the Past. Overall the manuscript is well-written and presents substantial new material and interesting interpretation of the results. However several aspects of the paper need improvements and clarifications and thus I believe that major revisions are needed before it can be considered for publication.

My first major comment is related to the fact that the authors interpret the differences in the Δage evolutions between the Taylor Glacier area and the Taylor Dome ice core site almost exclusively in term of a change in the accumulation gradient between the two areas. While this could be an acceptable interpretation, they absolutely need to build a much stronger case regarding why this is their favoured one (e.g. versus ice thinning) and thus provide a much more elaborated discussion of their new results. But also, they should discuss the other possible controlling factors, in particular, those are commonly identified as impacting the firnification processes e.g. the role of surface temperature vs accumulation rate vs ice impurity content have already been discussed over the past few years (e.g. Bréant et al. 2017, Capron et al. 2013; Hörhold et al. 2012). I believe that a summary of the current knowledge (and knowledge gaps) regarding the climate and environmental factors that impact changes in Δage would be useful. In particular, it would be of added value to further mention firn densification models that provide an

alternative method to estimate ∆age. At the moment the authors only acknowledge the Herron and Langway model (1980) although several other models building on this original work have been developed in the more recent years (e.g. Goujon et al. 2003 and more recent development in Bréant et al. 2017, dynamical version of Herron and Langway (1980) used in e.g. Buizert et al. 2015).

My second major comment is related to the form of the paper. First I believe that some reorganisations of some sections are necessary and I detail this in the next section. Second, I think that the Figures 2, 3 and 4 need to be revised so that the readers are able to better visualised the different records that are being presented but also so they better support the results and the proposed interpretation. More details are provided in the next section of the review.

Additional comments are also provided in the following and I would strongly advice the authors to consider them when preparing a revised version of their manuscript.

2- SPECIFIC COMMENTS:

- Section 2 (Field site and analytical methods) is not always easy to follow, in particular regarding which type of measurements has been performed on which core and where (on site or in labs back in the USA). I would suggest the authors to propose a summary table in the revised manuscript that detail clearly this information.

- The authors propose to treat the three ice cores covering the MIS5-4 transition as a single ice core record (unified depth and age scales). While I agree with them that it is justified, I believe that they should provide additional details on how they line up the different records together (and possibly provide a specific figure?) and discuss the attached uncertainties that arise from proceeding as such on the resulting "composite" record.

- Section 3.1 is hard to follow, the authors should consider restructuring it such as 1) they present how the ice age scale has been defined and then 2) as the gas age scale

has been defined. Regarding the definition of the tie points based on the alignment of the dust record, I find that some of them are quite ambiguous considering the number of spikes present in the TG records. For instance why would they assign the tie point at 73.6 ka to the spike at 12 m rather than the spike at 9 m? I believe that the authors have a good reason for doing so, however, it should be spelt out more explicitly. It is necessary that the figure is much enlarged to allow a detailed inspection of the records.

- I do not think that the analytical uncertainties should be discussed after the determination of the age model. The authors should consider adding a brief description of each dataset after the analytical method descriptions and there, add details regarding their specificity and limitations.

- It is a little strange that the presentation of the new measurements on the Taylor Dome ice core and the definition of the new age scale and for Taylor Dome are currently presented as part of the discussion. Why not instead presenting the new age model of Taylor Dome as an additional sub-section in the age model section that is currently only dedicated to the dating of the Taylor Glacier ice? And similarly for the new measurements, they should be also included in the analytical description section and information should be also added in the table I propose to add in the revised manuscript. Also, I think it would be very useful that more background information is provided regarding the Taylor Dome site, in particular regarding the previous age scales available for this time interval.

3- FIGURES

- I appreciate the effort of the authors to show how they defined the different tie points to link between the Taylor Glacier records on a depth scale the dated reference records. However, it should be bigger to allow a closer inspection of the different records and where the tie points have been chosen.

- Figures 3 and 4 should appear much bigger. Also, to facilitate the comparison of $\Delta$age evolutions between Taylor Glacier and Taylor Dome, I suggest to remove the panels b

from each figure and combine these panels b into a single and additional figure. They can be presented in parallel, making sure that the scale used for the $\Delta$age evolution is the same for both sites.

4- STYLISTIC, TYPOGRAPHICAL COMMENTS AND MINOR COMMENTS

P2, L16: You should also mention the work that has been done in the Patriot Hills blue ice area e.g. Fogwill et al. (Scientific Reports 2017).

P2, L34: I find the expression "MIS 4 paleoarchive" to be an awkward formulation; I would suggest to reformulate the sentence e.g. "(2) the description of a new climatic record from Taylor Glacier across MIS 4".

P4, L1: "second exploratory core": this is a bit confusion to say "secondary" since the PICO core was also referred to as a "secondary exploratory core". It should be rephrased e.g. "During the same 2014-2015, another exploratory core was obtained directly . . ..".

P4, L5: Again the numbering of the core is confusing (as in total, as far as I understand, four cores were drilled with only the last three having MIS5/4 transition ice). Hence it would be could to reformulate such as e.g. "In the 2015-2016, an additional core was drilled. . .".

P5, L26: The authors should be more specific in the title of the section e.g. "Determination of the ice age and gas age scales".

P6, L4: "minimal" please be more quantitative here and give a quantitative range at least.

P8, L11: Although you refer to the tables, the authors should also provide at least a quantitative range regarding the relative age uncertainties.

REFERENCES:

Capron et al. 2013. Glacial–interglacial dynamics of Antarctic firn columns: comparison between simulations and ice core. Climate of the Past, 9, 983–999.

Breant et al. 2017. Modelling firn thickness evolution during the last deglaciation: constraints on sensitivity to temperature and impurities. Climate of the Past, 833–853, 2017.

Buizert et al. 2015. The WAIS Divide deep ice core WD2014 chronology – Part 1: Methane synchronization (68–31 ka BP) and the gas age–ice age difference. Climate of the Past, 11, 153–173.

Goujon C. et al. 2003. Modeling the densification of polar firn including heat diffusion: Application to close-off characteristics and gas isotopic fractionation for Antarctica and Greenland sites, J. Geophys. Res., 108, 4792, doi:10.1029/2002JD003319, 2003.

Herron and Langway 1980. Firn densification-an empirical model, Journal of Glaciology, 25, 373-385, 1980.

Hörhold et al. 2012. On the impact of impurities on the densification of polar firn. EPSL, 325-326, 93-99.

Fogwill et al. 2017. Antarctic ice sheet discharge driven by atmosphere-ocean feedbacks at the Last Glacial Termination. Scientific Reports, DOI: 10.1038/srep39979.

---

## Referee Comment (RC3) · Anonymous Referee #3 · 27 Jun 2018

**General comments**

The manuscript presents the initial multi-tracer dating of recent large size ice cores from Taylor Glacier (TG), covering a period of about 25 ka around the MIS 4/5 transition, as well as new data aiming at improving the gas chronology of the Taylor Dome (TD) ice core during the same period. Such characterization of a blue ice field providing large amounts of ancient ice is certainly of interest for the paleoclimate community and well within the scope of Climate of the Past.

The results are discussed in terms of age difference between the gas and ice phases ($\Delta$age) and related varying accumulation rates. This interpretation involves some as-

sumptions and simplifications that are not enough described in my view. For example, a number of age synchronization tie points appear ambiguous to me and the remaining discrepancies between records are not sufficiently commented. The inferred very low accumulations are likely to imply erosion periods, and the impacts of the ice-flow (thinning, hiatuses, possible folding etc.) should be better considered. Even if firn modelling with somewhat empirical models well outside the calibration range of their parameters is not compulsory, the physical processes controlling $\Delta$age and $\delta^{15}$N fractionation should be better described.

Overall I think that major revisions are needed in order to better discuss the approximations made (e.g. ignored firn and ice physics), describe the consequences of alternative assumptions on ambiguous chronological tie points for multi-species consistency, age scales and $\Delta$age. I think that the paper should be more focused on an in depth discussion of the ice cores dating and dating issues, and less focused on somewhat spectacular but uncertain conclusions on $\Delta$age and accumulation. A number of suggestions are provided below.

**Specific comments**

p2 l34-35 and p3 l26-28: Missing MIS 4 and MIS 4/5 transition in previous TG records. The authors should provide references and introduce more the possibility of having different hiatuses in different TG ice cores. The ice flow in the area should be better illustrated, for example Figure 1 (a) could be further zoomed on the drill sites and some flow line directions could be provided.

p2 l37: a reference should be provided for the previous TD chronology

p3 l14-16: a reference should be provided for these site characteristics

p3 l25-28: a reference should be provided for the ice flow structure of the "main transect"

p3 l30-31: the exact location of the "-380m" drill site (coordinates) should be provided.

More site information could be provided (e.g. altitude, mean annual and summer temperatures etc.)

p5 l8-11 and p8 l1-4: the depth offsets, uncertainties and unification method between the different "TG 5/4" cores should be better described.

p5 l17: "The interpretations that follow do not depend on data taken from 0-4 m", and similar statement p7 l22. In Figure 2, the 3 TG $CH_4$ data series are not consistent above 5m depth, and in Figure 3 the $CO_2$ consistency with the composite in the upper part of the TG record mostly rely on the 2 upper points. What would be the consequence of matching the TG $CO_2$ record below 4 or 5 m depth to the composite $CO_2$ record instead of using the $CH_4$ record which is nearly flat between ∼4.5 and 7 m depth for multi-species consistency and $\Delta$age? In Table 2, two $CH_4$ tie points and half of the ice phase tie points are located well above 4 m depth.

p5 l27-30: In Figure 2, the TG $CH_4$ records look a lot smoother than the EDML record. The dissimilarity of the two signals limits the possibilities of unambiguously synchronizing them. This could be due to different processes such as analytical smoothing (Stowasser et al., 2012), longer gas trapping duration in firn at very low accumulation rates (Spahni et al., 2003; Köhler et al., 2011; Fourteau et al., 2017), gas diffusion through ice (Bereiter et al., 2014 and references therein). This should be discussed, possibly smoothing the EDML record to try to simulate the TG record, comparing with the lower accumulation EDC record etc.

p5 l34-35: I did not understand why the $\delta^{18}O_{atm}$ record is tied to NGRIP only: a North Hemisphere discontinuous record covering only parts of the studied period. Could other data also be used? (e.g. Petit et al., 1999; Kawamura et al., 2007; Buiron et al., 2011)

p5 l38: some tie points look ambiguous to me and the tie points assignment should be further discussed. For example, the EDC and TG $\delta^{18}O_{ice}$ records look quite different in Figure 2, thus the $\delta^{18}O_{ice}$ tie point does not look robust to me. On the dust plot in

Figure 2, I do not understand why the small EDC peak at 75.75 ka was tied to the TG particles peak at ~12m rather than the one at ~9m depth.

p6 l9-27: Due to the dissimilarities between the records in Figure 2, I believe that it is impossible to unambiguously assign the tie points. Thus I doubt that the choices were made without taking into account the constraints discussed in this section. An overall discussion of the constraints, what led to the current best guess dating and how other assumptions could be (or not) discarded would be most useful.

p6 l31-32 and Figure 3: I do not understand how the $CH_4$ record from the "-380m" core could be unambiguously tied to AICC2012. On the other hand the $CO_2$ records seem easier to match and matched. The overall dating constraints should be better described.

p6 l31 - p7 l14: I did not understand this discussion of the differences between the TG records. The dating of the "-380m" core is presented in one line and the $CO_2$ mismatch with "TG 5/4" not discussed, nor the $\delta^{18}O_{atm}$ mismatch with NGRIP at ~66 ka. The lack of information on flow line directions make the direct comparison between TG records difficult to understand, and few references are provided. I suggest to focus more this section on gas scales consistency between the "-380m" and "TG 5/4" cores, and how the $CO_2$ mismatch between the two TG cores in the 60-64ka age range could be explained. Is the ice phase of the "-380m" ice core also dated? Are large $\triangle$age values also inferred?

p7 l8-9: As this paragraph comes just after the section comparing the "-380m" and aggregated "TG 5/4" cores, readers may wonder which one is the new ice core.

p7 l8-13 and p9 l25-31: providing and discussing plots of annual layer thicknesses (based on depth - ice age, depth - gas age relationships at TG and TD) would help understanding the interpretations related to accumulation and thinning variations.

p7 l24 - p8 l12: This discussion of uncertainties should appear earlier in the article and

be more detailed (see also above comments on p5 l27-30, p5 l38, p6 l9-27).

p8 l1-4: this is not consistent with p5 l11. Due to the strongly varying depth-age gradients on Figure 3 (b), the overall largest age bias related to depth offset/uncertainty should be mentioned.

p8 l5-8: the smoothing due to gas trapping duration most likely dominates the diffusive smoothing in the open pores of the firn. It is accumulation rate dependent (e.g. Spahni et al., 2003; Köhler et al., 2011; Fourteau et al., 2017) and thus likely different at EDML and TG. In Figure 2, the TG $CH_4$ record looks much smoother than the EDML record. It would thus be interesting to discuss the gas trapping duration consistent with the firn sinking speed due to the estimated accumulation rates (time needed by the firn to sink by a few meters).

p8 l16-18 and l35-36: a much more in depth presentation of firn processes influencing $\Delta$age, $\Delta$depth and the physics of $\delta^{15}$N should be provided. The consistency between a very large $\Delta$age and a very shallow firn ($\delta^{15}$N indication) should be commented.

p8 l24-30: the example of the successive datings of the Taylor Dome ice core, well discussed in Baggenstos et al. (2018) could be used as a base for a more realistic uncertainty discussion.

p8 l35 - p9 l4: the fact that the physics of $\delta^{15}$N (thermal and convection effects) is much more complicated than a pure gravitational effect can't be ignored (e.g. Severinghaus et al., 2001; Severinghaus et al., 2010). The very low $\delta^{15}$N values measured in TG ice suggest that either the firn is very thin (an estimate should be provided) or nongravitational effects are important.

p9 l6-19: the new Taylor Dome age scales presentation repeats methodological information already provided for TG cores but does not discuss the remaining inconsistencies between records and ambiguous tie points. A more in depth discussion of the Taylor Dome age scales should be provided.

[Figure]

p9 l21-31: this section is unclear to me. If TG and TD ice cores have strongly different Δage in the study period (assuming that the tie points sufficiently constrain the age difference between the gases and ice in a single ice sample), TD can't be the origin site of TG ice even considering differential thinning.

p9 l2-4 and p9 l33 - p10 l26: no accumulation values were derived from the Taylor Glacier record and the discussion is focused on different time periods (present and LGM), thus it could be shortened.

**Technical corrections**

p8 l18-20: smaller Δage values were obtained at very high accumulation rate sites such as DE08-2 (40 years, Etheridge et al., 1996)

p11 l24-25: twice "spanning the MIS 5/4 transition"

p13 l15: Baggenstos, 2015 (PhD) a web link could be provided.

p13 l23 and in article text: Update reference to Baggenstos et al. (2018), now available as a preprint.

p15 l49: suppress QUATERNARY

p16 l56: uppercase/lowercase issue

Figure 2, dust panel: some grey lines are not consistent with the tie points in Table 2 (chronology inversions in some grey lines)

Figure 3: the top part of the TG particles count record, including the tie point at 0.31 m depth, is not shown.

**References not cited in the manuscript**

Bereiter, B., Fischer, H., Schwander, J., and Stocker, T. F.: Diffusive equilibration of $N_2$, $O_2$ and $CO_2$ mixing ratios in a 1.5-million-years-old ice core, The Cryosphere, 8, 245-256, https://doi.org/10.5194/tc-8-245-2014, 2014.

Buiron, D., Chappellaz, J., Stenni, B., Frezzotti, M., Baumgartner, M., Capron, E., Landais, A., Lemieux-Dudon, B., Masson-Delmotte, V., Montagnat, M., Parrenin, F., and Schilt, A.: TALDICE-1 age scale of the Talos Dome deep ice core, East Antarctica, Clim. Past, 7, 1-16, https://doi.org/10.5194/cp-7-1-2011, 2011.

Etheridge, D. M., L. P. Steele, R. L. Langenfelds, R. J. Francey, J. M. Barnola, and V. I. Morgan, Natural and anthropogenic changes in atmospheric $CO_2$ over the last 1000 years from air in Antarctic ice and firn, J. Geophys. Res., 101, 4115–4128, 1996.

Fourteau, K., Faïn, X., Martinerie, P., Landais, A., Ekaykin, A. A., Lipenkov, V. Ya., and Chappellaz, J.: Analytical constraints on layered gas trapping and smoothing of atmospheric variability in ice under low-accumulation conditions, Clim. Past, 13, 1815-1830, https://doi.org/10.5194/cp-13-1815-2017, 2017.

Kawamura, K., Parrenin, F., Lisiecki, L., Uemura, R., Vimeux, F., Severinghaus, J.P., Hutterli, M.A., Nakazawa, T., Aoki, S., Jouzel, J., Raymo, M.E., Matsumoto, K., Nakata, H., Motoyama, H., Fujita, S., Goto-Azuma, K., Fujii, Y., Watanabe, O., 2007. Northern Hemisphere forcing of climatic cycles in Antarctica over the past 360,000 years. Nature 448, 912-916. https://doi.org/10.1038/nature06015.

Köhler, P., Knorr, G., Buiron, D., Lourantou, A., and Chappellaz, J.: Abrupt rise in atmospheric $CO_2$ at the onset of the Bølling/Allerød: in-situ ice core data versus true atmospheric signal, Clim. Past, 7, 473–486, https://doi.org/10.5194/cp-7-473-2011, 2011.

Petit, J.-R., Jouzel, J., Raynaud, D., Barkov, N.I., Barnola, J.-M., Basile, I., Bender, M.L., Chappellaz, J., Davis, M., Delaygue, G., Delmotte, M., Kotlyakov, V.M., Lorius, C., Pepin, L., Ritz, C., Saltzman, E., Stievenard, M., 1999. Climate and atmospheric history of the past 420,000 years from the Vostok ice core, Antarctica. Nature 399, 429-436.

Severinghaus, J.P., Grachev, A., Battle, M., 2001. Thermal fractionation of air in

polar firn by seasonal temperature gradients. Geochem. Geophys, Geosys. 2 2000GC000146.

Severinghaus, J. P., Albert, M. R., Courville, Z. R., Fahnestock, M. A., Kawamura, K., Montzka, S. A., Mühle, J., Scambos, T. A., Shields, E., Shuman, C. A., Suwa, M., Tans, P., and Weiss, R. F.: Deep air convection in the firn at a zero-accumulation site, central Antarctica, Earth Planet. Sci. Lett., 293, 359–367, 2010.

Spahni, R., Schwander, J., Flückiger, J., Stauffer, B., Chappellaz, J., and Raynaud, D.: The attenuation of fast atmospheric $CH_4$ variations recorded in polar ice cores, Geophys. Res. Lett., 30, 25-1–25-4, https://doi.org/10.1029/2003gl017093, 2003.

Stowasser, C., Buizert, C., Gkinis, V., Chappellaz, J., Schüpbach, S., Bigler, M., Faïn, X., Sperlich, P., Baumgartner, M., Schilt, A., and Blunier, T.: Continuous measurements of methane mixing ratios from ice cores, Atmos. Meas. Tech., 5, 999–1013, doi:10.5194/amt-5-999-2012, 2012.

---

## Referee Comment (RC4) · Anonymous Referee #4 · 12 Jul 2018

Review of the manuscript "SPATIAL PATTERN OF ACCUMULATION AT TAYLOR DOME DURING THE LAST GLACIAL INCEPTION: STRATIGRAPHIC CONSTRAINTS FROM TAYLOR GLACIER" by J. Menking et al.

General comments: The authors collected and analyzed a set of new ice cores from the Taylor Glacier blue ice area covering the MIS 5 to 4 transition and whole MIS4, and present a suite of data (d18Oice, dust, Ca ion, d18Oatm, CH4, CO2, d15N2). Through age synchronization of gas and ice with other dated ice cores, they find extremely large delta age (ice age - gas age difference), which suggests much reduced accumulation rate at the snow accumulation area for the analyzed core and, by comparing the results

with the Taylor Dome ice core data with their revised chronology, give climatic implications of the accumulation contrasts between the Taylor Glacier accumulation area and Taylor Dome.

Regional reconstructions of glaciological and climatological conditions in the glacial period in Antarctica are important for better understanding of the climate system in the Antarctic and its relation with wider areas, and thus the topic of this manuscript is well suited for the Climate of the Past, and the data presented in general seems to be of high quality. First I would like to respect and congratulate the authors for finding the ice from entire MIS4 after the years of fieldworks and high-quality investigations.

I review it mainly in terms of whether the ages, delta age and resulting accumulation rate reduction are reasonably estimated, because they are the basis for the climatic interpretation and conclusions, and also because they have the highest scientific value in this study in my opinion. In doing so I find that the manuscript needs a major revision to make much stronger cases for the extremely increased delta age and reduced accumulation rate (including its timing) at the Taylor Glacier accumulation area, which in turn are based on age synchronization and interpretation of the resulting delta age. In particular, I find it difficult to evaluate the robustness of their choice of the age tie points for some cases from the given text and figures/tables. There are also several tie points which I did not understand how they could match with the existing ice core records. The authors made poor use of the data (especially d15N) for the discussion of the accumulation rate. While I agree with the authors that the delta age increased and accumulation rate probably decreased in MIS 4, it should be based on much more rigorous considerations. Also, some parts of the manuscript needs to be reorganized to better present the field works, ice core samples, measurements and methods, results and discussion. I strongly encourage the authors to improve the article by deeper analyses, interpretation and better presentation of their excellent data.

Specific comments:

Abstract: Add description that there are different ice cores, and that how the delta age was estimated ("Dating the ice and air bubbles" is too short even for the abstract). Similarly, "A revised chronology for the Taylor Dome ice core" needs some more explanation. Also, give numbers and error ranges for delta age and accumulation rate ("very low accumulation" is too vague; later in the text it is stated as virtually zero accumulation rate).

Introduction: P2, L30: "glacial inception" is used differently (it is often used for MIS5e to 5d transition), so perhaps replace it with "major sea level fall" or "major ice sheet growth in the Northern Hemisphere".

P3, L1-2: "the differences in the ice age-gas age difference". Delete one of the "difference".

Field site and analytical methods: The title of the chapter should reflect the fact that it also describes ice cores drilled in different seasons.

P3, L33 and P4, L1: The phrase "a second exploratory core" appears twice for different cores (PICO and BID cores).

P3, L35-37: Please clarify if this measurement was done in the field (brown markers in Fig 2), and when and how did you conduct the whole CH4 measurements. It is unclear to me because you mention D/O 19 but not the larger increase at D/O 17 (D/O 17 is only mentioned earlier for the "-380 m" core). Did you obtain all data before you drilled the BID core? Didn't you take the D/O 17 CH4 transition in the 2014-15 PICO core into account for the preliminary age estimate in the field?

P4, L3-5: Please clarify which data you mean (Fig. 2, green markers). Also, ice sampling for d18Oatm and d15N is not mentioned here (2014-15 BID core) but there are data points in Fig. 2 that says the measurements were done in two years (2016 and 2017). Please give full explanation about the cores, sampling, measurements and periods for all data you present in a better way (not only about this core; using a table

may be a good way).

P5, L6-8: Didn't you take any samples to have overlaps with the previous cores? Did you make the sample cuttings for OSU and SIO in the field?

P5, L9-10: The description of the sampling of the 2015-16 core in the laboratory is better placed after describing the core transportation. Overall, the descriptions of field and lab samplings and analyses are scattered so they should be better organized.

P5, second paragraph: This part is about field measurement methods so it should come earlier before the first presentation of the relevant data. And, for which core this paragraph's description applies (2015-16 core only)?

P5, third paragraph: Please clarify which kinds of measurements were made for the different cores (maybe use a table). The measurement methods should come earlier than the first description of the data, or tell the readers that the methods are described later if you introduce the data first (like in the current manuscript).

P5, L16: The CH4 field data from 4 - 5 m in the 2014-15 core disagree with the CFA data of 2015-16 core by several tens of ppb, which is much more than your precision and should be discussed as well.

Results and discussion: P5, L36-37: The oldest tie point between the TG and EDC using d18Oice seems unacceptable given the different shapes of the isotopic curves of TG, EDC and EDML cores for this and other periods presented in Fig. 2.

P5, L38 - P6, L1: Some of the dust tie points seem unacceptable or maybe you didn't explain the details of the manual matching. You put one at 12 m but why did you choose that particular one and not other peaks? Another one at about 6.5 m is described as low point in dust, but the 2015-16 field data (purple in Fig. 2) actually show a peak there (I guessed that orange plot in Fig. 3 is the same as purple in Fig. 2, showing high values at the tie point). The grey lines for the dust (Fig. 2) are drawn between the EDC and TG purple data (orange in Fig.3), but is this particular one connects EDC

and TG DRI data instead? Why did you choose the point where the two dust records from the same core disagree? Around the one at 70.11 ka, the TG dust peak is offset compared to EDC dust peak (isn't it better not to match the highest point in the peak which have certain width?). Similar examples are at ∼65.6 and 63.9 ka. Overall, the lack of details on the matching force me to suspect that you chose the dust tie points while actually checking the resulting chronology by comparing Ca ion data from the two cores (I see that Ca ion data between TG and smoothed EDC compare much better than between the dust records), meaning that Ca is not just used for the validation of blind test (looking only dust) but effectively involved in the tuning. Otherwise, how could you choose the dust tie point at 12 m?

In fig. 3, dust data look like bar graph (vertical grey and orange bars) but they should actually be line plots. It is hard to evaluate the match in this figure so please improve the plots.

From the text (linear interpolation), I think the TG depth-age plot (Fig. 3b) should be straight lines between the tie points, but they don't look like so. A clear example is at an inflection point in the ice chronology at about 72.3 ka, for which there is no ice tie point. There might be my misunderstanding and if so please give full explanation for the interpolation. Please also plot markers at the tie points on the depth-age curves (Fig. 3b). The depth-age and delta-age lines in the figure are too low in resolution (the lines consist of tiny segments of horizontal and vertical lines, like aliasing in low resolution digital images).

You should reject two youngest CH4 tie points. Cracking and contamination should increase the measured CH4 concentration, so those two tie points are probably put on contamination peaks (note large disagreements between purple line, red line and brown markers). Perhaps you can use the peak at 3 m in brown data (if you take only low values in the two of the CFA data you see the same peak, which is uncertain but this could be a true atmospheric peak concentration).

The match of d18Oatm records look somewhat uncertain especially for the older one. Why did you connect the oldest d18Oatm data point to the beginning of the d18Oatm enrichment in NGRIP data (why not the second oldest data point in TG d18Oatm which is the highest)? I think the measurement precision is high for the TG dataset, but then I wonder what is the gap at 17m between the 2014-15 and 2015-16 cores. It might suggest depth offset between the two cores. Please discuss.

For matching d18Oatm records, why did you only use the NGRIP data as the reference? There are clear discrepancies in the values (probably regardless of the matching quality) for some periods (~73 and 64-49 ka). I think Siple Dome d18Oatm data (Severinghaus et al., 2009) is of higher precision (not only measurement precision but also smaller and smoother thermal fractionation) and resolution, and Siple Dome and TG were measured in the same lab. Siple Dome also has the data younger than ~63 ka where NGRIP data is lacking. So there seem good reasons that you should try using it as well (of course you have to match SD to AICC2012 using CH4). There may be a hope to match around 60-65 using small fluctuations in d18Oatm.

P6, L1-3: You should take into account the potential age error due to linear interpolation between tie points. The comparison between linear and cubic spline interpolation is insufficient as the demonstration of the age uncertainty between tie points. You should consider using available gas records as much as possible (CO2, d18Oatm; see comments above and below).

P6, L9-16: Agreement of TG CO2 with existing records is overall very good. However, I think the decision not to use CO2 for the synchronization between 60 and 65 ka is not satisfactory especially because there is no other tie points. You should at least try matching CO2 and look how the resulting chronology look reasonable or not.

P6, L20: See the comment above about the dust and Ca.

P7, L19-22: See comment above about CH4 for 0-4 m.

[Figure]

P7, L31-34: Explanation is insufficient. What do you mean by "surveying the value-matched or correlated data"? How exactly did you consider resolution and analytical errors.

P8, L1-4: The offset of 20 cm is quite large when comparing laboratory measurements by CFA and discrete samples. Please explain the possible causes for this. Why don't you correct the CFA depth assignment by matching the depths of the sharp CH4 features? Why is the 300 yr estimate a conservative one? You have other CH4 tie points where you have much steeper depth-age slope, so it does not sound conservative at all.

P8, L9-10: Absolute age uncertainty attached to AICC2012 for this age range is probably incorrectly cited. Please check. Also, it is useful to refer to Chinese speleothem ages (using CH4 and d18Oatm) for the possible range of absolute age error for the studied period.

P8, L14: It is common to use small 'a' for the term Delta-age (not Delta-Age).

P8, L16-18: A better explanation would be that delta-age depends on firn thickness (ice or water equivalent) and accumulation rate, and the firn thickness depends primarily on temperature and accumulation rate.

P8, L24: I think a weakness of the delta age estimation and whole discussion based on it is that there is no ice and gas age estimates for the same depth, so the uncertainty of delta-age depends on the uncertainties of ice and gas ages between tie points, which is not evaluated well. You should try to have more constraints on the gas age (with CO2 or d18Oatm) between 60 and 69 ka where you have the very large delta-age (which is the basis for your argument of "virtually zero accumulation rate").

P8, last paragraph: Discussion here is too qualitative (with the words like "near zero", "where ice accumulates very slowly"). Please be more quantitative by giving possible range for the surface mass balance in the TG accumulation zone in MIS 4 from your

data.

P8, L35-36: d15N is not only controlled by gravitational fractionation. You should introduce it appropriately.

P8, L37 - P9, L1: I agree that d15N likely reflect firn thinning during MIS4, but the change is not linear with respect to delta-age. About half of the d15N change actually occur at around 71 ka when delta-age is still stable at ~4 kyr, and it is in fact before MIS4 (so this should also be discussed in your climatic discussion part). You should definitely discuss this large change while delta-age is small and stable, and in doing so, also run firn models for the high and low d15N around 70-73 ka. You might get some idea on how much non-gravitational signal could be contained in d15N data, or how much accumulation reduction is needed to explain d15N at that change (assuming that the change is purely gravitational). Another exercise is to use the presumed ratio (firn thickness in ice-equivalent) / (real firn thickness), which seems stable over wide range of temperature and accumulation rate (~0.7 from Parrenin et al., 2012, CP) and use the d15N and delta age to estimate accumulation rate, for example at 72, 70 and 61 ka and some times in between. I think you obtain a mm or so for the 10000-yr delta-age with d15N around 61 ka.

P9, L18-20: You should actually put the constraint ice age > gas age.

P9, L22: "onset of the last glacial period" is confusing as it is often mean the MIS5e-5d transition.

P9, L30-31: Suggesting the accumulation control on the depth-age curve instead of thinning based on the delta-age and d15N while avoiding deeply discussing delta-age uncertainty and d15N is not acceptable (see above).

Conclusions P11, L26: The statement "virtually zero net accumulation" needs more solid basis and quantification as commented above.

The rest of the discussion (about atmospheric circulation and ice sheet) may change

after the revision with deeper look into your data, so I would not review it (and it is not my speciality in any case).

---

## Author Comment (AC1) · 13 Nov 2018

**Response to Referee #1**

**General comments:** I would strongly advise to reorganize the paper in separated sections for more clarity. The way it is now, you continuously go back and forth between the sites and methods. I would suggest the following organization: Introduction, Field sites and analytical methods (–> presentation of your sites and the measured data in the field and in the lab + analytical uncertainties), Age models (-> choice of tie-points and chronological uncertainty propagation for both TG and TD), Results -> Δage and Discussion. Your manuscript would gain in clarity and would guide the reader toward your results and interpretations. You should avoid the listing of sites and data in the text and instead propose tables summarizing the data/sites information you need. This is particularly true for the blue ice sites you cite in the text and the different measurements performed on your cores.

We thank referee #1 for helpful comments. We will reorganize the text of the paper so that there is an "Age Models" section preceding the "Results" section, and therein the choice of tie points and chronological uncertainty will be discussed with greater justification for our tie point selections. We will present all metadata concerning the measurements in a table so it is clear which measurements were made on which cores, at which institutions, field versus lab, etc. Please find responses to specific comments below.

**Specific comments & Technical corrections:**

ABSTRACT:

- line 24: "Taylor Glacier (Antarctica)"
We will add "(Antarctica)" to line 24.

- line 27: "low SNOW accumulation WITHIN the Taylor…"
We will add "snow" to line 27 and change "at the Taylor Glacier accumulation zone" to "within the Taylor Glacier accumulation zone."

- line 31: replace "Taylor Dome" (already used in the sentence) by "this area"
We will change "Taylor Dome" in line 31 to "this area."

INTRODUCTION:

Page 1:
-line 36: missing references for past atmospheric composition and a list of trace gases
We will add references for "paleoarchive of the Earth's past atmospheric composition" on line 36 including (Bauska et al., 2017; Petrenko et al., 2017; Schilt et al., 2014).

-lines 40-41: This statement is not true, close to bedrock folding can happen, disrupting the order of ice/gas layers, as seen for the bottom part of NEEM ice core in Greenland for example.
We will change "age of ice and air bubbles always increases with depth" on line 40 to "age of ice and air bubbles always increases with depth except at or near the ice/bedrock interface."

-line 41 "precise distance-age"-> from which reference is the distance measured?
The reference to "distance" on line 41 simply refers to any generic reference point from which distance is measured in a blue ice area. In the case of Taylor Glacier, distance is measured from a flag that marks the location of the Main Transect. The flag was originally placed at an arbitrary location along the transect, and it ensures continuity between different sampling efforts during different seasons. E.g. -58 m is always 58 m south of the flag.

These details are described by (Baggenstos et al., 2017), and we will indicate on page 1 line 41 that this paper provides these details.

-line 12: remove "with fast access to age information"

We will remove "with fast access to age information" on line 12.

-line 13: as precise as what? The previous method? Replace "have" by "present"

We will add "as precise as the aforementioned methods" on line 13. We will change "have" to "present" on line 13.

-paragraph 3: it would be easier for the reader if you summarize all in a table (site, location, period covered, references) and refer to it in the main text. Such a listing is difficult to follow with too many commas.

We do not think it is appropriate to add a table describing various blue ice areas because the paper is not a review of blue ice areas. There is already a published review of Antarctic blue ice areas that we cited in the original manuscript (Bintanja, 1999). We simply wished to point out that there are several blue ice areas that have been studied, however we will follow editorial guidance on this issue.

-line 29: replace "expands" by "extends" and replace "by developing ice and gas chronologies spanning" by "back to"

We will replace "expands" with "extends" on line 29. We would prefer not to replace "by developing ice and gas chronologies spanning the MIS 5/4 transition" with "back to" because "back to" implies that the archive is continuous back to the 5/4 transition, which it is not.

-Line 31-32: remove "the across-flow transect"

The relevant sentence in the original manuscript is, "In 2015 a new ice core was retrieved approximately 1 km down-glacier from the 'Main Transect,' the across-flow transect containing ice from Termination 1 through MIS 3 (Baggenstos et al., 2017) (Figure 1)."

We would prefer not to remove "the across-flow transect" from this sentence because we think it is important to define what the Main Transect is and to note its orientation with respect to the glacier flow explicitly.

-Line 34-36: "paleoarchive FROM TAYLOR GLACIER, where it was previously thought to be absent". Remove "larger context of". Replace "into" by "within". Replace "at Taylor Dome" by "of this region"

We will change line 34-35 to read "the description of a new MIS 4 paleoarchive from Taylor Glacier, where it was previously thought to be absent." We will remove "larger context of" in line 36. We will replace "into" with "within" in line 35. We will replace "Taylor Dome" with "of this region" in line 36.

FIELD SITE AND ANALYTICAL METHODS:

Page 3:
-line 5: if you are not citing an acronym, ice sheet is written without capital letters

We will change "Ice Sheet" to "ice sheet" in line 5.

-line 6: "northERN"

We will change "north" to "northern" on line 6.

-line 7: "ice EQUIVALENT accumulation"

We will change "ice accumulation" to "ice equivalent accumulation" on line 6.

-line 15: "80 km LONG ablation zone", and you already said it in the previous paragraph

We will remove "~ 80 km" from line 15.

-line 16: need rewording. I suggest the following: "Water stable isotopes obtained from an along-flow transect just below the equilibrium line"… "revealed uncontinuous ice covering the last glacial period" –

We will change line 16 to read "Water stable isotope data obtained from an along-flow transect from just below the equilibrium line to the terminus revealed ice from the last glacial period outcropping at sporadic places along the transect."

line 20: "revealed continuous records of ice from the Holocene to the last ice age, with ice of the last interglacial and older found…" references for this statement?
We included references for this statement. They are (Schilt et al., 2014), (Bauska et al., 2016), (Baggenstos et al., 2017), and (Buizert et al., 2014). They are found in lines 21-22.

-line 22: "the most COMMONLY USED archive" instead of utilized
We will change "utilized" to "commonly used" on line 22.

-line 27: Reference for the previous ice core study. Where was taken this new ice core compared to the previous study? Need more precision. What was the sampling problem with the previous record?
There is not a previous ice core study per se. We have worked on Taylor Glacier for over 5 years, and prior to the 2014-2015 field season the MIS 5/4 transition was thought to be missing from the glacier archive. Then in 2014-2015 we found the MIS 5/4 transition in a new location that was previously not sampled. The new location is 1 km down glacier from the Main Transect, which we note on line 34 and show in Figure 2a.

We will clarify in the text that we are not referring to a specific study, and we will reference relevant discussion in (Baggenstos, 2015).

-lines 30-33: need a reference
It is unclear what the referee wants referenced. If it is the $CH_4$ variability at DO16/17 then we will add an appropriate reference (Rhodes et al., 2015; Schilt et al., 2010). If it is the results from the -380 m PICO core there is not a reference because those data are unpublished until this manuscript.

-line 36: replace "in" by "of"…"$CH_4$ variations similar to those ASSOCIATED WITH DO19" or "corresponding to"
We will replace "in" by "of" on line 36. We will replace "similar to those at Dansgaard-Oeschger event 19" to "similar to those associated with Dansgaard-Oeschger event 19" on line 36-37.

-line 37: "$CH_4$ CONCENTRATION increase"
We will add "concentration" on line 37.

Page 4:
-line 3: replace "work" by "analysis", replace "spanning" by "section"
We will replace "work" with "analysis" on line 3. However we don't understand the meaning of the sentence if we replace "spanning" with "section" on line 3, so we prefer not to make that change.

-line 4: need rewording, proposition: "…$CH_4$ and $CO_2$ concentrations, which confirmed the MIS 4/5 transition record in the gas phase"
We will change line 4 to read, "sampled for laboratory analyses of $CH_4$ and $CO_2$ concentrations, which confirmed the MIS 5/4 transition record in the gas phase."

-line 6: spanning not properly used
We will state that the 0-9m and 17-19.8m sections were sampled, instead of using the word "spanning."

-lines 7-11: it would help to make a table for all the analyses performed on the different cores, with specification of the proxy measures, where, the time coverage of samples (or portion of core) and the method used for measurements, analytical uncertainty…
Other referees requested a similar table. We will make a table that summarizes the metadata for all analyses discussed in the manuscript including where the samples were taken, which coring device was used (BID or PICO), in which laboratory and what type of measurements were made, when the measurements were made, and the analytical uncertainty of each measurement.

-line 24: "resulted in a good agreement of our measurements with other…"
We will change line 24 to read, "resulted in a good agreement between our measurements and other Antarctic CH4 records."

Page 5:
-line 19: "… on archived Taylor Dome ICE CORE samples…"
We will change line 19 to read, "Discrete measurements of CH4 and CO2 were made at OSU on archived Taylor Dome ice core samples…"

-line 22: "(~10g OF ICE, …)"
We will add "of ice" after "~ 10 g" to line 22.

Page 6:
-line 10: 'CO2 CONCENTRATION decrease", again later
We will change line 10 to read, "CO2 concentration decrease" and "CO2 concentration increase"

-line 11: remove "and"
We will remove "and."

-line 13: value of the offset?
We will state the value of the offset on line 13. It is ~ 13 ppm at 61.5 ka.

-line 15: need rewording, a proposition: "…younger ages. Therefore, we refrain from further align the CO2 rises together for better consistency."
We will reword line 15 to read, "…younger ages. However $CO_2$ offsets between ice cores are observed (Luthi et al., 2008), and we cannot reject the possibility that the offsets are real. Therefore, we refrain from value-matching the $CO_2$ rise.

-line 17: why not use the d18Oatm of Vostok or TALDICE instead of NGRIP? You would have a complete record over your period of interest on AICC2012, but potentially with a lower resolution.
Referee 4 had a similar comment. TALDICE d18Oatm is unpublished, as far as we know, Vostok d18Oatm is low resolution, and the two records do not agree precisely in terms of when the light excursion begins at the MIS 4/3 transition. Siple Dome d18Oatm would be the best choice, but the new Seltzer timescale does not extend beyond 50,000 years ago and the old timescale is not synced with AICC2012 (Seltzer et al., 2017). We think it is beyond the scope of this paper to sync Siple Dome to AICC2012, and we are aware of other work already in progress towards this goal. NGRIP is helpful because it provides variability to match where CH4 variations are small, and it is consistent with AICC2012 (which is the timescale that we use to tie to EDML CH4). We note that the EDML d18Oatm shows quite good agreement with NGRIP d18Oatm in terms of the onset of the MIS 5/4 excursion. Since EDML is lower resolution than NRIP, we still pick tie points using the NGRIP d18Oatm, however we show the EDML agreement in our revised Figure 3 (black x' in figure below). (Capron et al., 2010; Landais et al., 2007)

Revised Figure 3:

[Figure]

-line 23: replace "has" by presents"
We will change "Taylor Glacier d18Oice has more variability…" to "Taylor Glacier d18Oice is more variable…"

-paragraph 3: I am not very much convinced by value matching for dating. We do not really understand the usefulness of the -380 core data until the idea of similar firn conditions. This and the following argument are important for your interpretation later. This paragraph needs rewording.

The reviewer points out that paragraph 3 on page 6 is poorly worded because the purpose of the -380 m

core is not clear from the beginning. We think the -380 m core is useful because it shows similar trends in the gas data ($CO_2$, d18Oatm, and CH4) as the MIS 5/4 cores from ~ 1 km down glacier. This suggests stratigraphic continuity between the Main Transect (where the -380 m core was drilled and where all previous work on Taylor Glacier has been conducted) and the new MIS 5/4 drill site. The d15N-N2 is similarly low in the -380 m core as in the 5/4 cores. The implication of this is that the archive of ice found at the Main Transect likely originated from the same accumulation zone as the 5/4 cores. In other words, the Taylor Glacier ablation zone is not a confounding mixture of ice that has flowed from different deposition areas at different times. Rather, the archive appears to be a stratigraphically continuous and intact record with a common source deposition zone.

We will reword the paragraph so that readers understand this point clearly and know the purpose of the -380 m core at the beginning of the paragraph.

The reviewer was also not convinced that our tie point choices for the -380 m core were robust. In the original manuscript we value-matched the -380 m $CH_4$ data because the data are sparse and we lack the context of a longer record to confidently match the beginning and ending of transitions and features like for the MIS 5/4 cores. However, we recognize that value-matching cannot provide unique ages for the -380 m core, especially before the MIS 4/3 transition where the variability in the gas records is small (i.e. one could assign different ages to a given depth). We intend to rewrite paragraph 3 on page 6 to de-emphasize the dating of the -380 m core, as it was not our intention to develop a robust chronology for that core. We think the important thing is that the -380 m core contains gas bubbles that span the MIS 4/3 transition and some of late MIS 4. We would like to emphasize that CO2, d18Oatm, and CH4 are all changing across the MIS 4/3 interval in the -380 m core, similar to in the new MIS 5/4 cores, and to find variability in all three of those parameters that is synchronous and of the right magnitude is unique. Thus we think assigning the age of the -380 m core broadly to the MIS 4/3 transition and late MIS 4 is robust, even if the exact chronology is uncertain.

In the revised text we will de-emphasize the dating of the -380 m core, present the tie points we chose more clearly in a table, and display the -380 m data in a new figure so that it is not cluttered with the 5/4 BID data. We will also emphasize the purpose of interpreting it – to show evidence for stratigraphic continuity between the MIS 5/4 drill site and the Main Transect, which implies that the source accumulation zone for the Taylor Glacier ice archive was the same through time. We think we are justified interpreting the -380 m core this way without necessarily improving the certainty of the -380 m chronology.

Page 7:
-paragraph 2: not useful, could be removed.
One puzzle that has emerged from our work at Taylor Glacier is why the MIS 4 dusty ice was so elusive to find, whereas the LGM dusty ice is clearly represented and is even visible at the surface. Paragraph 2 addresses this problem and offers an explanation - that the MIS 4 ice is quite thin. We prefer to keep this paragraph, but we will emphasize the usefulness of the paragraph at the beginning.

3.2 ANALYTICAL AND AGE MODEL UNCERTAINTIES

Page 7:
-line 18: "is likely"
We will change line 18 to read, "The mismatch between field and laboratory CH4 in the top 0-4 m of the core is likely due to…"

-lines 19-22: not clear. You say that you consider the 2015-2016 data as uncontaminated, but as the same record differ from the lab, in the end you do not interpret the data… but you did later in the text… Moreover, you did not discuss the reasons that could explain why the records are so different. I would possibly keep the tuning, but associate it with a much larger uncertainty than the other points due to the mismatch with the lab data. Then only use the CH4 data in grey area for dating purposes and no more.
Other referees had similar comments about the 0-4 m CH4 data and our choice to tie the field data to AICC 2012. We stated why we think the laboratory and field records are different – it is likely because resealed cracks in the glacier surface affected the CH4 in the lab samples but not the field samples. These kinds of

cracks tend to penetrate the top 4 m of ice (line 19) at Taylor Glacier, and CH4 measurements in the 0-4 m surface ice have looked wrong in the past, so this is not a new observation (Baggenstos, 2015).

We prefer not to assign larger uncertainty in this section because we think we matched the correct peaks in CH4. Instead we would prefer to emphasize how we do not interpret the top 4 m rigorously. We will explain and rationalize in the text more clearly what we chose to do. We only present the CH4 data from 0-4 m for completeness, and the delta age in the 0-4 m is not critical for our interpretations (the high delta age values occur at ~ 5.5 m). There are no CO2 or d18Oatm data from the 0-4 m section to interpret, and the delta age in 0-4 m section has very little bearing on the overall story we present. Thus we think it is justified to offer our best plausible gas age scale for 0-4 m, clearly show the discrepancy between the laboratory and the field data, and state that the 0-4 m section could be contaminated but that it will not be used in our interpretations that follow. We will follow the editor's guidance on this issue if needed.

The discussion about the analytical uncertainty should be following the presentation of the analytical methods.

We will reorganize the text so that the uncertainty discussion comes after the analytical methods, similar to comments from other referees.

–paragraph 4: I am not convinced about your argument for the confirmation of data. From the looks of the data presented on Figure 2, I would say that your choice of markers is not convincing, I would have chosen differently… From your Figure 3, I understand that your choices were made in order to align together the records you cite as confirming your alignment (e.g. nssCa). I would recommend to change the way you presented your figure 2 to make the reader see by himself why you choose these tie-points and not others. You should focus more on this aspect, which is the base of your discussion later, it would strengthen your work. Not necessarily in the main text, it could be an appendix.
Though referee 1 would have chosen tie points differently, he or she did not say exactly how. Thus it is difficult to defend our tie point choices specifically to this referee's criticism. Generally speaking, in the revision we will provide stronger justification for the tie point choices we prefer. Other reviewers also asked for information like this.

Specifically we will make Figure 2, Figure 3, and Figure 4 clearer so that readers can see easily why we picked certain tie points, and we will rationalize our choices thoroughly in the text.

We have revised our final tie point choices. These are summarized in the preceding summary document, but the main changes from the original manuscript include: (1) 6 new nssCa tie points that match variability between TG nssCa and EDC nssCa, (2) only 3 particle count tie points matching TG particle counts with EDC laser dust (instead of the original 9), and (3) 2 additional d18Oice tie points that match variability in TG water isotopes with EDC water isotopes. We opted to include more nssCa tie points instead of particle count tie points because the nssCa data are more quantitative, we can compare to EDC nssCa (a like-like comparison) instead of comparing insoluble particle counts to EDC laser dust (different measurements), and the nssCa record is less noisy than the particle count record. We hope that the addition of two more d18Oice tie points helps readers see the similarity in the d18Oice variability at TG and EDC for AIM 19 (72.5 ka) and AIM 20 (76 ka).

The gas tie points between CH4 have not changed substantially from the original manuscript. The two tie points that match TG d18Oatm to NGRIP d18Oatm have changed slightly based on feedback from reviewers. The oldest one linking 19.8 m to 74.65 ka now ties 19.27 m to 73.74 ka in order to tie the lowest measured d18Oatm to the local minimum in the NGRIP record. The other d18Oatm tie point was shifted to tie to the midpoint of the MIS 5/4 transition in NGRIP.

The tie points and the final match are shown in our revised Figure 3 (above).

Page 8:
-lines 3-4: "20 cm = 300 years", based on what? Which chronology?
The relevant part of the sentence in question is: "there is a 10 cm offset between the continuous field CH4

and the discrete laboratory CH4, and a 20 cm offset between the continuous laboratory CH4 and the discrete laboratory CH4. 20 cm depth uncertainty equates to, conservatively, 300 years on the gas age scale near the onset of Dansgaard-Oeschger event 19." Here we were estimating the age error associated with depth offsets between the cores, the largest of which was 20 cm at DO 19. We believe it is clear when we say "on the gas age scale" that we are using our chronology.

Referee 4 pointed out that our conservative estimate was not conservative enough. We think referee 4 was actually misreading the axes of Fig 3B, but we did realize upon closer inspection that the slope of the gas age-depth curve in its steepest segment is 20.8 yr/ cm. So our conservative estimate of the effect of a 20 cm depth offset is ~ 420 years. We will change "300" to "420" in the text and propagate the uncertainty to the chronologies and the delta age calculations.

Lines 5-8: This is not a proper argument. If you say that both CH4 data from TG and EDML are similarly smoothed in the firn column, you are implying that they have similar firn conditions (i.e. accumulation rates, firn depth…). Is it the case?

Our statement is based on the observation that the magnitude of the changes in CH4 and CO2 concentration are similar in TG and EDML. It appears that the CH4 signal in TG is more smoothed than EDML at DO 18 (65 ka), but this is the only place in the record where the magnitude of the changes is different. It makes sense that the amount of smoothing in the firn would be the increasing between 60-70 ka where delta age is increasing and accumulation is presumably decreasing. There are no abrupt events in the gases during this interval besides DO 18, so we must use this as our metric for estimating the smoothing. EDML CH4 increases to ~ 515 ppb while TG only reaches ~ 475 ppb. Of course the peak CH4 at DO 18 is only defined by one data point at EDML, but if we assume it is correct then the maximum CH4 concentration recorded during DO18 at TG is 40 ppb lower than that at EDML. We also note the differences between d18Oatm at TG versus NGRIP during the same interval. Therefore we think the firn smoothing must be different in the two cores during this interval, and we think it is likely due to increasing the height of the lock-in zone consistent with delta age increasing to extremely high values as accumulation decreased.

Our initial statement was meant to reflect how the cores generally agree in terms of the magnitude of smoothing across the whole record, but we neglected to explore the larger discrepancies near DO18 that are likely due to firn smoothing.

We will change what we wrote in the paper to more accurately reflect our assessment of the degree of smoothing at DO 18. We don't think the smoothing is significantly contributing to the uncertainty in our tie point selection.

-lines8-9: Analytical noise… why is that? What is the measurement uncertainty of your method?
Because there is very little analytical noise, < 0.5 ppb. We will elaborate in the text and provide a value.

-lines 9-12: Please, when using a chronology as reference, make sure of the uncertainty values you cite… What you wrote is not correct. The AICC2012 chronology uncertainty over your period of interest (i.e. ~65-74 ka) at EDML is ranging between 1500 years and 1400 years (cf. supplementary material of Veres et al., 2013). The values you have indicated correspond to the uncertainty of the ice and gas chronology at the orbital scale, prior to the last interglacial. -Following all this discussion of uncertainties, what are the uncertainties associated with your ice and gas chronologies for TG? You never gave a value and I do not see them on your figures. The same for your revised TD chronology.
The reviewer notes an error in the original manuscript – we cited the wrong absolute age uncertainty associated with the AICC2012, which we use as a reference scale for dating the gas and ice records in the new Taylor Glacier cores. The reviewer pointed out the 1-sigma uncertainty in EDML is actually 1400-1500 years between 74-65 ka and can be found in the supplementary material of (Veres et al., 2013), but the supplementary material only gives the uncertainty in the ice age chronology for 74-65 ka. The main text gives the uncertainty in the gas age chronology for EDML (figure 2 in Veres et al. 2013), which is also ~ 1500 years. For the ice chronological uncertainty, the uncertainty in EDC should be considered instead of EDML because we tie our dust data exclusively to EDC to obtain the ice age scale. The 1-sigma uncertainty for the EDC ice age scale is ~1800-2500 years for the time period 74-65 ka.

We will correct the uncertainty we cite for the AICC2012 reference age scale to 1500 years for the gas phase and 2500 years (taking the maximum) for the ice phase so that it is consistent with the information in (Veres et al., 2013).

In general we will present the uncertainty estimation in the revised manuscript similarly to the original manuscript – i.e., each tie point we picked for Taylor Glacier and Taylor Dome is assigned a maximum and minimum age to estimate the uncertainty of the match, and these estimated uncertainties are propagated through the chronology by interpolating between the maximum and minimum ages at each tie point. The age range at each tie point is assigned by considering (1) the resolution of the data for a given feature that we matched, (2) the analytical uncertainty of the data that we matched to, and (3) how robust (or possibly ambiguous) the matched feature was (i.e. could we be matching the wrong feature?).

In the revised text we are explicitly displaying the errors along with the age models (shading in Figure 5A and Figure 5C below). The uncertainty range is also included in the delta age calculation (shading in Figure 5B and Figure 5D below).

Figure 5 – Age models for new Taylor Glacier 5/4 BID cores (A), Taylor Glacier delta age and $\delta^{15}N$-$N_2$ (B), and Taylor Dome revised age models (C), and Taylor Dome delta age and $\delta^{15}N$-$N_2$ (D). Red shading on Taylor Glacier gas age chronology and delta age indicates ice shallower than 4 m where surface cracks may affect the $CH_4$ age matching.

[Figure]

**3.3 ΔAGE AND COMPARISON TO TAYLOR DOME**

Page 8:
-line 7: Temperature and accumulation are not the only factors influencing Δage. All factors acting on the firnification process do as they impact on the firn depth variability. What about insolation of wind stress affecting the snow metamorphism into ice?

We do not understand what referee 1 means by "insolation of wind stress." If he/she means wind stress, we did not include this because we think the effects on delta age are secondary. If he/she means insolation, then we also did not include this because insolation effects on delta age are also secondary. Temperature and accumulation are the primary controls on delta age. We are unaware of firn densification models that include wind stress or insolation with major influence on firn evolution. If insolation and wind stress affect delta age, we think they are of secondary importance to temperature and accumulation.

-line 19: remove "on the order of hundreds of years", it is given by the lower limits just before. Change "smaller" in "smallest"
We will remove "on the order of hundreds of years" from line 19, and we will change "smaller" to "smallest".

-line 21: replace "at" by "for"
We will replace "at" with "for" on line 21.

-lines 26-27: ok for the two sources of uncertainties, but you forgot to take into account the absolute uncertainty of the ice and gas chronologies. You have ~1500 years uncertainty from the AICC2012 age scale, consequently the uncertainty of your new chronology should be around ~1600 years for the gas age, and ~1530 years for the ice age (I took one random range from your choice of tie-points). Then your maximum and minimum Δage should be obtained from the (ice age - 1sigma)-(gas age + 1sigma) and (ice age + 1sigma)-(gas age - 1sigma). You should give an approximate value of the Δage uncertainty for the reader to have an idea of the significance of your Δage values later.

We accounted for the uncertainty in delta age in the original manuscript by propagating our tie point uncertainty (described above) using the calculation that the reviewer describes here. We did not propagate the absolute uncertainty from the reference age scale, but we note that the actual uncertainty in delta age acquired from the reference age scale should be much less than the total propagated uncertainty from EDML (1500 years) and EDC (2500 years) because these uncertainties are correlated in depth. I.e., it is unlikely for one to be too old while the other is too young. Because the uncertainty estimates that we placed on our tie points are very generous, we think that we already estimate a reasonable uncertainty for delta age (between ~ 2000-4000 years, Figure 5). This uncertainty range compares well with the uncertainty cited in (Baggenstos et al., 2018).

-line 30: "10 ka"+/- ??? uncertainty needed.
We will add the uncertainty to the text in line 30.

-line 33: then why is it so different? Replace "high" by "large"
We explain the difference in terms of accumulation gradients beginning on the last paragraph of page 9. We will replace "high" with "large" on line 33.

-line 34: now you talk of the influence of wind, but not before…
We talk about wind in terms of scouring, or removal of snow. Not in terms of influencing the snow grain metamorphism, which we think is of secondary importance in the firn densification. Wind scouring works to reduce net accumulation. Whenever we write "accumulation" in this manuscript, we are referencing the combined or net effect of addition of snow by precipitation AND removal of snow by wind scouring.

Page 9:
-paragraph 1: I do not think that the last sentence is necessary, you should delete it.
We will delete the last sentence in paragraph 1.

-paragraph 2: You should gather together in one section the chronology construction for your two sites, with the proper calculation of their respective uncertainties.
We will discuss the construction of the Taylor Dome chronology in a preceding section, analogous to the Taylor Glacier chronology.

-line 13: "in the same manner AS described"

We will add "as" to line 13.

We will simply give 0 values where the minimum error estimation causes the negative delta age artifact. No, the uncertainty is not Gaussian. It is not possible to assign a Gaussian error to our tie points given our methods.

This is explained in the Results and Discussion section of the text. The accumulation gradient switches at the LGM relative to MIS4.

-lines 22-25: I disagree with this statement. It comes too soon. For TG, not located on a dome, ice thinning and ice flow are very important factors that could affect the depth-age relationship. For TG you cannot interpret directly your variations on Δage in terms of accumulation. To distinguish between the major influences of thinning and accumulation, you need an ice flow model. If your ice flow model indicate that there are no significant thinning variations, then and only then you can interpret it in terms of accumulation. Moreover, you give absolutely no justification for your favour toward accumulation changes, and you do not explain why you disregarded the thinning influence.

The statement in question is "The implication of the relatively 'normal' delta age is that accumulation at Taylor Dome did not dramatically change at the onset of the last glacial period or throughout MIS4 as Taylor Glacier did. Comparing the depth-age relationships in the new Taylor Glacier core versus the Taylor Dome ice core highlights the difference in accumulation between the two sites."

We are confused by referee 1's comment. If he/she means that thinning of the ice could affect delta age, then we disagree. Ice thinning can affect the slope of the depth-age relationship, but it cannot affect the stratigraphic order of bubbles and ice at depth, i.e. the Δage = ice age-gas age at any given depth will remain constant with any degree of thinning. In the revised manuscript we will reference (Parrenin et al., 2012) and point out that delta depth (the difference in depth between ice and gas of the same age) can evolve with time due to thinning and glacier flow, but delta age is fixed when gas diffusion effectively ceases at the lock-in depth.

If referee 1 means thinning of *firn* at the original deposition site, then we agree that in an extreme case this could affect the delta age because the process occurs before bubble close off. However, in order to achieve a delta age of 10,000 years you would have to thin the firn such that 10,000 annual layers of snow were included in the firn pack before bubble close off. For example, if a typical delta age in east Antarctica is ~ 3000 years, then this means thinning firn to ~30% its thickness, which seems outside the realm of possibility even on the flank of a dome.

We do see how referee 1 takes issue with the second part of the relevant statement - interpreting the depth-age relationship strictly in terms of accumulation changes without considering thinning. We simply meant to state that the depth-age relationship supports our interpretation of the high delta age values. We will change the wording to read, "Comparing the depth-age relationships in the new Taylor Glacier core versus the Taylor Dome ice core supports the notion of large accumulation differences between the two sites. Though differences in thinning between the two accumulation zones likely exist, we think that this effect is secondary to accumulation in terms of setting layer thickness." We will also reference (Morse et al., 1999; Morse et al., 2007; Morse et al., 1998) and other relevant work, in which large differences in thinning across Taylor Dome were rejected based on multiple lines of evidence, one being that layers do not show the same thinning trends with depth.

-last paragraph: you should give the modern values of accumulation measured at these two sites. It would give an idea of how much your prior assumption of all differences are due to accumulation changes is valid for modern times.

Modern accumulation rates at Taylor Dome were determined by (Morse et al., 1999), and a good illustration of a steep gradient in the modern accumulation across a 30 km north-south transect on Taylor dome is shown in (Morse et al., 2007). Accumulation changes along the gradient from 14 cm/yr to 2 cm/ yr.

(Kavanaugh and Cuffey, 2009; Kavanaugh et al., 2009a; Kavanaugh et al., 2009b) describe the modern accumulation rate in the Taylor Glacier catchment, which is informed by the accumulation gradient reported in (Morse et al., 2007). Taylor Glacier is estimated to receive 3-5 cm/yr. (Kavanaugh et al., 2009b) also reports the fact that Taylor Glacier is in a rain shadow and is much drier than the regional average, with references to (Morse et al., 2007; Morse et al., 1998).

We will include this information in section 2 – Field site and analytical methods.

Page 10:
-lines 14-18: give values for the LGM reconstructed accumulation at both TD and the virtual sites. This gradient is reverse from yours. Why do you use it then? The useful result from this study to you is only "the opposite accumulation gradient (decreasing from south to north) for ice older than 60ka".
We include it because it is interesting to us that it shifted between the two time periods. It expands on a storyline in the literature that is related to the errors in the original TD age model. (Morse et al., 1998) first predicted the shift in storm gradients based on radar data, and we find it interesting that our delta age data support this.

-lines 18-26: bring nothing more, just show support for the LGM gradient that is different from yours. I would advise to remove these sentences.
This sentence becomes even more important given referee 1's prior comments about thinning. The authors of (Morse et al., 1998) rejected the notion of differential flow (i.e. thinning) because the layer thicknesses did not vary in the same way with depth.

-last paragraph: remove the first two sentences, you are only rewording your results.
We will remove the first two sentences of the last paragraph on page 10.

Page 11:
-line 4: need a reference for this statement.
We will add a reference for this statement.

-paragraph 2: the MIS 4 gradient is similar to modern conditions. Are modern conditions in agreement with your proposed hypothesis?
Yes, though there are no data available to constrain the modern delta age at the probable deposition site for our samples.

FIGURES & TABLES:

Figure 1: I would advise to change the organization: a-Antarctica map, b-landsat imagery, simplified map of TG.
We will change the organization of figure 1 according to referee 1's suggestion.

Figure 2: The way the data are presented now, one can strongly argue your chosen tuning points. The scales are two small to see the consistency between the associated variability. I am not at all convinced about your tie-point between the d18Oice of EDC and TG, records present different variability. I would advise to remove from the legend the last two sentences. -Tables 1&2: You should add some indications on your figure 2, on the reference records, to directly make the link between the tables and your chosen points (e.g. DO19…). In Table 2 legend, remove the sentence "Ice phase…"
Unfortunately referee 1 did not state how he or she believes that one can strongly argue against the chosen tie points, which makes it difficult to rebut this point specifically. In the revision we will further justify the d18Oice tie point along with other tie points used to construct the chronologies. We will add labels for important features to ease comparison between the graphical display of tie points in Figure 2 and the list of tie points in Table 1, e.g. "DO 19" or "DO 18". We will also add a figure graphically displaying the Taylor

Dome tie points, analogous to Figure 2, which only shows Taylor Glacier tie points. Generally speaking, we will justify our tie point choices more clearly in the text.

Figure 3: I would say that there is absolutely no point in plotting together records that were tuned together, or if you really want to, it should be in an appendix. You already use some other untunned records to validate your chronologies. I would leave here only 1 gas, 1 ice records, and then the (b) part of the figure. You should extend the lines for the identification of MIS limits to the bottom of the figure for more clarity. In the legend your last sentence is not necessary, you could delete it.

We think displaying the tuned records helps the reader to see the variability we were matching in Figure 2 and shows how the data between the tie points agree. Showing the matched data in this way is common practice. The figure also puts the environmental records we are discussing in context. We will put the MIS and other identifiers at the bottom for clarity. We will delete the last sentence in the caption.

Figure 4: Same comments as for Figure 3. Your should keep consistent the colours of curves from one figure to another. Why didn't you remove the three points in questions and simply state it in the measurement section?

Same responses for Figure 3. We will keep the colors consistent between the two figures.

Revised Figure 4:

[Figure]

Why didn't you remove the three points in questions and simply state it in the measurement section?
We want to keep the three data points for completeness. It particularly aids readers who are using the same data set, or want to verify that the data set is similar to his/her own copy of the data.

Baggenstos, D., 2015. Taylor Glacier as an archive of ancient ice for large-volume samples: Chronology, gases, dust, and climate, Scripps Insittute of Oceanography. University of California, San Diego.

Baggenstos, D., Bauska, T.K., Severinghaus, J.P., Lee, J.E., Schaefer, H., Buizert, C., Brook, E.J., Shackleton, S., Petrenko, V.V., 2017. Atmospheric gas records from Taylor Glacier, Antarctica, reveal ancient ice with ages spanning the entire last glacial cycle. Climate of the Past 13, 943-958.

Baggenstos, D., Severinghaus, J.P., Mulvaney, R., McConnell, J.R., Sigl, M., Maselli, O., Petit, J.R., Grente, B., Steig, E.J., 2018. A Horizontal Ice Core From Taylor Glacier, Its Implications for Antarctic Climate History, and an Improved Taylor Dome Ice Core Time Scale. Paleoceanogr. Paleoclimatology 33, 778-794.

Bauska, T., Baggenstos, D., Brook, E.J., Mix, A.C., Marcott, S.A., Petrenko, V.V., Schaefer, H., Severinghaus, J.P., Lee, J.E., 2016. Carbon isotopes characterize rapid changes in atmospheric carbon dioxide during the last deglaciation. Proceedings of the National Academy of Sciences of the United States of America 113, 3465-3470.

Bauska, T., Brook, E., Marcott, S.A., Baggenstos, D., Shackleton, S., Severinghaus, J., Petrenko, V., 2017. Abrupt climate change events and atmospheric CO2: constrains from ice core d13C-CO2 during the last glacial period, PAGES Open Science Meeting, Zaragoza, Spain.

Bintanja, R., 1999. On the glaciological, meteorological, and climatological significance of Antarctic blue ice areas. Rev. Geophys. 37, 337-359.

Buizert, C., Baggenstos, D., Jiang, W., Purtschert, R., Petrenko, V.V., Lu, Z.T., Muller, P., Kuhl, T., Lee, J., Severinghaus, J.P., Brook, E.J., 2014. Radiometric Kr-81 dating identifies 120,000-year-old ice at Taylor Glacier, Antarctica. Proceedings of the National Academy of Sciences of the United States of America 111, 6876-6881.

Capron, E., Landais, A., Lemieux-Dudon, B., Schilt, A., Masson-Delmotte, V., Buiron, D., Chappellaz, J., Dahl-Jensen, D., Johnsen, S., Leuenberger, M., Loulergue, L., Oerter, H., 2010. Synchronising EDML and NorthGRIP ice cores using delta O-18 of atmospheric oxygen (delta O-18(atm)) and CH4 measurements over MIS5 (80-123 kyr). Quat. Sci. Rev. 29, 222-234.

Kavanaugh, J.L., Cuffey, K.M., 2009. Dynamics and mass balance of Taylor Glacier, Antarctica: 2. Force balance and longitudinal coupling. J. Geophys. Res.-Earth Surf. 114, 11.

Kavanaugh, J.L., Cuffey, K.M., Morse, D.L., Bliss, A.K., Aciego, S.M., 2009a. Dynamics and mass balance of Taylor Glacier, Antarctica: 3. State of mass balance. J. Geophys. Res.-Earth Surf. 114, 7.

Kavanaugh, J.L., Cuffey, K.M., Morse, D.L., Conway, H., Rignot, E., 2009b. Dynamics and mass balance of Taylor Glacier, Antarctica: 1. Geometry and surface velocities. J. Geophys. Res.-Earth Surf. 114, 15.

Landais, A., Masson-Delmotte, V., Nebout, N.C., Jouzel, J., Blunier, T., Leuenberger, M., Dahl-Jensen, D., Johnsen, S., 2007. Millenial scale variations of the isotopic composition of atmospheric oxygen over Marine Isotopic Stage 4. Earth Planet. Sci. Lett. 258, 101-113.

Luthi, D., Le Floch, M., Bereiter, B., Blunier, T., Barnola, J.M., Siegenthaler, U., Raynaud, D., Jouzel, J., Fischer, H., Kawamura, K., Stocker, T.F., 2008. High-resolution carbon dioxide concentration record 650,000-800,000 years before present. Nature 453, 379-382.

Morse, D.L., Waddington, E.D., Marshall, H.P., Neumann, T.A., Steig, E.J., Dibb, J.E., Winebrenner, D.P., Arthern, R.J., 1999. Accumulation rate measurements at Taylor Dome, East Antarctica: Techniques and strategies for mass balance measurements in polar environments. Geogr. Ann. Ser. A-Phys. Geogr. 81A, 683-694.

Morse, D.L., Waddington, E.D., Rasmussen, L.A., 2007. Ice deformation in the vicinity of the ice-core site at Taylor Dome, Antarctica, and a derived accumulation rate history. Journal of Glaciology 53, 449-460.

Morse, D.L., Waddington, E.D., Steig, E.J., 1998. Ice age storm trajectories inferred from radar stratigraphy at Taylor Dome, Antarctica. Geophysical Research Letters 25, 3383-3386.

Parrenin, F., Barker, S., Blunier, T., Chappellaz, J., Jouzel, J., Landais, A., Masson-Delmotte, V., Schwander, J., Veres, D., 2012. On the gas-ice depth difference (Delta depth) along the EPICA Dome C ice core. Climate of the Past 8, 1239-1255.

Petrenko, V.V., Mith, A.M.S., Chaefer, H.S., Riedel, K., Brook, E., Baggenstos, D., Harth, C., Hua, Q., Buizert, C., Schilt, A., Fain, X., Mitchell, L., Bauska, T., Orsi, A., Weiss, R.F., Everinghaus, J.P.S., 2017. Minimal geological methane emissions during the Younger Dryas-Preboreal abrupt warming event. Nature 548, 443-446.

Rhodes, R.H., Brook, E.J., Chiang, J.C.H., Blunier, T., Maselli, O.J., McConnell, J.R., Romanini, D., Severinghaus, J.P., 2015. Enhanced tropical methane production in response to iceberg discharge in the North Atlantic. Science 348, 1016-1019.

Schilt, A., Baumgartner, M., Schwander, J., Buiron, D., Capron, E., Chappellaz, J., Loulergue, L., Schupbach, S., Spahni, R., Fischer, H., Stocker, T.F., 2010. Atmospheric nitrous oxide during the last 140,000 years. Earth Planet. Sci. Lett. 300, 33-43.

Schilt, A., Brook, E.J., Bauska, T.K., Baggenstos, D., Fischer, H., Joos, F., Petrenko, V.V., Schaefer, H., Schmitt, J., Severinghaus, J.P., Spahni, R., Stocker, T.F., 2014. Isotopic constraints on marine and terrestrial N2O emissions during the last deglaciation. Nature 516, 234-+.

Seltzer, A.M., Buizert, C., Baggenstos, D., Brook, E.J., Ahn, J., Yang, J.W., Severinghaus, J.P., 2017. Does delta O-18 of O-2 record meridional shifts in tropical rainfall? Climate of the Past 13, 1323-1338.

Veres, D., Bazin, L., Landais, A., Kele, H.T.M., Lemieux-Dudon, B., Parrenin, F., Martinerie, P., Blayo, E., Blunier, T., Capron, E., Chappellaz, J., Rasmussen, S.O., Severi, M., Svensson, A., Vinther, B., Wolff, E.W., 2013. The Antarctic ice core chronology (AICC2012): an optimized multi-parameter and multi-site dating approach for the last 120 thousand years. Climate of the Past 9, 1733-1748.

---

## Author Comment (AC2) · 13 Nov 2018

**Response to Referee #2**

**1- SUMMARY AND GENERAL COMMENTS:**
The study by J. Menking and collaborators presents three new ice cores from the Taylor Glacier Blue ice area that they combine to provide the first "composite" ice core record from this location that covers the transition between Marine isotopic Stage (MIS) 5 an MIS 4 (~74 to 65 ka). The chronology for the air trapped in the ice is defined based on the analysis of the global atmospheric tracers CH4 and atmospheric d18O of O2 (d18Oatm) and their synchronisation with well-dated CH4 and d18Oatm records from other Antarctic ice cores. The ice age scale is defined mostly based on the ice dust content synchronisation, again with other well-dated Antarctic dust profiles. From these two ice and gas age scales, they infer the evolution of the age difference between ice and gas at the same depth – the so-called delta age – through this MIS5-MIS4 climatic transition. Substantial delta age changes are observed through time over this time interval i.e. with values from ~2000-3000 years at ~ 74 ka and approaching ~ 10 000 years at ~ 60 ka. The authors also provide a new evaluation of the delta age evolution throughout the same period in the Taylor Dome ice core (located south of the glacier), which suggests no significant delta age changes for this site. The authors attribute these contrasting delta age evolutions between the two sites to a steep accumulation gradient across Taylor Dome that intensified across the transition from MIS 5 to MIS 4.

This paper presents a study that will be of great interest for the ice core community and to the extended paleoclimate community. It is thus well within the scope of Climate of the Past. Overall the manuscript is well written and presents substantial new material and interesting interpretation of the results. However several aspects of the paper need improvements and clarifications and thus I believe that major revisions are needed before it can be considered for publication.

My first major comment is related to the fact that the authors interpret the differences in the delta age evolutions between the Taylor Glacier area and the Taylor Dome ice core site almost exclusively in term of a change in the accumulation gradient between the two areas. While this could be an acceptable interpretation, they absolutely need to build a much stronger case regarding why this is their favoured one (e.g. versus ice thinning) and thus provide a much more elaborated discussion of their new results. But also, they should discuss the other possible controlling factors; in particular, those are commonly identified as impacting the firnification processes e.g. the role of surface temperature vs accumulation rate vs ice impurity content have already been discussed over the past few years (e.g. Bréant et al. 2017, Capron et al. 2013; Hörhold et al. 2012). I believe that a summary of the current knowledge (and knowledge gaps) regarding the climate and environmental factors that impact changes in delta age would be useful. In particular, it would be of added value to further mention firn densification models that provide an alternative method to estimate delta age. At the moment the authors only acknowledge the Herron and Langway model (1980) although several other models building on this original work have been developed in the more recent years (e.g. Goujon et al. 2003) and more recent development in Bréant et al. 2017, dynamical version of Herron and Langway (1980) used in e.g. Buizert et al. 2015).

The role of surface temperature was discounted in our initial interpretation because the differences in delta age between Taylor Glacier and Taylor Dome are so large, but the accumulation sites are quite close to each other and likely to not differ in surface temperature history very much. Accumulation seems much more likely to vary between the two sites, particularly given the previous work by (Morse et al., 1998), cited in our manuscript, showing different layer thicknesses across the dome. This interpretation is consistent also with the notion that accumulation has a greater control on delta age than temperature does. We think ice impurity

content likely has a secondary effect compared to accumulation. We have a measure of impurity content in the particle count data and Ca concentrations. Particle count and Ca begin to rise at 7.5 m depth (moving up core), but delta age has already begun rising in non dusty ice at 11.5 m depth – so impurities do not seem to be driving delta age to first order.

We do agree that a summary of the factors controlling delta age would be appropriate, and in revision we will add text that includes the points made above.

The reviewer also mentions thinning. We believe the reviewer is suggesting that thinning due to flow from the dome to the sample site would somehow impact the age difference between gas and ice. Referee 1 made a similar point, to which we responded in detail. While thinning obviously could impact the depth difference between coeval points in the gas and ice phase, we do not see that it affects delta age because it does not disrupt the stratigraphic order of bubbles in relation to the ice matrix that encloses them. Our depth-age relationships are determined independently for the gas and ice phases, thus we make no assumption about accumulation to determine delta age. If we did assume accumulation rate to get delta age, and if we had assumed constant thinning for both Taylor Dome and Taylor Glacier, thinning could have been an issue.

We will cite other firn model studies, but all models support the general statements in the paper about the relationships between temperature, accumulation, and delta age.

My second major comment is related to the form of the paper. First I believe that some reorganizations of some sections are necessary and I detail this in the next section. Second, I think that the Figures 2, 3 and 4 need to be revised so that the readers are able to better visualized the different records that are being presented but also so they better support the results and the proposed interpretation. More details are provided in the next section of the review.

See detailed comments below where these issues arise.

Additional comments are also provided in the following and I would strongly advice the authors to consider them when preparing a revised version of their manuscript.

See detailed comments below.

2- SPECIFIC COMMENTS:
- Section 2 (Field site and analytical methods) is not always easy to follow, in particular regarding which type of measurements has been performed on which core and where (on site or in labs back in the USA). I would suggest the authors to propose a summary table in the revised manuscript that detail clearly this information.

We will add a table that details the metadata for all measurements made – i.e. which core, which measurement, at which institution, and in the field or lab.

- The authors propose to treat the three ice cores covering the MIS5-4 transition as a single ice core record (unified depth and age scales). While I agree with them that it is justified, I believe that they should provide additional details on how they line up the different records together (and possibly provide a specific figure?) and discuss the attached uncertainties that arise from proceeding as such on the resulting "composite" record.

The cores are not "aligned" in depth, per se. They were drilled adjacent to one another, so we assume that, e.g., 10.0 m depth in the 2014-2015 core is the same as 10.0 m depth in the 2015-

2016 core. There was no shifting or stretching the depth scales to make the records match better between different cores. The only problem that leads to errors in the depth scales is irregular angle breaks at the ends of individual blue ice drill cores that were not properly aligned in the field immediately after recovery. This could theoretically lead to depth offsets of no more than 20 cm between cores as most angle breaks are < 10 cm. Our view is that the effect of depth offsets is visible in the comparison of the discrete CH4 records from the 2014-2015 core versus the field CH4, where you see up to a 10 cm depth offset between records at DO 19. 10cm conservatively equates to 210 years on our depth scale where age changes the most with depth. The continuous CH4 measured at DRI versus in the field (same 2015-2016 core) actually exhibit larger offsets (up to 20 cm = 300 years on our age scale), likely from errors in the depth logging or again from angle breaks that cause depth offsets between sticks cut from the same core for field versus lab continuous flow analysis. Since this is the largest depth offset observed, we think this sufficiently estimates (and probably overestimates) the error due to depth offsets. Thus we propagated 20cm = 420 years error into our delta age calculations.

We will explain all of this more clearly in the text by adding a paragraph in the analytical methods section that elaborates on the depth uncertainties. We will also display the propagated error on the depth-age plots, not just the delta age (as in Figure 5 below).

Figure 5 – Age models for new Taylor Glacier 5/4 BID cores (A), Taylor Glacier delta age and $\delta^{15}$N-N$_2$ (B), and Taylor Dome revised age models (C), and Taylor Dome delta age and $\delta^{15}$N-N$_2$ (D). Red shading on Taylor Glacier gas age chronology and delta age indicates ice shallower than 4 m where surface cracks may affect the CH$_4$ age matching.

[Figure]

- Section 3.1 is hard to follow, the authors should consider restructuring it such as 1) they present how the ice age scale has been defined and then 2) as the gas age scale has been defined. Regarding the definition of the tie points based on the alignment of the dust record, I find that

some of them are quite ambiguous considering the number of spikes present in the TG records. For instance why would they assign the tie point at 73.6 ka to the spike at 12 m rather than the spike at 9 m? I believe that the authors have a good reason for doing so, however, it should be spelt out more explicitly. It is necessary that the figure be much enlarged to allow a detailed inspection of the records.

We appreciate Referee #2's suggestion to restructure section 3.1. We will reorganize the text such that the 'Age Models' section comes before Results and Discussion. In this section ice age and the gas age models are explained in separate sub-sections or paragraphs. We will also move the explanation of the revised Taylor Dome age scales, though this will require some additional text to explain why Taylor Dome is of interest here.

Regarding the tie points based on aligning the dust records – we feel these tie points are justified because they produce the best overall match between the Taylor Glacier dust and water isotope records with EDC. We explored a large number of alternate strategies, which did not perform as well. For example, the specific tie point questioned by the reviewer (12 m versus 9 m) is best justified with the d18Oice data. If the dust is matched at 9m, the correlation between Taylor Glacier and EDC d18Oice deteriorates substantially because of mismatches in the variability around AIM 19 and AIM 20. The uniqueness of the d18Oice and dust records together justifies the tie point.

We will justify our tie point choices more clearly in the Age Models section of the main text. We also eliminated the tie point to dust completely and instead chose 2 new tie points from the d18Oice record so that readers clearly see which variability we are matching instead of potentially ambiguous variations in nssCa. We think matching directly to d18Oice instead of using d18Oice as justification for a possibly more ambiguous nssCa match makes a stronger case for the age model in this section of the core.

- I do not think that the analytical uncertainties should be discussed after the determination of the age model. The authors should consider adding a brief description of each dataset after the analytical method descriptions and there, add details regarding their specificity and limitations.

We will move the discussion of analytical uncertainties to section 2 where the analytical methods are first introduced.

- It is a little strange that the presentation of the new measurements on the Taylor Dome ice core and the definition of the new age scale and for Taylor Dome are currently presented as part of the discussion. Why not instead presenting the new age model of Taylor Dome as an additional sub-section in the age model section that is currently only dedicated to the dating of the Taylor Glacier ice? And similarly for the new measurements, they should be also included in the analytical description section and information should be also added in the table I propose to add in the revised manuscript. Also, I think it would be very useful that more background information is provided regarding the Taylor Dome site, in particular regarding the previous age scales available for this time interval.

We understand this point, but ordered the text the way we did because there has been a lot of previous work on Taylor Dome. However, we are willing to reorganize the text as suggested. We will split the Age Models section into 3.1 Taylor Glacier Chronology, 3.2 Taylor Dome Revised Chronology. We will need to add some text in 3.2 explaining why Taylor Dome is of interest. At this point in the paper we can also add more background information about the Taylor Dome site with special attention to the previous age scale.

Included in this change will be the addition of metadata about the new Taylor Dome measurements in the new metadata table, and discussion of methods for Taylor Dome samples in the methods section.

**3- FIGURES**
- I appreciate the effort of the authors to show how they defined the different tie points to link between the Taylor Glacier records on a depth scale the dated reference records. However, it should be bigger to allow a closer inspection of the different records and where the tie points have been chosen.

We will make Figure 2 larger so that the tie point picks are more clearly visible. We think this will make the picks more justifiable with closer inspection.

- Figures 3 and 4 should appear much bigger. Also, to facilitate the comparison of delta age evolutions between Taylor Glacier and Taylor Dome, I suggest to remove the panels b from each figure and combine these panels b into a single and additional figure. They can be presented in parallel, making sure that the scale used for the delta age evolution is the same for both sites.

We will enlarge figures 3 and 4. We will reorganize so that the "b" panels are together in one separate figure for easier comparison (Figure 5 above).

Revised Figure 3:

[Figure]

Revised Figure 4:

[Figure]

**4- STYLISTIC, TYPOGRAPHICAL COMMENTS AND MINOR COMMENTS**

P2, L16: You should also mention the work that has been done in the Patriot Hills blue ice area e.g. Fogwill et al. (Scientific Reports 2017).

We will include the Patriot Hills work in our list of blue ice areas.

P2, L34: I find the expression "MIS 4 paleoarchive" to be an awkward formulation; I would suggest to reformulate the sentence e.g. "(2) the description of a new climatic record from Taylor Glacier across MIS 4".

We will rewrite the sentence to read, "(2) the description of a new climatic record from Taylor Glacier across MIS 4" on line 34.

P4, L1: "second exploratory core": this is a bit confusion to say "secondary" since the PICO core was also referred to as a "secondary exploratory core". It should be rephrased e.g. "During the same 2014-2015, another exploratory core was obtained directly : : :.".

We will rewrite the sentence to read, "… another exploratory core was obtained directly…" on line 1.

P4, L5: Again the numbering of the core is confusing (as in total, as far as I understand, four cores were drilled with only the last three having MIS5/4 transition ice). Hence it would be could to reformulate such as e.g. "In the 2015-2016, an additional core was drilled: : :".

We will make the listing of various cores more clear with a table.

P5, L26: The authors should be more specific in the title of the section e.g. "Determination of the ice age and gas age scales".

We will rename the title of the section to "Determination of the ice age and gas age scales" on line 26.

P6, L4: "minimal" please be more quantitative here and give a quantitative range at least.

We will provide a quantitative range on line 4.

P8, L11: Although you refer to the tables, the authors should also provide at least a quantitative range regarding the relative age uncertainties.

We provided a quantitative range for each tie point in the original manuscript, though we did not state the uncertainties clearly in the main text. While we propagated our estimated uncertainty to the delta age calculations to provide maximum and minimum delta age estimations, we did not clearly show the error propagated into the age models themselves. In the revised manuscript we will shade the uncertainty around the depth-age plots in Figure 5 (above). We will also state an age uncertainty clearly in the text. We also enlarged the estimated uncertainty range for tie points that are possibly more ambiguous, particularly the dust tie point at ~ 61.5 ka.

REFERENCES:

Capron et al. 2013. Glacial–interglacial dynamics of Antarctic firn columns: compari- son between simulations and ice core. Climate of the Past, 9, 983–999.

Breant et al. 2017. Modelling firn thickness evolution during the last deglaciation:

constraints on sensitivity to temperature and impurities. Climate of the Past, 833–853, 2017.

Buizert et al. 2015. The WAIS Divide deep ice core WD2014 chronology – Part 1: Methane synchronization (68–31 ka BP) and the gas age–ice age difference. Climate of the Past, 11, 153–173.

Goujon C. et al. 2003. Modeling the densification of polar firn including heat diffusion: Application to close-off characteristics and gas isotopic fractionation for Antarctica and Greenland sites, J. Geophys. Res., 108, 4792, doi:10.1029/2002JD003319, 2003.

Herron and Langway 1980. Firn densification-an empirical model, Journal of Glaciology, 25, 373-385, 1980.

Hörhold et al. 2012. On the impact of impurities on the densification of polar firn. EPSL, 325-326, 93-99.

Fogwill et al. 2017. Antarctic ice sheet discharge driven by atmosphere-ocean feedbacks at the Last Glacial Termination. Scientific Reports, DOI: 10.1038/srep39979.

Morse, D.L., Waddington, E.D., Steig, E.J., 1998. Ice age storm trajectories inferred from radar stratigraphy at Taylor Dome, Antarctica. Geophysical Research Letters 25, 3383-3386.

---

## Author Comment (AC3) · 13 Nov 2018

**Response to Referee #3**

The manuscript presents the initial multi-tracer dating of recent large size ice cores from Taylor Glacier (TG), covering a period of about 25 ka around the MIS 4/5 transition, as well as new data aiming at improving the gas chronology of the Taylor Dome (TD) ice core during the same period. Such characterization of a blue ice field providing large amounts of ancient ice is certainly of interest for the paleoclimate community and well within the scope of Climate of the Past. The results are discussed in terms of age difference between the gas and ice phases (delta age) and related varying accumulation rates. This interpretation involves some assumptions and simplifications that are not enough described in my view. For example, a number of age synchronization tie points appear ambiguous to me and the remaining discrepancies between records are not sufficiently commented. The inferred very low accumulations are likely to imply erosion periods, and the impacts of the ice-flow (thinning, hiatuses, possible folding etc.) should be better considered. Even if firn modeling with somewhat empirical models well outside the calibration range of their parameters is not compulsory, the physical processes controlling delta age and $d^{15}N$ fractionation should be better described.

Overall I think that major revisions are needed in order to better discuss the approximations made (e.g. ignored firn and ice physics), describe the consequences of alternative assumptions on ambiguous chronological tie points for multi-species consistency, age scales and delta age. I think that the paper should be more focused on an in depth discussion of the ice cores dating and dating issues, and less focused on somewhat spectacular but uncertain conclusions on delta age and accumulation. A number of suggestions are provided below.

We thank Referee #3 for helpful comments.

As discussed in the response to other reviewers, we are addressing the perceived ambiguity of tie point selection by justifying them with more extensive discussion in the text. We will enlarge the figures (particularly Figure 2, but also Figure 3 and Figure 4) so that it is easier to see why we chose to match variations the way we did. We will also describe why alternative tie point selections produce poorer matches with EDC records. A specific example of a possibly ambiguous tie point choice was brought to our attention by reviewer 2 and reviewer 4 – why assign the peak in dust at 73.6 ka to 12 m instead of 9 m? The reason is that if we assign the 9 m peak to 73.6 ka then the d18Oice is shifted such that the minimum between AIM 19 and AIM 20 no longer aligns with the EDC d18O record. The correlation between d18Oice EDC and d18Oice TG gets worse due to stretching the TG AIM 19 peak by several thousand years. This way we also would not align the nssCa peak at 73.6 ka (there is no nssCa variability in our record at 9 m).

A second possible ambiguity is the dust peak at 15 m. In our set of tie points we do not align this peak, so we tried two alternatives to align it to variations in EDC nssCa. If we align it with the EDC nssCa peak at ~77 ka (1) this stretches the d18Oice record out such that the signal no longer matches EDC d18Oice at AIM 20, (2) the nssCa variability in TG doesn't really match the variability seen at 77 ka in EDC, and (3) the delta age gets unreasonably high (we expect accumulation to be higher in stage 5 versus stage 4 due to warmer average temperatures and thus delta age to be relatively lower than during stage 4). We also explored aligning the 15 m dust peak with the EDC variability at ~73.6 ka, but again this causes a mismatch the d18Oice in EDC at AIM 19/20.

A third possible ambiguity is in the dust peaks between 0-1 m. We could align the large dust peak at 0.3 m to the nssCa peak in EDC at ~ 64 ka. This shifts other aligned dust peaks back in time - i.e. the peak at 1.25 m aligns with a very small dust peak at 65 ka and seems out of place, and the

three particle count peaks between 1-3 m depth do not have corresponding 3 laser dust peaks to align with. Instead one peak has to be skipped. We prefer to align the particle count peak at 0.3 m with the smaller EDC peak at 61.5 ka because we observe that background particle count appears to be decreasing toward shallower depths (see minima in the particle count record between peaks) similar to how EDC nssCa and laser dust decrease between from 64 ka to about 60 ka. However, we recognize this interpretation puts the 0.3 m nssCa peak at a place on the AICC 2012 age scale where EDC nss Ca has no corresponding peak (see figure below). We have contacted the original authors of the data in question, and no logs of contamination or processing errors exist for these depths in EDC. The existence of the laser dust peak without a corresponding nssCa peak is as of yet unexplained (Fischer, H. and Lambert, F. personal communication). It is possible the dust captured in EDC at 61.5 ka had very little Ca while the dust at TG did.

The nssCa mismatch we are describing above is shown in this figure (black arrows show particle count peak and corresponding laser dust peak (red) on AICC2012, the blue trace has a nssCa peak without a corresponding peak in EDC (brown)).

[Figure]

One plausible alternative for the 0.3 m tie point is to shift it to older ages, which causes a mismatch in the rest of the data and increases the delta age estimate by 2.5 ka.

Another plausible alternative is that the "stray" nssCa peak in the Taylor Glacier record is from local, wind-blown $Ca^{2+}$ dust and is not representative of a larger-scale Antarctic dust event. The peak occurs in the top 30 cm of the ice core where dust data have been rejected previously ((Baggenstos et al., 2018) rejected top 40 cm) due to contamination of vertical cracks by local wind-blown dust.

We will discuss completely our justification for tying the 0.3 m dust peak, but we will also emphasize that we do not interpret the age scale in the top 40 cm rigorously, similar to (Baggenstos et al., 2018).

Because we are discussing the shallow part of the core here, we think it is appropriate to inform the editor about a mistake we made in the presentation of data in the original manuscript. We cut off the top meter of the TG records in the original Figure 3. This is why there are tie points for ice as young as 61.5 ka but no data that young in the original Figure 3. We did this for the gas data

because there is clearly CH4 contamination up to 1200 ppb in the 0-1m section of our cores (which appeared in all measurements, no disagreement between DRI and field CH4). We suspect this is due to snow machine oil/ exhaust at the drill site.

We will include text that describes this issue in the revised manuscript. We do not, however, see any reason to reject the entire 0-1 m in the dust, particle count, or d18Oice records. Rather, wind blown dust contamination has only been observed in the top 40 cm at Taylor Glacier (Baggenstos et al., 2018).

We prefer to show all data in the revised manuscript and revised figures for completeness, but the ice records shallower than 40 cm and the gas records shallower than 4 m will not be interpreted rigorously. This will be described and justified clearly in the text.

Regarding interpretations of high values of delta age – we noted in our response to reviewer 2 that while differential ice thinning would affect the depth-age relationships, it would have no effect on delta age because thinning does not disrupt the stratigraphic relationship between ice and gas bubbles at depth. The reviewer also referred to hiatuses in accumulation. We think an accumulation hiatus is in line with (if an extreme example of) how we are currently interpreting the high delta age values – i.e. high delta ages correspond to low accumulation rates. We did not explicitly discuss what our records would look like if a complete cessation of accumulation occurred. We will include text that discusses how the records might look if accumulation hiatuses occurred. A hiatus, if it did occur, is most likely in the section 60-64 ka where CH4 is flat, d18Oatm variability is small, and full MIS 4 conditions are underway with extremely cold temperatures and low accumulation at the TG catchment. Because our record does contain the complete CO2 rise for the MIS 4/3 transition, we think there is good reason to believe there is no significant hiatus in bubble trapping. 60-64 ka on the gas age scale corresponds to 68-71 ka on the ice age scale, where there is still clear variability in our particle count and nssCa records. We think this is further proof that there is not a hiatus in accumulation. We will include these justifications in the text. Regarding folding, there is no evidence of folding in the records we developed, which would show up as reversals in gas and ice phase records as compared to known trends from other ice cores. We see no reversals in our gas records and ice phase records. For the sake of demonstrating our thinking - one might for example question whether the CO2 variability at AIM 19 is in fact two limbs of a fold with its center at the CO2 peak. Looking at CH4, d18Oatm, nssCa, and insoluble particle count tracers on depth axes rules out the possibility that the ice is folded because the records are not identical on both sides of the hypothesized fold axis. The same can be said even where the gas records are relatively flat – e.g. between 8.5-10.5 m depth when gas concentrations are relatively low, d18Oatm is relatively enriched, and there is little variability. Here there is also little variability in the nssCa and insoluble particle count records to resolve the problem. We note in this ambiguous section that the d18Oatm is steadily becoming more enriched and d15N is becoming steadily more depleted with no evidence that the trends reverse, as you would expect if the ice were folded there.

We will include similar discussion of hiatuses and folding in the revised text.

We will include more information in our discussion on the physical parameters controlling delta age (lines 15-22, page 8) and d15N (line 35-36, page 8). Specifically we will include discussion of secondary effects on firn evolution/ delta age, including impurity concentrations, wind stress (wind pumping), and surface temperature (here I am citing topics mentioned by other reviewers as well as referee 3). We will also state specifically that thinning can affect the depth-age relationship but not delta age because there seems to be some confusion about this.

p2 l34-35 and p3 l26-28: Missing MIS 4 and MIS 4/5 transition in previous TG records. The authors should provide references and introduce more the possibility of having different hiatuses in different TG ice cores. The ice flow in the area should be better illustrated, for example Figure 1 (a) could be further zoomed on the drill sites and some flow line directions could be provided.

The missing MIS 4 is explicitly discussed in (Baggenstos et al., 2017), which we will add as a reference for lines 34-35 on page 2 and lines 26-28 on page 3, with explanation.

It is unlikely that there are different accumulation hiatuses in the different ice cores presented in this work, if that is what the referee means. The 5/4 BID cores as well as the PICO auger exploratory core were drilled within ~ 1 m of one another and so must have traveled down glacier as a unit from the exact same accumulation area. Even the -380 m core on the Main Transect, which is < 1 km from the location where the 5/4 BID cores and PICO core were drilled, came from the same accumulation zone as the 5/4 BID cores (and all other TG stratigraphic units) without experiencing a hiatus or any sort of prolonged difference in accumulation relative to the 5/4 cores. Our point here is that the whole accumulation zone sourcing the TG ice archive would have experienced accumulation hiatuses at the same time, broadly speaking. We recognize it is possible for a glacier accumulation zone to have small-scale heterogeneity in accumulation rate either due to differences in precipitation rate or due to different magnitudes of wind scouring. We think this kind of variability would not affect the 5/4 BID cores or the PICO exploratory core because they were obtained so close to one another, but it is conceivable that prolonged heterogeneity in the accumulation zone caused discrepancies between the -380 m Main Transect core and the 5/4 cores. However we observe that the -380 m core d15N values are quite comparable to those measured in the 2015-2016 5/4 BID core. We interpret this as evidence that the two cores came from firn columns with similar characteristics, implying that the accumulation zone was more or less the same for both the Main Transect and the 5/4 drill site. In other words, the stratigraphy is continuous between the two drill sites. In case this is unclear or seems weak due to the arguable dating of the -380 m core, we are basically saying that the d15N/CH4, d15N/CO2, and d15N/d18Oatm ratios are the same in the -380m core as in the 5/4 BID core, supporting the conclusion that the stratigraphy on Taylor Glacier is continuous and that different hiatuses in accumulation, or even different accumulation zone sources altogether, were unlikely.

We will add general direction of flow lines to Figure 1a.

p2 l37: a reference should be provided for the previous TD chronology

The original chronology st9810 (Steig et al., 1998) was based on CH4 matching to GISP2 and inferring delta age to get the ice chronology. But the ice chronology was incorrect because it assumed accumulation could not be exceptionally low (and thus delta age could not be exceptionally high). The error was pointed out by aligning the TD Ca record to EDC (Mulvaney et al., 2000). The TD gas chronology was updated by synchronization to the Vostok GT4 timescale (Petit et al., 1999; Barnola et al., 1991) that extends back to ~ 68 ka (Indermuhle et al., 2000). A full chronology (gas and ice) was most recently updated by (Baggenstos et al., 2018) back to 60 ka.

We adopt tie points from (Baggenstos et al., 2018) where our age scales overlap. We will include the aforementioned references and a summary of the previous TD chronology in the revised manuscript.

p3 l14-16: a reference should be provided for these site characteristics

We will add references for the sublimation and flow rates in lines 14-16 on page 3 – (Kavanaugh et al., 2009a; Kavanaugh et al., 2009b).

p3 l25-28: a reference should be provided for the ice flow structure of the "main transect"

We will add references to lines 25-28 on page 3 for the vertical dip of layers on the Main Transect (Bauska et al., 2016; Schilt et al., 2014; Baggenstos et al., 2017; Petrenko et al., 2017; Petrenko et al., 2016).

p3 l30-31: the exact location of the "-380m" drill site (coordinates) should be provided. More site information could be provided (e.g. altitude, mean annual and summer temperatures etc.)

The mean altitude, mean annual, and mean summer temperatures are not different from any other site on Taylor Glacier discussed in the paper. It is a drill site on the Main Transect (Figure 1), 380 m from a flag that marks the center "0 m" on the Main Transect. We will add all of this specific information in the revision.

p5 l8-11 and p8 l1-4: the depth offsets, uncertainties and unification method between the different "TG 5/4" cores should be better described.

Referee 2 had a similar question and we repeat our response here. There is not a unification method, per se. Each core was drilled adjacent (within 1 m) to the original borehole drilled with the PICO auger. Each core has a depth scale determined by summing the lengths of individual, meter-long BID cores. We assume, for example, that 15.0 m in one core = 15.0 m in another core. The cores are not "aligned" in the sense that we did not stretch or alter the depth scales to match the data precisely. When you view all measurements on depth there are very small offsets between the records, indicating slight depth offsets, likely due to short angle breaks at core ends that effect the depth summation along the core. We conservatively estimated the effect of these offsets on our age model and propagated them through the delta age calculations (discussion of this begins at p7 line 36).

p5 l17: "The interpretations that follow do not depend on data taken from 0-4 m", and similar statement p7 l22. In Figure 2, the 3 TG CH4 data series are not consistent above 5m depth, and in Figure 3 the CO2 consistency with the composite in the upper part of the TG record mostly rely on the 2 upper points. What would be the consequence of matching the TG CO2 record below 4 or 5 m depth to the composite CO2 record instead of using the CH4 record which is nearly flat between ~4.5 and 7 m depth for multi-species consistency and delta age? In Table 2, two CH4 tie points and half of the ice phase tie points are located well above 4 m depth.

We see why Referee #3 would be suspicious about the data 5m and shallower – the CH4 records depart from one another substantially above 4m with smaller differences between 4-5m depth, and the CO2 appears to date too young relative to the composite data. But what Referee 3 says here is not entirely correct. The shallowest CO2 measurement is at 4m depth, where the CH4 differences are much smaller, and the CH4 rise evident in both datasets (associated with DO17) is one of the most robust features. The discrepancy in the CO2 depends highly on the tie point at 5.4 m – the "low point before DO 16/17" in table 1. We think this is the most robust gas tie point of the entire set. If we shifted this to younger ages, it would smear the CH4 rise out such that TG CH4 would lead EDML. The next tie point is at DO18. We did not choose other tie points from the CH4 record because the CH4 variability between DO18 and DO17 is minimal, thus any tie

points chosen there would be ambiguous. We could choose tie points deliberately from the CO2 record such that the slopes of the CO2 increases are more similar, but we refrained from doing this given that CO2 offsets between different ice cores are a known but relatively poorly understood phenomenon (Luthi et al., 2008). We addressed this in lines 13-16 on page 6. In fact, as an example, there are CO2 offsets between the TG CO2 and the composite record from (Bereiter et al., 2015) of even larger magnitude than at the 4/3 transition during the middle of stage 4 (Figure 3).

The consequence of matching the CO2 would be that the records would be more consistent (value-matched), and delta age would be lower by ~1.5 ka at the most. The uncertainty we estimated for the delta age calculation is already larger than this.

We would like to stress that the parameters in the ice phase (i.e. d18Oice and dust) are only affected by the surface cracks in the top 40 cm, not the entire top 4 m. This is stated in the Table 1 caption, but we will state it more clearly in the main text too. So the ice phase tie points are not an issue except potentially in the top 40 cm.

Referee #3 is correct that two tie points for the gases are chosen above 4m, which is why we shaded those tie points gray in Table 1. Our intention was that those points be interpreted cautiously. The CH4 record shows variability that looks very much like the CH4 variability associated with DO 16/17, hence the temptation to choose tie points and extend the gas chronology to depths shallower than 4m. But the mismatch in CH4 between the DRI and field data sets leaves us unable to reject the possibility that both data sets are wrong < 4m. This wouldn't change the conclusions of the paper because delta age begins to rise at 11.5 m depth in our core, with maximum delta age occurring at ~ 5.5 m. We would like to reemphasize that we do not interpret the gas data shallower than 4 m rigorously and that those data do not inform our interpretations of the high delta age values.

p5 l27-30: In Figure 2, the TG CH4 records look a lot smoother than the EDML record. The dissimilarity of the two signals limits the possibilities of unambiguously synchronizing them. This could be due to different processes such as analytical smoothing (Stowasser et al., 2012), longer gas trapping duration in firn at very low accumulation rates (Spahni et al., 2003; Köhler et al., 2011; Fourteau et al., 2017), gas diffusion through ice (Bereiter et al., 2014 and references therein). This should be discussed, possibly smoothing the EDML record to try to simulate the TG record, comparing with the lower accumulation EDC record etc.

We think analytical noise in the EDML record is the main reason for the dissimilarity between the EDML and TG CH4. We would prefer to plot the error bars on the EDML data, which visually help the reader see the smooth atmospheric signal, rather than smooth the data set directly. EDC CH4 looks quite similar in resolution and smoothness to EDML. The relative amplitudes of abrupt CH4 features can be an indication of relative smoothing. EDC, EDML, and TG all have the same magnitude CH4 feature at DO 19. At DO 18 EDML CH4 is higher, followed by EDC CH4, followed by TG CH4. The CH4 rise at DO 16/17 (near the MIS 4/3 transition) is largest in EDML, only slightly smaller in TG, and lowest in EDC. Using this as an indication of smoothing, then the effect in TG is largest at DO 18 and negligible at other times in the record.

We don't think the CFA system is smoothing beyond what the firn has already done to the gas record. The main justification for this is that the discrete CH4 measured in the lab (green dots at DO19 in Fig 2) and CFA CH4 (purple and red lines in Fig 2) agree well.

We will include discussion of smoothing in the text including justification of why we don't think

smoothing effects are significantly impacting our tie point choices.

p5 l34-35: I did not understand why the $_{18}O_{atm}$ record is tied to NGRIP only: a North Hemisphere discontinuous record covering only parts of the studied period. Could other data also be used? (e.g. Petit et al., 1999; Kawamura et al., 2007; Buiron et al., 2011)

The TALDICE and Dome Fuji datasets are unpublished and/or unavailable publicly) to our knowledge, though they appear in figures in the referee's citations. Both are low resolution through the time period of interest, and Vostok d18Oatm is also quite low resolution. To our knowledge the Dome Fuji DFO 2006 age scale is not synchronized to AICC 2012, though Vostok and TALDICE are. We do not think synchronizing to any of the three records helps eliminate ambiguity that CH4 doesn't already solve. Where d18Oatm is helpful is syncing TG to NGRIP in the older part of the gas record where CH4 variability is comparatively smaller but d18Oatm variability is large. Also worth noting here is that NGRIP d18Oatm is relatively high resolution across the 71-76 ka section. The match to NGRIP is further justified by the close agreement with EDML d18Oatm, which we plot in the revised Figure 3 (below).

We will justify the synchronization to NGRIP in the text.

Revised Figure 3:

[Figure]

p5 l38: some tie points look ambiguous to me and the tie points assignment should be further discussed. For example, the EDC and TG $\delta^{18}O_{ice}$ records look quite different in Figure 2, thus the $\delta^{18}O_{ice}$ tie point does not look robust to me. On the dust plot in Figure 2, I do not understand why the small EDC peak at 75.75 ka was tied to the TG particles peak at ~12m rather than the one at ~9m depth.

We refer back to our response at the beginning of this document following the general comments.

The referee here likely made a typo because our 75.75 ka age is aligned with the d18Oice peak at 16.62 m. Thus we assume the referee means the 12.05 m dust tie point that we aligned with 73.58 ka (now updated to 12.20 m and 73.62 ka). We specifically addressed this tie point in the response above, as well as two other ambiguous tie points.

We will provide further justification of our tie point selections in the text as already described.

p6 l9-27: Due to the dissimilarities between the records in Figure 2, I believe that it is impossible to unambiguously assign the tie points. Thus I doubt that the choices were made without taking into account the constraints discussed in this section. An overall discussion of the constraints, what led to the current best guess dating and how other assumptions could be (or not) discarded would be most useful.

We agree and will include more discussion of the rationale that led to our tie point choices. The way the manuscript is written now, it sounds like we picked CH4, d18Oatm, particle count, and d18Oice tie points and were happy to find that the $CO_2$ and nssCa also looked good. In reality the referee is correct that there was some iterative feedback from the $CO_2$ and nssCa records even though we didn't actually select any depth-age tie points from those records.

We will rewrite the text so that it more accurately reflects how we reasoned through the tie point choices, especially now that the tie point choices have been revised (described in the summary document).

p6 l31-32 and Figure 3: I do not understand how the CH4 record from the "-380m" core could be unambiguously tied to AICC2012. On the other hand the $CO_2$ records seem easier to match and matched. The overall dating constraints should be better described.

We will rewrite the text to explain in more detail how we aligned the -380 m core to AICC 2012, including presenting the tie points in a table and discussing our tie point choices in the text. The main revision here is that we will deemphasize the -380 m core dating, presenting our tie points as a plausible chronology, and explaining more clearly why we think it is robust that the -380 m core is roughly late MIS 4 and MIS 4/3 age.

p6 l31 - p7 l14: I did not understand this discussion of the differences between the TG records. The dating of the "-380m" core is presented in one line and the $CO_2$ mismatch with "TG 5/4" not discussed, nor the d18Oatm mismatch with NGRIP at ~66 ka. The lack of information on flow line directions make the direct comparison between TG records difficult to understand, and few references are provided. I suggest to focus more this section on gas scales consistency between the "-380m" and "TG 5/4" cores, and how the $CO_2$ mismatch between the two TG cores in the 60-64ka age range could be explained. Is the ice phase of the "-380m" ice core also dated? Are large _age values also inferred?

We will address more completely the dating of the -380 m core in the text as well as include a table with tie points. We will also move the data to a separate figure where we compare the -380 m gas data with those from Taylor Glacier 5/4 BID cores as well as the reference records on AICC 2012. The $CO_2$ mismatch with TG 5/4 was not discussed in the text, but we will address it explicitly. The d18Oatm mismatch with NGRIP was also not discussed.

We think the -380 m core implies that there is stratigraphic continuity between the Main Transect and the drill site of the new MIS 5/4 BID cores. We think the exact dating of the -380 m core is unimportant; rather the important part is that the gases appear to be late stage MIS 4, the d15N is

similarly low, and the age-depth relationship is similarly steep. This supports the idea that different accumulation zones are not sourcing the Taylor Glacier blue ice area at different times. Instead, Taylor Glacier ice has likely come from the same deposition zone throughout the last ice age. We intend to deemphasize the exact dating of the -380 m gas age scale and instead will simply argue that the methane and CO2 rises and the d18Oatm depletion are roughly what we would expect if the gas age was ~ last MIS 4 and MIS 4/3, and that the d15N is similarly low in that core, implying that there is continuity between the Main Transect records and the new MIS 5/4 records.

Unfortunately there is no ice phase data for -380 m, so we cannot infer delta age.

p7 l8-9: As this paragraph comes just after the section comparing the "-380m" and aggregated "TG 5/4" cores, readers may wonder which one is the new ice core.

We will clarify in the text that the "new" ice core is the TG 5/4 core.

p7 l8-13 and p9 l25-31: providing and discussing plots of annual layer thicknesses (based on depth - ice age, depth - gas age relationships at TG and TD) would help understanding the interpretations related to accumulation and thinning variations.

We calculated annual layer thickness, but we do not think it adds any information that is not already visible in the depth-age plots. The annual layer thickness is smallest where the age changes the most with depth.

p7 l24 - p8 l12: This discussion of uncertainties should appear earlier in the article and be more detailed (see also above comments on p5 l27-30, p5 l38, p6 l9-27).

Other referees also suggested this. We will move the uncertainty discussion to the Field Site and Analytical Methods section. We will more thoroughly discuss the uncertainties involved.

p8 l1-4: this is not consistent with p5 l11. Due to the strongly varying depth-age gradients on Figure 3 (b), the overall largest age bias related to depth offset/uncertainty should be mentioned.

Correct, we will update the text on p8 l1-4 to say 20 cm depth uncertainty instead of 10 cm.

p8 l5-8: the smoothing due to gas trapping duration most likely dominates the diffusive smoothing in the open pores of the firn. It is accumulation rate dependent (e.g. Spahni et al., 2003; Köhler et al., 2011; Fourteau et al., 2017) and thus likely different at EDML and TG. In Figure 2, the TG CH4 record looks much smoother than the EDML record. It would thus be interesting to discuss the gas trapping duration consistent with the firn sinking speed due to the estimated accumulation rates (time needed by the firn to sink by a few meters).

We agree that the smoothing is probably somewhat different between EDML and TG, but not so different that it affects our tie point choices or age model significantly. See our comment above about smoothing in EDML, EDC, and TG, particularly the part about the amplitude of CH4 variability at DO 18. The smoothness of the TG CH4 record versus EDML is due less to the gas trapping process and more to (1) the different analytical methods employed – continuous measurements in TG (and thus some degree of smoothing in TG, though small compared to smoothing in firn), versus discrete measurements in EDML, and (2) higher analytical noise in EDML (the TG continuous CH4 is within the EDML CH4 error bars at all parts of the records).

We agree it would be interesting to estimate gas-trapping duration, but we do not have a robust estimate of accumulation rate given that the d15N and delta ages are well outside of the calibration range of firn models. We might estimate accumulation rate given the depth-age plots if we knew the thinning function, but we do not know the thinning function and think it is unwise to trust fundamental thinning approximations given the archive's unconventional path to the drill site.

We will more clearly demonstrate the analytical noise in EDML by plotting the error bars on Figures 2 and 3.

p8 l16-18 and l35-36: a much more in depth presentation of firn processes influencing delta age, _depth and the physics of _15N should be provided. The consistency between a very large _age and a very shallow firn (_15N indication) should be commented.

In response to this as well as other referees' comments, we will provide a more in-depth summary of the processes influencing delta age as well as the fractionation of 15N in the firn column. We will more clearly elucidate the correlation between large delta age and shallow firn. We will also discuss secondary effects of wind pumping in the convective zone and impurity content on grain metamorphosis.

p8 l24-30: the example of the successive datings of the Taylor Dome ice core, well discussed in Baggenstos et al. (2018) could be used as a base for a more realistic uncertainty discussion.
We are unsure what the reviewer is suggesting here. If the reviewer is referring to the evolution of the Taylor Dome ice core chronology, which was described at length in (Baggenstos et al., 2018), then it is unclear to us how this would be the basis for the discussion of the uncertainty on our delta age calculations discussed on p8 l24-20. Our uncertainty (for both Taylor Glacier and Taylor Dome cores) is based on independently estimating the uncertainty of individual tie points and interpolating the max/min possible chronologies. The evolution of the Taylor Dome timescale was the result of errors in determining the delta age in the Taylor Dome timescale during the LGM. The chronology was revised by dust synchronization (Mulvaney et al., 2000) to obtain a correct ice age scale, and later refined and extended further back in time by (Baggenstos et al., 2018). The successive datings of the Taylor Dome core are useful in understanding the history of the timescale, but they do not provide much useful information about how to estimate more realistic uncertainty for our time period (57-77 ka).

We will add discussion of the history/ evolution of the Taylor Dome chronology insofar as it puts our work in context, but we will refer to the discussion in (Baggenstos et al., 2018) in lieu of re-summarizing everything.

p8 l35 - p9 l4: the fact that the physics of _15N (thermal and convection effects) is much more complicated than a pure gravitational effect can't be ignored (e.g. Severinghaus et al., 2001; Severinghaus et al., 2010). The very low _15N values measured in TG ice suggest that either the firn is very thin (an estimate should be provided) or nongravitational effects are important.

We will provide an estimate of firn thickness in the text. Based on gravitational effects alone the firn thickness is estimated ~ 15 m, though the height of the convective zone is a major uncertainty in this. A deep convective zone would drive d15N to lower values despite a thicker total firn.

p9 l6-19: the new Taylor Dome age scales presentation repeats methodological information already provided for TG cores but does not discuss the remaining inconsistencies between records and ambiguous tie points. A more in depth discussion of the Taylor Dome age scales should be

provided.

We will clean up the text with respect to repeated methodological information. We will discuss Taylor Dome dating in more detail including rationale for the tie points we chose. We will discuss in more detail the inconsistencies between the records.

p9 l21-31: this section is unclear to me. If TG and TD ice cores have strongly different _age in the study period (assuming that the tie points sufficiently constrain the age difference between the gases and ice in a single ice sample), TD can't be the origin site of TG ice even considering differential thinning.

The TG accumulation site is to the north of the TD ice core site, which we showed in Figure 1 and also stated in line 33 page 9. The point we are trying to argue in the paper is that over a small distance, accumulation varied significantly. At the LGM this trend is reversed. This is the interesting implication of the delta age histories.

We will describe this better in the text so that the point comes across more clearly.

p9 l2-4 and p9 l33 - p10 l26: no accumulation values were derived from the Taylor Glacier record and the discussion is focused on different time periods (present and LGM), thus it could be shortened.

We will shorten this part of the text to explain what we mean more succinctly. We are also awaiting the editor's guidance concerning whether providing accumulation rate estimates (given the very large uncertainties) is necessary.

**Technical corrections**

p8 l18-20: smaller _age values were obtained at very high accumulation rate sites such as DE08-2 (40 years, Etheridge et al., 1996)

We will include DE08-2 as an example of very small delta age.

p11 l24-25: twice "spanning the MIS 5/4 transition"

Here we are referencing the gas age scale separately from the ice age scale. We will rewrite this to seem less redundant.

p13 l15: Baggenstos, 2015 (PhD) a web link could be provided.

We will provide the link https://escholarship.org/content/qt9wn8789k/qt9wn8789k.pdf to the electronic version of the thesis.

p13 l23 and in article text: Update reference to Baggenstos et al. (2018), now available as a preprint.

We will update the reference to the accepted version of the manuscript.

p15 l49: suppress QUATERNARY

We will change QUATERNARY to Quaternary.

p16 l56: uppercase/lowercase issue Figure 2, dust panel: some grey lines are not consistent with the tie points in Table 2 (chronology inversions in some grey lines) Figure 3: the top part of the TG particles count record, including the tie point at 0.31 m depth, is not shown.

We are unsure what the uppercase/lowercase issue is that referee 3 refers to here. The chronology does not actually invert in the dust panel, though we see where the referee is talking about – it appears to invert in the gray lines where the dust begins to rise. We will expand Figure 2 so generally it is easier to see/ read. This should help readers not only understand why we chose tie points, but also make it clear that the chronology is not inverted.

**References not cited in the manuscript**
Bereiter, B., Fischer, H., Schwander, J., and Stocker, T. F.: Diffusive equilibration of $N_2$, $O_2$ and $CO_2$ mixing ratios in a 1.5-million-years-old ice core, The Cryosphere, 8, 245-256, https://doi.org/10.5194/tc-8-245-2014, 2014.

Buiron, D., Chappellaz, J., Stenni, B., Frezzotti, M., Baumgartner, M., Capron, E., Landais, A., Lemieux-Dudon, B., Masson-Delmotte, V., Montagnat, M., Parrenin, F., and Schilt, A.: TALDICE-1 age scale of the Talos Dome deep ice core, East Antarctica, Clim. Past, 7, 1-16, https://doi.org/10.5194/cp-7-1-2011, 2011.

Etheridge, D. M., L. P. Steele, R. L. Langenfelds, R. J. Francey, J. M. Barnola, and V. I. Morgan, Natural and anthropogenic changes in atmospheric $CO_2$ over the last 1000 years from air in Antarctic ice and firn, J. Geophys. Res., 101, 4115–4128, 1996.

Fourteau, K., Faïn, X., Martinerie, P., Landais, A., Ekaykin, A. A., Lipenkov, V. Ya., and Chappellaz, J.: Analytical constraints on layered gas trapping and smoothing of atmospheric variability in ice under low-accumulation conditions, Clim. Past, 13, 1815- 1830, https://doi.org/10.5194/cp-13-1815-2017, 2017.

Kawamura, K., Parrenin, F., Lisiecki, L., Uemura, R., Vimeux, F., Severinghaus, J.P., Hutterli, M.A., Nakazawa, T., Aoki, S., Jouzel, J., Raymo, M.E., Matsumoto, K., Nakata, H., Motoyama, H., Fujita, S., Goto-Azuma, K., Fujii, Y., Watanabe, O., 2007. Northern Hemisphere forcing of climatic cycles in Antarctica over the past 360,000 years. Nature 448, 912-916. https://doi.org/10.1038/nature06015.

Köhler, P., Knorr, G., Buiron, D., Lourantou, A., and Chappellaz, J.: Abrupt rise in atmospheric $CO_2$ at the onset of the Bølling/Allerød: in-situ ice core data versus true atmospheric signal, Clim. Past, 7, 473–486, https://doi.org/10.5194/cp-7-473-2011, 2011.

Petit, J.-R., Jouzel, J., Raynaud, D., Barkov, N.I., Barnola, J.-M., Basile, I., Bender, M.L., Chappellaz, J., Davis, M., Delaygue, G., Delmotte, M., Kotlyakov, V.M., Lorius, C., Pepin, L., Ritz, C., Saltzman, E., Stievenard, M., 1999. Climate and atmospheric history of the past 420,000 years from the Vostok ice core, Antarctica. Nature 399, 429-436.

Severinghaus, J.P., Grachev, A., Battle, M., 2001. Thermal fractionation of air in polar firn by seasonal temperature gradients. Geochem. Geophys, Geosys. 2 2000GC000146.

Severinghaus, J. P., Albert, M. R., Courville, Z. R., Fahnestock, M. A., Kawamura, K., Montzka, S. A., Mühle, J., Scambos, T. A., Shields, E., Shuman, C. A., Suwa, M., Tans, P., and Weiss, R. F.: Deep air convection in the firn at a zero-accumulation site, central Antarctica, Earth Planet.

Sci. Lett., 293, 359–367, 2010.

Spahni, R., Schwander, J., Flückiger, J., Stauffer, B., Chappellaz, J., and Raynaud, D.: The attenuation of fast atmospheric $CH_4$ variations recorded in polar ice cores, Geophys. Res. Lett., 30, 25-1–25-4, https://doi.org/10.1029/2003gl017093, 2003.

Stowasser, C., Buizert, C., Gkinis, V., Chappellaz, J., Schüpbach, S., Bigler, M., Faïn, X., Sperlich, P., Baumgartner, M., Schilt, A., and Blunier, T.: Continuous measurements of methane mixing ratios from ice cores, Atmos. Meas. Tech., 5, 999–1013, doi:10.5194/amt-5-999-2012, 2012.

Baggenstos, D., Bauska, T. K., Severinghaus, J. P., Lee, J. E., Schaefer, H., Buizert, C., Brook, E. J., Shackleton, S., and Petrenko, V. V.: Atmospheric gas records from Taylor Glacier, Antarctica, reveal ancient ice with ages spanning the entire last glacial cycle, Climate of the Past, 13, 943-958, 10.5194/cp-13-943-2017, 2017.

Baggenstos, D., Severinghaus, J. P., Mulvaney, R., McConnell, J. R., Sigl, M., Maselli, O., Petit, J. R., Grente, B., and Steig, E. J.: A Horizontal Ice Core From Taylor Glacier, Its Implications for Antarctic Climate History, and an Improved Taylor Dome Ice Core Time Scale, Paleoceanogr. Paleoclimatology, 33, 778-794, 10.1029/2017pa003297, 2018.

Barnola, J. M., Pimienta, P., Raynaud, D., and Korotkevich, Y. S.: CO2-CLIMATE RELATIONSHIP AS DEDUCED FROM THE VOSTOK ICE CORE - A REEXAMINATION BASED ON NEW MEASUREMENTS AND ON A REEVALUATION OF THE AIR DATING, Tellus Ser. B-Chem. Phys. Meteorol., 43, 83-90, 10.1034/j.1600-0889.1991.t01-1-00002.x, 1991.

Bauska, T., Baggenstos, D., Brook, E. J., Mix, A. C., Marcott, S. A., Petrenko, V. V., Schaefer, H., Severinghaus, J. P., and Lee, J. E.: Carbon isotopes characterize rapid changes in atmospheric carbon dioxide during the last deglaciation, Proceedings of the National Academy of Sciences of the United States of America, 113, 3465-3470, 10.1073/pnas.1513868113, 2016.

Bereiter, B., Eggleston, S., Schmitt, J., Nehrbass-Ahles, C., Stocker, T. F., Fischer, H., Kipfstuhl, S., and Chappellaz, J.: Revision of the EPICA Dome C CO2 record from 800 to 600kyr before present, Geophysical Research Letters, 42, 542-549, 10.1002/2014gl061957, 2015.

Indermuhle, A., Monnin, E., Stauffer, B., Stocker, T. F., and Wahlen, M.: Atmospheric CO2 concentration from 60 to 20 kyr BP from the Taylor Dome ice core, Antarctica, Geophysical Research Letters, 27, 735-738, 10.1029/1999gl010960, 2000.

Kavanaugh, J. L., Cuffey, K. M., Morse, D. L., Bliss, A. K., and Aciego, S. M.: Dynamics and mass balance of Taylor Glacier, Antarctica: 3. State of mass balance, J. Geophys. Res.-Earth Surf., 114, 7, 10.1029/2009jf001331, 2009a.

Kavanaugh, J. L., Cuffey, K. M., Morse, D. L., Conway, H., and Rignot, E.: Dynamics and mass balance of Taylor Glacier, Antarctica: 1. Geometry and surface velocities, J. Geophys. Res.-Earth Surf., 114, 15, 10.1029/2009jf001309, 2009b.

Luthi, D., Le Floch, M., Bereiter, B., Blunier, T., Barnola, J. M., Siegenthaler, U., Raynaud, D., Jouzel, J., Fischer, H., Kawamura, K., and Stocker, T. F.: High-resolution carbon dioxide

concentration record 650,000-800,000 years before present, Nature, 453, 379-382, 10.1038/nature06949, 2008.

Mulvaney, R., Rothlisberger, R., Wolff, E. W., Sommer, S., Schwander, J., Hutteli, M. A., and Jouzel, J.: The transition from the last glacial period in inland and near-coastal Antarctica, Geophysical Research Letters, 27, 2673-2676, 10.1029/1999gl011254, 2000.

Petit, J. R., Jouzel, J., Raynaud, D., Barkov, N. I., Barnola, J. M., Basile, I., Bender, M., Chappellaz, J., Davis, M., Delaygue, G., Delmotte, M., Kotlyakov, V. M., Legrand, M., Lipenkov, V. Y., Lorius, C., Pepin, L., Ritz, C., Saltzman, E., and Stievenard, M.: Climate and atmospheric history of the past 420,000 years from the Vostok ice core, Antarctica, Nature, 399, 429-436, 10.1038/20859, 1999.

Petrenko, V. V., Severinghaus, J. P., Schaefer, H., Smith, A. M., Kuhl, T., Baggenstos, D., Hua, Q., Brook, E. J., Rose, P., Kulin, R., Bauska, T., Harth, C., Buizert, C., Orsi, A., Emanuele, G., Lee, J. E., Brailsford, G., Keeling, R., and Weiss, R. F.: Measurements of C-14 in ancient ice from Taylor Glacier, Antarctica constrain in situ cosmogenic (CH4)-C-14 and (CO)-C-14 production rates, Geochim. Cosmochim. Acta, 177, 62-77, 10.1016/j.gca.2016.01.004, 2016.

Petrenko, V. V., Mith, A. M. S., Chaefer, H. S., Riedel, K., Brook, E., Baggenstos, D., Harth, C., Hua, Q., Buizert, C., Schilt, A., Fain, X., Mitchell, L., Bauska, T., Orsi, A., Weiss, R. F., and Everinghaus, J. P. S.: Minimal geological methane emissions during the Younger Dryas-Preboreal abrupt warming event, Nature, 548, 443-446, 10.1038/nature23316, 2017.

Schilt, A., Brook, E. J., Bauska, T. K., Baggenstos, D., Fischer, H., Joos, F., Petrenko, V. V., Schaefer, H., Schmitt, J., Severinghaus, J. P., Spahni, R., and Stocker, T. F.: Isotopic constraints on marine and terrestrial N2O emissions during the last deglaciation, Nature, 516, 234-+, 10.1038/nature13971, 2014.

Steig, E. J., Brook, E. J., White, J. W. C., Sucher, C. M., Bender, M. L., Lehman, S. J., Morse, D. L., Waddington, E. D., and Clow, G. D.: Synchronous climate changes in Antarctica and the North Atlantic, Science, 282, 92-95, 10.1126/science.282.5386.92, 1998.

---

## Author Comment (AC4) · 13 Nov 2018

**Response to Referee #4**

**General comments:** The authors collected and analyzed a set of new ice cores from the Taylor Glacier blue ice area covering the MIS 5 to 4 transition and whole MIS4, and present a suite of data (d18Oice, dust, Ca ion, d18Oatm, CH4, CO2, d15N2). Through age synchronization of gas and ice with other dated ice cores, they find extremely large delta age (ice age - gas age difference), which suggests much reduced accumulation rate at the snow accumulation area for the analyzed core and, by comparing the results with the Taylor Dome ice core data with their revised chronology, give climatic implications of the accumulation contrasts between the Taylor Glacier accumulation area and Taylor Dome.

Regional reconstructions of glaciological and climatological conditions in the glacial period in Antarctica are important for better understanding of the climate system in the Antarctic and its relation with wider areas, and thus the topic of this manuscript is well suited for the Climate of the Past, and the data presented in general seems to be of high quality. First I would like to respect and congratulate the authors for finding the ice from entire MIS4 after the years of fieldworks and high-quality investigations.

I review it mainly in terms of whether the ages, delta age and resulting accumulation rate reduction are reasonably estimated, because they are the basis for the climatic interpretation and conclusions, and also because they have the highest scientific value in this study in my opinion. In doing so I find that the manuscript needs a major revision to make much stronger cases for the extremely increased delta age and reduced accumulation rate (including its timing) at the Taylor Glacier accumulation area, which in turn are based on age synchronization and interpretation of the resulting delta age. In particular, I find it difficult to evaluate the robustness of their choice of the age tie points for some cases from the given text and figures/tables. There are also several tie points, which I did not understand how they could match with the existing ice core records. The authors made poor use of the data (especially d15N) for the discussion of the accumulation rate. While I agree with the authors that the delta age increased and accumulation rate probably decreased in MIS 4, it should be based on much more rigorous considerations. Also, some parts of the manuscript need to be reorganized to better present the field works, ice core samples, measurements and methods, results and discussion. I strongly encourage the authors to improve the article by deeper analyses, interpretation and better presentation of their excellent data.

We thank referee 4 for helpful comments. In general we will add stronger justification for the tie points that we chose, including adding text that explains our reasoning and improving the figures to facilitate the reader being able to see clearly why we chose the tie points the way we did. We will strengthen our discussion of the low accumulation rate interpretation and include a conservative estimate (~ 0.5 mm/yr – 2 cm/yr ice equivalent). We still think it is not possible to make robust, quantitative estimates of accumulation rate from firn models given that the models are not built or calibrated to describe firn columns where delta age is this high. Nevertheless for the purpose of strengthening our claims, we will (1) discuss in more detail the controls on delta age, (2) discuss the controls on d15N including more detailed discussion of why d15N is low at Taylor Glacier, (3) provide stronger justification for why firn models are not appropriate to estimate accumulation rates at these extremes, and (4) discuss the Megadunes, Antarctica site as a point of comparison to the Taylor Glacier accumulation zone. We will also reorganize the text following comments from referee 4 as well as referees 1-3. Please see specific answers to the referee's comments below for more details.

**Specific comments:**

**Abstract:** Add description that there are different ice cores, and that how the delta age was estimated ("Dating the ice and air bubbles" is too short even for the abstract). Similarly, "A revised chronology for the Taylor Dome ice core" needs some more explanation. Also, give numbers and error ranges for delta age and accumulation rate ("very low accumulation" is too vague; later in the text it is stated as virtually zero accumulation rate).

We will change "A new ice core" on p1 l24 to "New ice cores…" We will change "Dating the ice and air bubbles in the new ice core" on p1 l26 to read "We determine chronologies for the ice and air bubbles in the new ice cores by visually matching variations in gas and ice phase tracers to preexisting ice core records. The chronologies reveal…"

We stated in the text that we refrained from estimating accumulation rate quantitatively because we recognized that our large delta age value would require accumulation rates that are well below the empirical calibration range of the Herron-Langway firn densification model (the lowest accumulation site in the Herron & Langway paper is Vostok at 2.2 cm/ yr) (Herron and Langway, 1980). To our knowledge there is not a more appropriate model that accurately predicts firn densification under conditions of extremely low accumulation. We maintain the view that extrapolating beyond the empirical range of firn densification models may lead to errors that cast any determined accumulation rate into considerable doubt. Nevertheless we proceed with caution to determine a conservative maximum accumulation rate for the Taylor Glacier accumulation zone given delta age = 10 kyr as determined from our new ice core records. We used Herron and Langway to solve a matrix of density profiles for different temperatures (-20 to -70C) and accumulation rates (0.001 to 0.3 m/yr water equivalent). Assuming the density of snow is 0.36 g/mL and the close-off density is 0.83 g/mL we computed the age of the firn (the delta age) at the close-off depth using Herron and Langway's equation 11 (Herron and Langway, 1980). We then computed the $d15N$ due to gravitational enrichment for a matrix of diffusive zone heights using the barometric equation (Craig et al., 1988). A contour plot of $d15N$ and delta age on temperature and accumulation axes allowed us to examine the range of temperatures and accumulation rates expected given our independently determined delta age and $d15N$. For a site like Taylor Dome (likely between -40C to -50C at MIS 4), accumulation rates < 5 mm/yr water equivalent are needed to get delta age near 10 kyr and $d15N$ = 0.08 ‰, like we see at Taylor Glacier. We understand there are uncertainties in this estimate: (1) we did not correct for the thermal fractionation of $d15N$, though we expect it to be small on the order of < 0.01‰, (2) we do not know the height of the convective zone, which could be quite deep given the very low values of $d15N$ at Taylor Glacier. Even if we assume a deep convective zone of 25 m, similar to Megadunes site in Antarctica where the firn receives ~ 0 cm/yr accumulation and is highly cracked (Severinghaus et al., 2010), accumulation rates at the probable Taylor Glacier deposition site must be between 1.0-1.5 cm/yr to get $d15N$ below 0.1‰. We reassert that the uncertainties in these estimates are quite high, but we will provide the quantitative estimate of accumulation rate as well text that cautions readers to not interpret the numbers strictly. Our best estimate is that the accumulation rate must be between ~ 0.5 mm-2 cm/yr, less than but possibly similar to the accumulation rate at Vostok, the lowest accumulation site used to calibrate the firn model.

**Introduction:**

P2, L30: "glacial inception" is used differently (it is often used for MIS5e to 5d transition), so perhaps replace it with "major sea level fall" or "major ice sheet growth in the Northern Hemisphere".

We will change "glacial inception" to "ice sheet expansion."

P3, L1-2: "the differences in the ice age-gas age difference". Delete one of the "difference". Field site and analytical methods: The title of the chapter should reflect the fact that it also describes ice cores drilled in different seasons.

We will replace "ice age-gas age differences" with "delta age."

P3, L33 and P4, L1: The phrase "a second exploratory core" appears twice for different cores (PICO and BID cores).

We will remove the redundancy by changing the second "a second exploratory core" to "a third core."

P3, L35-37: Please clarify if this measurement was done in the field (brown markers in Fig 2), and when and how did you conduct the whole CH4 measurements. It is unclear to me because you mention D/O 19 but not the larger increase at D/O 17 (D/O 17 is only mentioned earlier for the "-380 m" core). Did you obtain all data before you drilled the BID core? Didn't you take the D/O 17 CH4 transition in the 2014-15 PICO core into account for the preliminary age estimate in the field?

We will clarify that the measurements described on P3, L35-37 were done in the field. We mention the DO 19 increase here because it is a marker in the gas phase for the MIS 5/4 transition. The larger rise at DO 16/17 is the marker for the MIS 4/3 transition. We did take into consideration that we found the MIS 4/3 transition in the early exploratory PICO core at -380 m on the Main Transect, which is why we moved 1 km down glacier to hopefully find slightly older ice that contained the MIS 5/4 transition.

We will clarify in the text how we used the early CH4 data to inform our site selection for the second PICO core and the BID cores.

P4, L3-5: Please clarify which data you mean (Fig. 2, green markers). Also, ice sampling for d18Oatm and d15N is not mentioned here (2014-15 BID core) but there are data points in Fig. 2 that says the measurements were done in two years (2016 and 2017). Please give full explanation about the cores, sampling, measurements and periods for all data you present in a better way (not only about this core; using a table may be a good way).

We will clarify which data we mean with a reference to the figure that says "(Figure 2, green markers)." We will give a more complete explanation of the sampling and measurements by reporting a table that lists each core, whether the core was drilled with the BID or the PICO, which measurements were made including where and when, and the analytical uncertainty of the measurement.

P5, L6-8: Didn't you take any samples to have overlaps with the previous cores? Did you make the sample cuttings for OSU and SIO in the field?

We think here the referee means P4, not P5. No, unfortunately we did not take overlapping samples. The BID cores were cut into quarter cores for SIO and OSU in the field, but the samples were cut at the respective laboratories.

P5, L9-10: The description of the sampling of the 2015-16 core in the laboratory is better placed after describing the core transportation. Overall, the descriptions of field and lab samplings and analyses are scattered so they should be better organized.

We think here the referee means P4, not P5. Here we are describing the sampling that happened in the field, so it makes sense to us to describe it before the transportation of the cores. We will clarify that this sampling occurred in the field before storing and transporting the cores. The organization of the sampling and analyses will be much improved with the table we described.

P5, second paragraph: This part is about field measurement methods so it should come earlier before the first presentation of the relevant data. And, for which core this paragraph's description applies (2015-16 core only)?

We think the referee means P4, not P5. We will move this paragraph to come earlier, before the discussion of the lab measurements. Correct, the continuous melter analyses in the field occurred in the 2015-2016 season, whereas only discrete analyses occurred in the field in 2014-2015.  We will clarify this.

We will clarify this point in the text.

P5, third paragraph: Please clarify which kinds of measurements were made for the different cores (maybe use a table). The measurement methods should come earlier than the first description of the data, or tell the readers that the methods are described later if you introduce the data first (like in the current manuscript).

We will make sure that all descriptions of the measurements come before we present the data. We will include a table that details the measurement metadata.

P5, L16: The CH4 field data from 4 - 5 m in the 2014-15 core disagree with the CFA data of 2015-16 core by several tens of ppb, which is much more than your precision and should be discussed as well.

It is not uncommon to have an outlier of several tens of ppb in discrete field measurements, as the precision is much worse than the laboratory analyses due to environmental conditions in the field laboratory and the use of a small, portable instrument. There is not a blank correction applied to these measurements, and there may also be depth offsets from the BID core. These factors likely cause the offsets seen in the top 4-5 m that referee 4 mentions here. We will include discussion of these factors so that readers are aware that the quality of the discrete field measurements is less than the lab data that we compare to in Figure 2.

**Results and discussion:**

P5, L36-37: The oldest tie point between the TG and EDC using d18Oice seems unacceptable given the different shapes of the isotopic curves of TG, EDC and EDML cores for this and other periods presented in Fig. 2.

The AIM events we recognize in the Taylor Glacier core are large features that exist in EDC and EDML. We state in the text that we recognize Taylor Glacier d18O is noisier than the records we match to. Nevertheless, the AIM features are unmistakable changes of up to ~ 3‰ that we think represent robust features for tie point selection. Our smoothing of the d18O noise helps identify the peaks and troughs of the features more clearly. These features are important for our record because they provide tie points that importantly resolve ambiguities in the nssCa (because nssCa varies little in the deeper part of the record).

We will expand the axes on Figure 3 so that the shapes of the isotope curves are more visible to

readers. We will add text clarifying what we see as robust features in the d18O ice record. We also picked tie points directly from the d18Oice records so that readers clearly see the variability at the AIM events that we are matching in the d18Oice records.

Revised Figure 3:

[Figure]

P5, L38 - P6, L1: Some of the dust tie points seem unacceptable or maybe you didn't explain the details of the manual matching. You put one at 12 m but why did you choose that particular one and not other peaks?

Referees 1-3 made similar comments concerning ambiguous tie points, including the tie point at

12 m. We assign the dust peak at 12 m (rather than at 9 m or at 15 m) to the dust peak in EDC at 73.6 ka because this way the AIM 19 and AIM 20 that we identify in the d18Oice line up with the AIM events in EDC. If the peaks at 9 m or 15 m are fit to the dust peak at 73.6 ka instead, the d18O no longer matches.

We will justify this in the text in the revised manuscript. We also picked tie points directly from d18Oice to avoid the ambiguity.

Another one at about 6.5 m is described as low point in dust, but the 2015-16 field data (purple in Fig. 2) actually show a peak there (I guessed that orange plot in Fig. 3 is the same as purple in Fig. 2, showing high values at the tie point). The grey lines for the dust (Fig. 2) are drawn between the EDC and TG purple data (orange in Fig.3), but is this particular one connects EDC and TG DRI data instead? Why did you choose the point where the two dust records from the same core disagree?

The peak that appears as a thin purple line in Figure 2 that exceeds the axis limits is either a measurement artifact in the raw data or too small of an event to match to EDC. The raw data shown in Figure 2 are not filtered for outliers, and we prefer to show the full raw data set to demonstrate the data quality. In the 40 cm above this noise there is a real peak (smaller amplitude up to 0.4 ug/g, 6.1 m) that appears to lead the dust rise in EDC in Figure 3.

We understand the confusion because we did not describe our criteria for matching dust peaks. We only fit features that span a range of depths on the order of at least 10's of centimeters and show structure (more than one data point comprising the peak). The peak that exceeds the axis limits is an example of noise because the high dust concentrations span less than 2 mm of ice. The smaller peak centered at 6.1 m that spans ~ 30 cm of ice is a real dust event.

In any case, the new tie point for the period before the MIS 4 onset is 7.75 m, 71.95 ka, chosen from the d18Oice variability. This way the dust ambiguity is avoided altogether.

We will expand Figure 2 so it is easier for reviewers to see the variability we are matching in the dust records. We will also describe in the text what we consider a true feature in the dust versus noise. We are also plotting smoothed versions of the d18Oice and particle count records (the two noisiest records from Taylor Glacier) so that the large-scale variations are seen clearly.

Around the one at 70.11 ka, the TG dust peak is offset compared to EDC dust peak (isn't it better not to match the highest point in the peak which have certain width?). Similar examples are at _65.6 and 63.9 ka.

The small adjustments to center peaks perfectly are not important considering that those differences are well within the errors we place on the ice age scale. Originally we left some of these peaks off center because we did not want to "over fit" the data. However, we understand the value of matching the peaks more perfectly in terms of communicating what we did and convincing readers that our matches are good.

We adjusted the tie points so that the peaks are more centered.

Overall, the lack of details on the matching force me to suspect that you chose the dust tie points while actually checking the resulting chronology by comparing Ca ion data from the two cores (I see that Ca ion data between TG and smoothed EDC compare much better than between the dust records), meaning that Ca is not just used for the validation of blind test (looking only dust) but

effectively involved in the tuning. Otherwise, how could you choose the dust tie point at 12 m? In fig. 3, dust data look like bar graph (vertical grey and orange bars) but they should actually be line plots. It is hard to evaluate the match in this figure so please improve the plots.

This is true that there was some iterative feedback from the other records. We thought the way that this was described in the original text was sufficient, but we see how the reader could be misled by the very good fits between, for example, nssCa but not particle count. We revised the tie point scheme so that more tie points are chosen for all of the records (d18Oice, nssCa, and particle counts) to be more transparent about our tie point choosing process. For the gas tie points we do not see good candidate tie points in the d18Oatm or CO2 that would improve upon the tie points already picked from CH4. Also see the response to other referees above about our hesitation to value-matching CO2.

We will revise the ice age scale tie points including more tie points for d18O ice and nssCa in addition to particle counts. We will present them clearly in the tables and justify our choices clearly in the text.

From the text (linear interpolation), I think the TG depth-age plot (Fig. 3b) should be straight lines between the tie points, but they don't look like so. A clear example is at an inflection point in the ice chronology at about 72.3 ka, for which there is no ice tie point. There might be my misunderstanding and if so please give full explanation for the interpolation. Please also plot markers at the tie points on the depth-age curves (Fig. 3b). The depth-age and delta-age lines in the figure are too low in resolution (the lines consist of tiny segments of horizontal and vertical lines, like aliasing in low resolution digital images).

Curvature in the age scales in Fig. 2 and Fig. 3 is an illusion because of tie points that are close to one another in depth or because of the low graphic resolution of the figures. We moved the "B" panels in Figure 3 and Figure 4 to their own figure that is easier to see (Figure 5).

Figure 5:

[Figure]

You should reject two youngest CH4 tie points. Cracking and contamination should increase the
measured CH4 concentration, so those two tie points are probably put on contamination peaks
(note large disagreements between purple line, red line and brown markers). Perhaps you can use
the peak at 3 m in brown data (if you take only low values in the two of the CFA data you see the
same peak, which is uncertain but this could be a true atmospheric peak concentration). The
match of d18Oatm records looks somewhat uncertain especially for the older one.

We do not want to match the brown data in Fig 2 because it was measured on a system in the
field that is lower precision. That tool is used as a rough guide for determining ages in the field,
and those data must always be verified in the lab. Other referees also had issues with tie points/
data in the top 4 meters of our cores.

We again defer to the editor's guidance on this issue as we see two options for moving forward –
(1) throw out all data 0-4 m and only interpret the section of core where there is the most reliable
delta age data, or (2) emphasize more strongly in the text that we are NOT rigorously interpreting
0-4 m, merely providing a plausible gas chronology for the 0-4 m section based on our view that
the CH4 field data are showing the true atmospheric signal.  Choice 2 is our preference.

Why did you connect the oldest d18Oatm data point to the beginning of the d18Oatm enrichment
in NGRIP data (why not the second oldest data point in TG d18Oatm which is the highest)? I
think the measurement precision is high for the TG dataset, but then I wonder what is the gap at
17m between the 2014-15 and 2015-16 cores. It might suggest depth offset between the two
cores. Please discuss.

We will change the tie point to match the second (and most depleted d18Oatm) data point to the

NGRIP data. Yes, there is an offset between the 2015-2016 core and the 2014-2015 core d18Oatm (~0.05‰) at 17m. This could imply a depth offset - that the cores are in fact not supposed to overlap there because there is a depth logging error. The measurement precision, which is quite good, seems to suggest this is more likely the case. Unfortunately it is not possible to deduce what the depth error actually is here, but it is worth noting that shifting the red data 20 cm deeper (our estimated depth uncertainty, stated in the paper) would result in a plausible scenario where d18Oatm is decreasing monotonically with depth.

For matching d18Oatm records, why did you only use the NGRIP data as the reference? There are clear discrepancies in the values (probably regardless of the matching quality) for some periods (_73 and 64-49 ka). I think Siple Dome d18Oatm data (Severinghaus et al., 2009) is of higher precision (not only measurement precision but also smaller and smoother thermal fractionation) and resolution, and Siple Dome and TG were measured in the same lab. Siple Dome also has the data younger than _63 ka where NGRIP data is lacking. So there seem good reasons that you should try using it as well (of course you have to match SD to AICC2012 using CH4). There may be a hope to match around 60-65 using small fluctuations in d18Oatm.

We agree that Siple Dome would be ideal, but the recently published age scale (Seltzer et al., 2017) only extends to 50 ka. We would need to first sync the rest of the age scale to AICC 2012 to be consistent with our record, which we think is outside the scope of this work. We are also aware of other efforts to sync the Siple Dome age scale (Buizert, personal communication) and would thus prefer not to do it. If you look at other d18Oatm records for this time period (TALDICE, Vostok) you will see that they are very low resolution and offer no clear alternative for tying the d18Oatm more robustly than what we have done. You will also see that the offsets between TALDICE and Vostok are of the same magnitude as the offsets between TG and NGRIP.

P6, L1-3: You should take into account the potential age error due to linear interpolation between tie points. The comparison between linear and cubic spline interpolation is insufficient as the demonstration of the age uncertainty between tie points. You should consider using available gas records as much as possible (CO2, d18Oatm; see comments above and below).
We interpret this comment to mean that we should value-match in between tie points where our Taylor Glacier data show differences from the reference records. There is only one section where our records depart significantly from the reference record (CO2 mismatch between 64-60 ka), and we deliberately chose not to value-match the CO2 data given that CO2 offsets between different ice cores are a known and as of yet unresolved issue (Luthi et al., 2008). It is unclear to us where else we might be introducing large errors due to linear interpolation. We think that the uncertainty we estimate with our method (interpolating between maximum and minimum ages at each tie point to generate an oldest and youngest age model) reasonably estimates the uncertainties between tie points (Figure 5 above).

P6, L9-16: Agreement of TG CO2 with existing records is overall very good. However, I think the decision not to use CO2 for the synchronization between 60 and 65 ka is not satisfactory especially because there is no other tie points. You should at least try matching CO2 and look how the resulting chronology look reasonable or not.

If we match the CO2 to the (Bereiter et al., 2015) composite where referee 4 describes (around 60.5 ka), the gas record shifts to older ages by ~ 650 years. The error we stated (in Table 1) is already larger than this, so the case where CO2 is matched is essentially already accounted for if you consider our error range. In this section we would have to value-match the CO2 because there are no robust or obvious inflection points. We prefer not to value-match using CO2 because

of the unresolved issue of CO2 offsets between different ice cores, described in the text.

We will draw the errors in the age scale on our plot of ice age and gas age versus depth so that the errors in the chronology are more visible to readers.

P6, L20: See the comment above about the dust and Ca.
P7, L19-22: See comment above about CH4 for 0-4 m.

P7, L31-34: Explanation is insufficient. What do you mean by "surveying the value matched or correlated data"? How exactly did you consider resolution and analytical errors.
In the original manuscript we assigned maximum/ minimum ages to each tie point that estimated the range of possible ages. Our choice of age range for each tie point was based on consideration of (1) the resolution of the data for a given feature that we matched (i.e. do we know the age of a true peak or trough in the data, or is it masked by low resolution?), (2) the analytical uncertainty of the data that we matched to, and (3) how robust (or possibly ambiguous) the matched feature was (i.e. could we be matching the wrong feature?). If any of the three criteria were poor or ambiguous then we enlarged the age uncertainty range to reflect a worse quality match. We then propagated the uncertainties by interpolating through the maximum and minimum age at each tie point, which resulted in an oldest and youngest possible chronology (and therefore also a maximum and minimum delta age via calculation). We considered calculating a fit index for each tie point and a probability distribution for each match, but this method is more suited for value-matching data whereas we are matching features where multiple parameters are changing at the same time (i.e. peaks and troughs in $d^{18}O_{atm}$ and $CH_4$, or in $nssCa^{2+}$ and particle count). We think that an algorithm will not necessarily do this better than we can do by eye, or at least the difference will be negligible for the delta age story we are telling in this manuscript.

We think the uncertainties estimated by the methods described above are justified because (1) even with assigning very generous uncertainty to each tie point, the uncertainty does not affect our interpretations about delta age (i.e., the delta age that we calculate after propagating the uncertainties to our chronologies is still large during MIS 4 and supports the notion of the development of a steep accumulation gradient between the Taylor Dome coring site and the Taylor Glacier accumulation zone), (2) the uncertainty we estimate for delta age is realistic and is similar in magnitude to the uncertainty in delta age from other Antarctic ice cores, including the delta age uncertainties in Baggenstos et al. 2018, and (3) the $CH_4$ record on our new gas age scale matches Hulu speleothem $\delta^{18}O$ very closely at the onset of DO 16/17 and DO 19 (Figure 6 below). The last point supports our choice of tie points for synchronizing to the AICC2012 gas age scale because the Hulu data are independently dated.

In the revised text we will explain more clearly how we assigned uncertainty to each tie point, and we will justify more clearly why we think the uncertainty is reasonable.

Figure 6 - Comparison of the timing of abrupt $CH_4$ changes in the new Taylor Glacier ice core with abrupt events in the Hulu speleothem record.

[Figure]

P8, L1-4: The offset of 20 cm is quite large when comparing laboratory measurements by CFA and discrete samples. Please explain the possible causes for this. Why don't you correct the CFA depth assignment by matching the depths of the sharp CH4 features? Why is the 300 yr estimate a conservative one? You have other CH4 tie points where you have much steeper depth-age slope, so it does not sound conservative at all.

The 20 cm depth offset does not have to do with CFA versus discrete samples. We explained the cause of the depth offset in the text – it is because of angle breaks in BID cores that were not aligned and accounted for during drilling. It is possible that there are smaller depth errors due to mistakes in depth logging, but these must be smaller than the offsets introduced due to angle breaks.

The place in the ice core where age is changing the most with depth is where the slope is the shallowest on Figure 3 (age axis is on the bottom). If you compute the age change for 20 cm along this slope, you get 416 years. So the referee is right that we did not estimate the value high enough. We will change the conservative estimate to 420 years and propagate it accordingly.

P8, L9-10: Absolute age uncertainty attached to AICC2012 for this age range is probably incorrectly cited. Please check. Also, it is useful to refer to Chinese speleothem ages (using CH4 and d18Oatm) for the possible range of absolute age error for the studied period.

We cited the age uncertainty from AICC2012 incorrectly. We will correct the absolute uncertainty that we cite to 1 σ = 1500 years for the EDML gas age scale and 1 σ = 2500 years for the EDC ice age scale. A comparison of Taylor Glacier CH4 to Hulu d18O shows very good agreement in terms of the onsets of DO 19 and DO 16/17 (Figure 6, above). Though we necessarily acquire the aforementioned uncertainties when using AICC2012 as our reference age scale, we think that the absolute age uncertainty in our gas age scale is probably less than this given the close match to Hulu. We also note that the relative errors in our ice cores will be less

than the total propagated EDC and EDML 1 σ uncertainties because the uncertainties in gas age and ice age are correlated with depth.

P8, L14: It is common to use small 'a' for the term Delta-age (not Delta-Age).

We will change Delta-Age to Delta-age.

P8, L16-18: A better explanation would be that delta-age depends on firn thickness (ice or water equivalent) and accumulation rate, and the firn thickness depends primarily on temperature and accumulation rate.

We agree that our explanation is lacking. We will summarize the controls on delta age more thoroughly. We like referee 4's explanation of delta age here and will include it in our revised delta age discussion.

P8, L24: I think a weakness of the delta age estimation and whole discussion based on it is that there is no ice and gas age estimates for the same depth, so the uncertainty of delta-age depends on the uncertainties of ice and gas ages between tie points, which is not evaluated well. You should try to have more constraints on the gas age (with CO2 or d18Oatm) between 60 and 69 ka where you have the very large delta-age (which is the basis for your argument of "virtually zero accumulation rate").

We are hesitant to value-match the CO2 data (there is not another way to tie the CO2 given the nature of the variability – inflection points are somewhat unclear/ poorly defined, a ramp-fit algorithm or some other statistical tool would be needed to define them. We also chose not to fit the d18Oatm in this interval given the offsets between TG and NGRIP d18Oatm between 62-68 ka. The most convincing variability in d18Oatm occurs during DO 18 where we already have a more robust CH4 tie point. If you look at other d18Oatm records (e.g. Vostok, TALDICE, or EDC) the resolution is too low to match in this range. You will also see that the offsets in d18Oatm between those cores are larger than the offsets between TG and NGRIP.

We agree with the referee that it would be better to have ice and gas age tie points for the same depths, but the reality is there are not robust features in the gas and ice phases at all depths.

P8, last paragraph: Discussion here is too qualitative (with the words like "near zero", "where ice accumulates very slowly"). Please be more quantitative by giving possible range for the surface mass balance in the TG accumulation zone in MIS 4 from your data.

See other comments above regarding estimating the accumulation rate quantitatively. We think that a quantitative estimate is not meaningful given the uncertainties in firn modeling. If we use the d15N approach, the uncertainties in the convective zone are so large that the accumulation rate estimate spans at least 1 order of magnitude (more detailed discussion of this below).

We preferred not to provide quantitative estimates of accumulation rate given that they require extrapolating beyond the calibration range of firn models. An estimate is 0.5 mm/yr-2 cm/yr (described above and below), which we will include in the text with the caveat that it is a very cautious estimate.

P8, L35-36: d15N is not only controlled by gravitational fractionation. You should introduce it appropriately.

We will summarize the d15N controls more completely including discussing the effects of temperature gradients in the diffusive zone as well as mixing in the convective zone.

P8, L37 - P9, L1: I agree that d15N likely reflect firn thinning during MIS4, but the change is not linear with respect to delta-age. About half of the d15N change actually occurs at around 71 ka when delta-age is still stable at _4 kyr, and it is in fact before MIS4 (so this should also be discussed in your climatic discussion part). You should definitely discuss this large change while delta-age is small and stable, and in doing so, also run firn models for the high and low d15N around 70-73 ka. You might get some idea on how much non-gravitational signal could be contained in d15N data, or how much accumulation reduction is needed to explain d15N at that change (assuming that the change is purely gravitational). Another exercise is to use the presumed ratio (firn thickness in ice-equivalent) / (real firn thickness), which seems stable over wide range of temperature and accumulation rate (_0.7 from Parrenin et al., 2012, CP) and use the d15N and delta age to estimate accumulation rate, for example at 72, 70 and 61 ka and some times in between. I think you obtain a mm or so for the 10000-yr delta-age with d15N around 61 ka.

We agree with referee 4 that the changes in d15N and delta age are not linear with each other. We will discuss this point.

We performed this exercise following (Parrenin et al., 2012) and (Buizert et al., 2012) and found that d15N implied accumulation between < 1 mm/yr – 2 cm/yr depending on the height of the convective zone. The convective zone is the major uncertainty in this calculation. If the Taylor Glacier accumulation area resembles something like Megadunes, Antarctica (Severinghaus et al., 2010), then a convective zone of up to 25 m may be possible, which would drive d15N to much lower values for accumulation rates on the order of a couple cm/yr. On the other hand, if the convective zone is 5-10 m deep then accumulation rates must fall to something on the order of a couple mm/yr.

P9, L18-20: You should actually put the constraint ice age > gas age.

We will add this constraint by removing delta age data where it is < 0. This does bring up the issue of what the minimum delta age should be… delta age = 0 is equally impossible, for example. Rather than force delta age data to be an arbitrary minimum when < 0, we will simply remove data that is < 0 and explain in the text that we also think values near 0 are likely wrong but that we do not know the minimum delta age so chose not to impose that constraint.

P9, L22: "onset of the last glacial period" is confusing as it is often mean the MIS5e-5d transition.

We will change it to "onset of full MIS 4 glaciation."

P9, L30-31: Suggesting the accumulation control on the depth-age curve instead of thinning based on the delta-age and d15N while avoiding deeply discussing delta-age uncertainty and d15N is not acceptable (see above).

We will develop our discussion of delta age more deeply by summarizing more completely the controls on delta age, by summarizing the meaning of d15N including reference to the barometric equation for computing the height of the diffusive zone, and including more references to previous work on delta age (Parrenin et al., 2012; Buizert et al., 2012; Martinerie et al., 1994; Martinerie et al., 1992) as well as published firn models (Goujon et al., 2003; Herron and Langway, 1980). We still think that accumulation and temperature have the greatest first-order

control on delta age, and that an extremely high delta age as found here is very unlikely to occur without extremely low accumulation. When discussing controls on delta age we will distinguish which controls we think are secondary (e.g. impurities, wind stress, thinning of firn) versus primary (e.g. temperature and accumulation). Since HL is our firn model of choice we will add 1-2 sentences describing the model and the particular controls on densification (i.e. densification to 0.55 g/mL does not depend on accumulation, but second-stage densification to 0.84 g/mL depends highly on accumulation.)

Conclusions P11, L26: The statement "virtually zero net accumulation" needs more solid basis and quantification as commented above. The rest of the discussion (about atmospheric circulation and ice sheet) may change after the revision with deeper look into your data, so I would not review it (and it is not my speciality in any case).

See above comments about accumulation rate estimates. We will include 0.5 mm/yr-2 cm/yr as our estimate of accumulation rate, and we will clearly state that it is a conservative estimate that is limited by the fact that we lack a firn model to accurately describe very low accumulation conditions.

Bereiter, B., Eggleston, S., Schmitt, J., Nehrbass-Ahles, C., Stocker, T. F., Fischer, H., Kipfstuhl, S., and Chappellaz, J.: Revision of the EPICA Dome C CO2 record from 800 to 600kyr before present, Geophysical Research Letters, 42, 542-549, 10.1002/2014gl061957, 2015.

Buizert, C., Martinerie, P., Petrenko, V. V., Severinghaus, J. P., Trudinger, C. M., Witrant, E., Rosen, J. L., Orsi, A. J., Rubino, M., Etheridge, D. M., Steele, L. P., Hogan, C., Laube, J. C., Sturges, W. T., Levchenko, V. A., Smith, A. M., Levin, I., Conway, T. J., Dlugokencky, E. J., Lang, P. M., Kawamura, K., Jenk, T. M., White, J. W. C., Sowers, T., Schwander, J., and Blunier, T.: Gas transport in firn: multiple-tracer characterisation and model intercomparison for NEEM, Northern Greenland, Atmospheric Chemistry and Physics, 12, 4259-4277, 10.5194/acp-12-4259-2012, 2012.

Craig, H., Horibe, Y., and Sowers, T.: GRAVITATIONAL SEPARATION OF GASES AND ISOTOPES IN POLAR ICE CAPS, Science, 242, 1675-1678, 10.1126/science.242.4886.1675, 1988.

Goujon, C., Barnola, J. M., and Ritz, C.: Modeling the densification of polar firn including heat diffusion: Application to close-off characteristics and gas isotopic fractionation for Antarctica and Greenland sites, Journal of Geophysical Research-Atmospheres, 108, 18, 10.1029/2002jd003319, 2003.

Herron, M. M., and Langway, C. C.: FIRN DENSIFICATION - AN EMPIRICAL-MODEL, Journal of Glaciology, 25, 373-385, 1980.

Luthi, D., Le Floch, M., Bereiter, B., Blunier, T., Barnola, J. M., Siegenthaler, U., Raynaud, D., Jouzel, J., Fischer, H., Kawamura, K., and Stocker, T. F.: High-resolution carbon dioxide concentration record 650,000-800,000 years before present, Nature, 453, 379-382, 10.1038/nature06949, 2008.

Martinerie, P., Raynaud, D., Etheridge, D. M., Barnola, J. M., and Mazaudier, D.: PHYSICAL AND CLIMATIC PARAMETERS WHICH INFLUENCE THE AIR CONTENT IN POLAR ICE, Earth Planet. Sci. Lett., 112, 1-13, 10.1016/0012-821x(92)90002-d, 1992.

Martinerie, P., Lipenkov, V. Y., Raynaud, D., Chappellaz, J., Barkov, N. I., and Lorius, C.: AIR CONTENT PALEO RECORD IN THE VOSTOK ICE CORE (ANTARCTICA) - A MIXED RECORD OF CLIMATIC AND GLACIOLOGICAL PARAMETERS, Journal of Geophysical Research-Atmospheres, 99, 10565-10576, 10.1029/93jd03223, 1994.

Parrenin, F., Barker, S., Blunier, T., Chappellaz, J., Jouzel, J., Landais, A., Masson-Delmotte, V., Schwander, J., and Veres, D.: On the gas-ice depth difference (Delta depth) along the EPICA Dome C ice core, Climate of the Past, 8, 1239-1255, 10.5194/cp-8-1239-2012, 2012.

Seltzer, A. M., Buizert, C., Baggenstos, D., Brook, E. J., Ahn, J., Yang, J. W., and Severinghaus, J. P.: Does delta O-18 of O-2 record meridional shifts in tropical rainfall?, Climate of the Past, 13, 1323-1338, 10.5194/cp-13-1323-2017, 2017.

Severinghaus, J. P., Albert, M. R., Courville, Z. R., Fahnestock, M. A., Kawamura, K., Montzka, S. A., Muhle, J., Scambos, T. A., Shields, E., Shuman, C. A., Suwa, M., Tans, P., and Weiss, R. F.: Deep air convection in the firn at a zero-accumulation site, central Antarctica, Earth Planet. Sci. Lett., 293, 359-367, 10.1016/j.epsl.2010.03.003, 2010.

---

## Author Comment (AC5) · 13 Nov 2018

**Summary of changes to manuscript CP-2018-53:**

**Spatial pattern of accumulation at Taylor Dome during the last glacial inception: Stratigraphic constraints from Taylor Glacier**

Given substantial changes to the manuscript following comments from reviewers, we summarized the main changes here so that the editor can more easily keep track. The major revisions generally fall into the numbered categories below. Please see the other posted documents for detailed responses to the referees' specific comments.

1. Adjust tie points

Four referees questioned our choice of tie points, mainly because they found certain tie points ambiguous. We will add text to the revised manuscript to justify our tie point selections more clearly. We increased the size and resolution of Figures 3 and 4 (see below) in the revised manuscript so that it is more apparent to readers why we chose to match the variability the way we did. We will also increase the size and resolution of Figure 2 for the same purpose. We are plotting smoothed particle count data (instead of raw data as in the original manuscript) so readers can see the dust events during MIS4 more easily and can tell the difference between true events versus noise. We also plotted the tie points on Figure 3 and Figure 4 so the features we matched are clear to readers. We refined/ adjusted the tie points as described below, based on feedback from reviewers and what we deduced was the reason for the perceived ambiguities. The final tie point selections are listed in Tables 1-4 below, and the revised versions of Figure 3 and Figure 4 are also included below so the quality of the matches to the reference ice core records can be assessed.

   a. Taylor Glacier gas tie points remain the same as in the original manuscript except for the two oldest tie points.

      i. Reviewers had few specific comments about the Taylor Glacier gas tie point choices, except one referee questioned the validity of the gas chronology in the top 4 m where we cannot rule out contamination of the gases by shallow cracks in the glacier surface. We decided to keep the tie points the same in the revised manuscript as in the original, including the top 4 m, but we will emphasize in the text that we do not use or interpret the chronology in the top 4 m rigorously. Our interpretations of the high delta age values do not depend on these data because the highest delta age values occur at ~ 5.5 m with delta age increasing steadily and significantly between 5.5-10 m depth. We also clearly mark the top 4 m in the gas data in gray in Figure 3 so that readers can see which portion of the record is shallower than 4 m.

      ii. One reviewer suggested picking tie points using the $CO_2$ data. We originally chose not to do this because $CO_2$ offsets between ice cores are a known and unresolved issue, hence we hesitate to value-match the $CO_2$ data. We prefer not to pick tie points using the $CO_2$ data in the revised manuscript for the same

reason. We would like to point out that even if we did choose tie points using $CO_2$, it would not change the gas age scale enough to alter our interpretations about the high delta age values. This point is discussed in more detail in response to the reviewer's specific comment.

iii. We changed the two oldest tie points slightly. Based on a reviewer's comment we changed the oldest $\delta^{18}O_{atm}$ tie point to match the low inflection point in the NGRIP $\delta^{18}O_{atm}$ data at 73.74 ka (instead of 74.65 ka in the original manuscript) before the transition. We also changed the second oldest $\delta^{18}O_{atm}$ tie point to match the mid point in the NGRIP $\delta^{18}O_{atm}$ transition at 72.7 ka.

b. Taylor Glacier ice tie points changed somewhat from the original manuscript, following the comments from reviewers:

i. For the majority of the Taylor Glacier ice tie points (6 out of 9 dust tie points) we decided to match Taylor Glacier $nssCa^{2+}$ variability to EDC $nssCa^{2+}$ instead of matching Taylor Glacier insoluble particle count to EDC laser dust. The reasons are: (1) the $nssCa^{2+}$ measurements are more quantitative than the insoluble particle count measurements, (2) our $nssCa^{2+}$ record is less noisy than the insoluble particle count record, and (3) comparing Taylor Glacier $nssCa^{2+}$ directly to EDC $nssCa^{2+}$ is a "like-to-like" comparison, whereas the methods for measuring laser dust and insoluble particle counts are less similar. As a consequence of picking $nssCa^{2+}$ variability instead of insoluble particle counts, many of the final tie points are shifted in depth and age slightly (< 10 cm and < 0.1 ka) relative to the tie points presented in the original manuscript in order to align features exactly.

ii. We added two additional $\delta^{18}O_{ice}$ tie points (at 7.75 m and 12.20 m) for a total of three $\delta^{18}O_{ice}$ tie points. This is so readers can see more clearly how we interpret the isotope variability at AIM 19 and AIM 20 to match EDC (Table 2, Figure 3). This avoids an issue that two referees commented on: matching $nssCa^{2+}$ in the deeper part of the core where the $nssCa^{2+}$ variability is comparatively small and alignment of individual peaks is possibly more ambiguous. By picking tie points directly from $\delta^{18}O_{ice}$ we avoid this ambiguity – readers see immediately how the variations in the $\delta^{18}O_{ice}$ correspond to those at AIM19 and AIM20 in EDC.

iii. We added one additional $nssCa^{2+}$ tie point (4.94 m) to match a low value in the EDC $nssCa^{2+}$ during early MIS4.

iv. We changed the 4.19 m tie point to 4.47 m because there is an offset at this depth between the peak in Taylor Glacier insoluble particle count measured in

the field versus the peak in Taylor Glacier nssCa$^{2+}$ measured in the laboratory, and we prefer to match the nssCa$^{2+}$ instead of insoluble particle counts.

    v. We eliminated the 6.3 m particle count tie point that matched the low dust concentration before the MIS4 dust onset. This inflection point in dust is possibly ambiguous, and we prefer the $\delta^{18}O_{ice}$ maximum at AIM19 (7.75 m) to provide age constraints in this section of the core.

    vi. We still chose to match the variability in the Taylor Glacier insoluble particle count to the EDC laser dust for three tie points where nssCa$^{2+}$ variability was small but where particle count variability was larger and showed similar features to laser dust (Table 2).

c. Taylor Dome gas age tie points changed somewhat from the original manuscript, following comments from reviewers and following the publication of (Baggenstos et al., 2018):

    i. We adopted three published tie points from (Baggenstos et al., 2018) that tie the Taylor Dome $CO_2$ (Indermuhle et al., 2000) to the WAIS Divide $CO_2$ record (currently unpublished). Note that the $CO_2$ data are not value-matched by (Baggenstos et al., 2018), rather the tie points represent the mid points of transitions at A3 and A4.

    ii. Otherwise the tie points matching Taylor Dome $CH_4$ variability to EDML $CH_4$ (Schilt et al., 2010) are the same as in the original manuscript.

d. Taylor Dome ice age tie points changed somewhat from the original manuscript, following comments from reviewers and following the publication of (Baggenstos et al., 2018):

    i. We adopted four published tie points from Baggenstos et al. 2018 that tie Taylor Dome nssCa$^{2+}$ (Mayewski et al., 1996) to WAIS nssCa$^{2+}$ (Table 4).

    ii. We eliminated the 456.3 m tie point and the 466.8 m tie point from the original manuscript because the tie points from (Baggenstos et al., 2018) provide age constraints around these depths.

    iii. We added tie points to more precisely match Taylor Dome Ca$^{2+}$ variations to EDC laser dust and EDC nssCa$^{2+}$ (Table 4, Figure 4).

    iv. We added 4 $\delta^{18}O_{ice}$ tie points to clearly demonstrate the variability we see at AIM 19 and AIM 20.

Table 1 – Tie points relating Taylor Glacier depth to gas age on the AICC 2012 timescale. Gray shading indicates tie points < 4 m depth where abundant cracks in shallow ice may cause contamination of the gas records. "DO" refers to Dansgaard-Oeschger event.

| Depth (m) | Gas Age (ka) | Parameter, Data Source | Feature Description | Tie Point Source |
|---|---|---|---|---|
| 1.74 | 58.21 | $CH_4$/ this study | Peak during DO16/17, synch. to EDML $CH_4$ | This study |
| 3.15 | 59.10 | $CH_4$/ this study | Peak during DO16/17, synch. to EDML $CH_4$ | This study |
| 4.19 | 59.66 | $CH_4$/ this study | Midpoint transition DO16/17, synch. to EDML $CH_4$ | This study |
| 5.40 | 59.94 | $CH_4$/ this study | Low before DO16/17, synch. to EDML $CH_4$ | This study |
| 7.79 | 64.90 | $CH_4$/ this study | Peak during DO18, synch. to EDML $CH_4$ | This study |
| 11.24 | 69.92 | $CH_4$/ this study | Small peak between DO19 and DO18, synch. to EDML $CH_4$ | This study |
| 12.43 | 70.62 | $CH_4$/ this study | Low after DO19, synch. to EDML $CH_4$ | This study |
| 13.25 | 71.21 | $CH_4$/ this study | High before transition late DO19, synch. to EDML $CH_4$ | This study |
| 16.20 | 72.27 | $CH_4$/ this study | Midpoint transition DO19, synch. to EDML $CH_4$ | This study |
| 16.94 | 72.70 | $\delta^{18}O_{atm}$/ this study | Midpoint transition, synch. to NGRIP $\delta^{18}O_{atm}$ | This study |
| 19.27 | 73.74 | $\delta^{18}O_{atm}$/ this study | Low before transition, synch. to NGRIP $\delta^{18}O_{atm}$ | This study |

Table 2 - Tie points relating Taylor Glacier depth to ice age on the AICC 2012 timescale. Ice phase parameters (dust and $\delta^{18}O_{ice}$) are unaffected by surface cracks below the top 40 cm (Baggenstos et al. 2018). Gray shading indicates tie points < 0.4 m depth where abundant cracks in shallow ice may cause contamination of the ice records due to local dust deposition. "AIM" refers to Antarctic Isotope Maximum event. "MIS" refers to Marine Isotope Stage.

| Depth (m) | Ice Age (ka) | Parameter/ Data Source | Feature Description | Tie Point Source |
|---|---|---|---|---|
| 0.34 | 61.47 | Insoluble particles/ this study | Peak near end of MIS4, synch. to EDC laser dust | This study |
| 1.25 | 63.93 | $nssCa^{2+}$/ this study | Peak late MIS4, synch. to EDC $nssCa^{2+}$ | This study |
| 1.80 | 64.91 | Insoluble particles/ this study | Peak late MIS4, synch. to EDC laser dust | This study |
| 2.45 | 65.65 | Insoluble particles/ this study | Peak mid MIS4, synch. to EDC laser dust | This study |
| 3.10 | 66.73 | $nssCa^{2+}$/ this study | Peak mid MIS4, synch. to EDC $nssCa^{2+}$ | This study |
| 4.47 | 68.63 | $nssCa^{2+}$/ this study | Peak mid MIS4, synch. to EDC $nssCa^{2+}$ | This study |
| 4.94 | 69.72 | $nssCa^{2+}$/ this study | Low early MIS4, synch to EDC $nssCa^{2+}$ | This study |
| 5.60 | 70.20 | $nssCa^{2+}$/ this study | Peak early MIS4, synch. to EDC $nssCa^{2+}$ | This study |

| | | | | |
|---|---|---|---|---|
| 7.75 | 71.95 | $\delta^{18}O_{ice}$/ this study | Peak AIM19, synch. to EDC $\delta^{18}O_{ice}$ | This study |
| 12.20 | 73.62 | $\delta^{18}O_{ice}$/ this study | Low between AIM19 and AIM20, synch. to EDC $\delta^{18}O_{ice}$ | This study |
| 16.62 | 75.75 | $\delta^{18}O_{ice}$/ this study | Peak AIM20, synch. to EDC $\delta^{18}O_{ice}$ | This study |
| 19.76 | 76.50 | $nssCa^{2+}$/ this study | End of record, loosely constrained, synch. to EDC $nssCa^{2+}$ | This study |

Table 3 – Tie points relating Taylor Dome depth to gas age on the AICC 2012 timescale.

| Depth (m) | Gas Age (ka) | Parameter/ Data Source | Feature Description | Tie Point Source |
|---|---|---|---|---|
| 455.95 | 54.667 | $CO_2$/ Indermühle et al. 2000 | Midpoint transition A3, synch. to WAIS $CO_2$ | Baggenstos et al. 2018 |
| 460.90 | 57.913 | $CO_2$/ Indermühle et al. 2000 | Midpoint transition A4, synch. to WAIS $CO_2$ | Baggenstos et al. 2018 |
| 464.62 | 59.99 | $CH_4$/ Brook et al. 2000 | Low before DO16/17, synch. to EDML $CH_4$ | This study |
| 467.10 | 62.303 | $CO_2$/ Indermühle et al. 2000 | Midpoint transition A4, synch. to WAIS $CO_2$ | Baggenstos et al. 2018 |
| 474.95 | 65.50 | $CH_4$/ this study | Low before DO18, synch. to EDML $CH_4$ | This study |
| 487.83 | 73.10 | $CH_4$ /this study | Low after DO19, synch. to EDML $CH_4$ | This study |
| 493.50 | 76.05 | $CH_4$/ this study | Midpoint transition DO19, synch. to EDML $CH_4$ | This study |

Table 4 - Tie points relating Taylor Dome depth to ice age on the AICC 2012 timescale.

| Depth (m) | Ice Age (ka) | Parameter/ Data Source | Feature Description | Tie Point Source |
|---|---|---|---|---|
| 455.10 | 55.80 | $Ca^{2+}$/ Mayewski et al. 1996 | $nssCa^{2+}$ synch. to WAIS | Baggenstos et al. 2018 |
| 457.60 | 58.85 | $Ca^{2+}$/ Mayewski et al. 1996 | $nssCa^{2+}$ synch. to WAIS | Baggenstos et al. 2018 |
| 463.30 | 61.47 | $Ca^{2+}$/ Mayewski et al. 1996 | Peak late MIS4, synch. to EDC laser dust | This study |
| 466.40 | 63.50 | $Ca^{2+}$/ Mayewski et al. 1996 | $nssCa^{2+}$ synch. to WAIS | Baggenstos et al. 2018 |
| 467.80 | 64.30 | $Ca^{2+}$/ Mayewski et al. 1996 | $nssCa^{2+}$ synch. to WAIS | Baggenstos et al. 2018 |
| 468.10 | 64.66 | $Ca^{2+}$/ Mayewski et al. 1996 | Low late MIS4, synch. to EDC laser dust | This study |
| 471.37 | 65.57 | $Ca^{2+}$/ Mayewski et al. 1996 | Peak mid MIS4, synch. to EDC laser dust | This study |
| 472.70 | 66.71 | $Ca^{2+}$/ Mayewski et al. 1996 | Peak mid MIS4, synch. to EDC $nssCa^{2+}$ | This study |
| 475.12 | 67.47 | $Ca^{2+}$/ Mayewski et al. 1996 | Low mid MIS4, synch. to EDC $nssCa^{2+}$ | This study |
| 476.90 | 68.63 | $Ca^{2+}$/ Mayewski et al. 1996 | Peak early MIS4, synch. to EDC $nssCa^{2+}$ | This study |
| 478.70 | 69.70 | $Ca^{2+}$/ Mayewski et al. 1996 | Low early MIS4, synch. to EDC $nssCa^{2+}$ | This study |
| 479.90 | 70.15 | $Ca^{2+}$/ Mayewski et al. 1996 | Peak early MIS4, synch. to EDC $nssCa^{2+}$ | This study |

| 484.30 | 71.95 | $\delta^{18}O_{ice}$/ Steig et al. 1998 | Peak AIM19, synch. to EDC $\delta^{18}O_{ice}$ | This study |
|---|---|---|---|---|
| 487.40 | 73.62 | $\delta^{18}O_{ice}$/ Steig et al. 1998 | Low between AIM19 and AIM20, synch. to EDC $\delta^{18}O_{ice}$ | This study |
| 490.80 | 75.75 | $\delta^{18}O_{ice}$/ Steig et al. 1998 | Peak AIM20, synch. to EDC $nssCa^{2+}$ | This study |
| 493.40 | 77.08 | $\delta^{18}O_{ice}$/ Steig et al. 1998 | Low before AIM20, synch. to EDC $nssCa^{2+}$ | This study |

Revised Figure 3:

[Figure]

Revised Figure 4:

[Figure]

Figure 5 – Age models for new Taylor Glacier 5/4 BID cores (A), Taylor Glacier delta age and $\delta^{15}$N-N$_2$ (B), and Taylor Dome revised age models (C), and Taylor Dome delta age and $\delta^{15}$N-N$_2$ (D). Red shading on Taylor Glacier gas age chronology and delta age indicates ice shallower than 4 m where surface cracks may affect the CH$_4$ age matching.

[Figure]

2. Estimate error

A number of reviewers commented on how we assessed the uncertainty in our chronologies - specifically reviewers said the uncertainty was not clearly presented, and one reviewer thought there might be a more realistic way to assess the uncertainty. Since our tie points are chosen by hand, there is not a probability distribution associated with the matches from which we can give a true 1-sigma uncertainty. In the original manuscript we assigned maximum/ minimum ages to each tie point that estimated the range of possible ages. Our choice of age range for each tie point was based on consideration of (1) the resolution of the data for a given feature that we matched, (2) the analytical uncertainty of the data that we matched to, and (3) how robust (or possibly ambiguous) the matched feature was (i.e. could we be matching the wrong feature?). If any of the three criteria were poor or ambiguous then we enlarged the age uncertainty range to reflect a worse quality match. We then propagated the uncertainties by interpolating through the maximum and minimum age at each tie point, which resulted in an oldest and youngest possible chronology (and therefore also a maximum and minimum delta age). We considered calculating a fit index for each tie point and a probability distribution for each match, but this method is more suited for value-matching data whereas we are matching features where multiple parameters are changing at the same time (i.e. peaks and troughs in d$^{18}$O$_{atm}$ and CH$_4$, or in nssCa$^{2+}$ and particle count). We think a computer algorithm will not necessarily do this better than we can do by eye, or at least the difference will be negligible for the delta age story we are telling in this manuscript.

We think the uncertainties estimated by the methods described above are justified because (1) even with assigning very generous uncertainty to each tie point, the uncertainty does not affect our interpretations about delta age (i.e., the delta age that we calculate after propagating the uncertainties to our chronologies is still large during MIS 4 and supports the notion of the development of a steep accumulation gradient between the Taylor Dome coring site and the Taylor Glacier accumulation zone), (2) the uncertainty we

estimate for delta age is realistic and is similar magnitude to the uncertainty in delta age from other Antarctic ice cores, including the delta age uncertainties for Taylor Glacier and Taylor Dome published in (Baggenstos et al., 2018), and (3) the $CH_4$ record on our new gas age scale matches Hulu speleothem $\delta^{18}O$ very closely at the onset of DO 16/17 and DO 19 (Figure 6). The last point supports our choice of tie points for synchronizing to the AICC2012 gas age scale because the Hulu data are independently dated.

In the revised manuscript we would prefer to estimate our uncertainty ranges the way we did originally, but we propose to (1) more clearly show the uncertainty on the age model by plotting the max/min chronologies on Figure 5 (above), not just the max/min delta ages that were shown in the original manuscript, and (2) enlarge the uncertainty ranges in response to reviewers' scrutiny, particularly where tie points were possibly more ambiguous. The revised uncertainties are shown in Figure 3 and Figure 4 as horizontal error bars on tie points, and the propagated max/min chronologies are displayed in Figure 5 as shading. We will also justify how we assessed the uncertainty more clearly in the text. We will also show the comparison to Hulu because it independently supports our gas age scale (Figure 6, below).

Two reviewers pointed out that we made a mistake when citing the absolute uncertainty in the AICC2012 chronology. We will correct the absolute uncertainty that we cite to 1 σ = 1500 years for the EDML gas age scale and 1 σ = 2500 years for the EDC ice age scale. Though we naturally acquire these uncertainties when using AICC2012 as our reference age scale, we think that the absolute age uncertainty in our gas age scale is probably less than this given the close match to Hulu. We also note that the relative errors in our ice cores will be less than the total propagated EDC and EDML 1 σ uncertainties because the uncertainties in gas age and ice age are correlated with depth.

Figure 6 – Comparison of the timing of abrupt $CH_4$ changes in the new Taylor Glacier ice core with abrupt events in the Hulu speleothem record.

[Figure]

3. Calculate accumulation rates/ firn modeling

Two referees suggested that we estimate quantitatively the accumulation rate that you would expect for Δage = 10,000 years. In the original manuscript we stated that we preferred not to do this because it requires extrapolating the firn model beyond its empirical calibration range. The mechanics of bubble trapping in very slowly accumulating firn are poorly known, which is why we hesitate to push the firn model to such extremes. Nevertheless, we will report a cautious estimate in the revised manuscript. Using the Herron-Langway densification model and the barometric equation, we computed the expected Δage and

$\delta^{15}N$ for a range of accumulation rates and temperatures. For $\Delta age$ = 10,000 years and $\delta^{15}N$ = 0.08 ‰, we estimate the accumulation rate to be between $\sim$ 0.05-2 cm/yr ice equivalent, conservatively. We note that this estimate depends strongly on the height of the convective zone, which is unknown. A deep convective zone would drive $\delta^{15}N$ to lower values, consistent with the low $\delta^{15}N$ that we measured in the -380 m MT core as well as the new 5/4 BID cores.

4. Reorganize text

All reviewers recommended reorganizing the main text, and we intend to follow their suggestions. We will reorganize the text into this outline, consistent with referee 1's comments.

1. Introduction
2. Field site and analytical methods
    a. presentation of field site
    b. description of measurements and methods (including new table with metadata to present more clearly which measurements were performed on which cores)
        i. Taylor Glacier
        ii. Taylor Dome
    c. analytical uncertainties
3. Age models
    a. Taylor Glacier
        i. justification of tie point choices
        ii. age model uncertainties
    b. Taylor Dome
        i. justification of tie point choices
        ii. age model uncertainties
4. Results
    a. $\Delta age$
5. Discussion
    a. implications of high $\Delta age$ and how it relates to the previous work on Taylor Dome that suggested steep accumulation gradients

5. -380 m core chronology

Referee 1 and 3 commented on how we dated the -380 m core. We agree that the chronology of the -380 m core is more uncertain than the BID cores because there are fewer data points to match features in the gas records. The core also appears to cover the period 59-68 ka where the variability in the gases is relatively small (besides the large rise in $CH_4$ and $CO_2$ at the onset of DO16/17). In the original manuscript we value-matched $CH_4$ to create the gas age timescale for the -380 m core, but we did not list the tie points explicitly like we did for the Taylor Glacier 5/4 BID cores and the Taylor Dome core. We also did not describe the dating sufficiently. In the revised manuscript, we will explicitly list the tie points in a new table (Table 5). We will also plot the -380 m core data in a separate figure that shows how it compares to data from the TG 5/4 BID cores and the reference records on AICC2012 (Figure 7 below); we think it will simplify Figure 3 to not include the -380 m data and keep it less cluttered.

We will also describe and justify the tie point choices for the -380 m core more clearly in the main text. The discussion of the -380 m core, including justification of the tie points, will be moved to its own sub-heading in the main text. We will clearly state why we date and interpret the -380 m core: it is evidence of stratigraphic continuity between the new 5/4 BID cores and the Main Transect in the Taylor Glacier blue ice area. We intend to revise the text so that the exact chronology of the -380 m core is deemphasized; we present it as a plausible interpretation. The important part of the -380 m core is that it is *generally* of late MIS4 and MIS 4/3 transition age, which is robust because the $\delta^{18}O_{atm}$, $CO_2$, and $CH_4$ all change at the same time, consistent with the variations that occurred during the MIS 4/3 transition in other ice cores. Because the $\delta^{15}N$ is similarly low as in the 5/4 BID cores at this time, it suggests that the ice from both sites came

from the same accumulation region and that there are not different deposition zones sourcing Taylor Glacier ice at different times.

Table 5 – Tie points relating -380 m Main Transect core depth to gas age on the AICC 2012 timescale.

| Depth (m) | Gas Age (ka) | Parameter/ Data Source | Feature Description | Tie Point Source |
|---|---|---|---|---|
| 3.751 | 59.53 | $CH_4$/ this study | High value at start of DO16/17, synch. to EDML $CH_4$ | This study |
| 5.301 | 59.83 | $CH_4$/ this study | Low before DO16/17, synch. to EDML $CH_4$ | This study |
| 9.929 | 64.40 | $CH_4$ /this study | Low after DO18, synch. to EDML $CH_4$ | This study |
| 14.849 | 66.00 | $CH_4$/ this study | Low before DO18, synch. to EDML $CH_4$ | This study |

Figure 5: -380 m core from the Main Transect.

[Figure]

Baggenstos, D., Severinghaus, J. P., Mulvaney, R., McConnell, J. R., Sigl, M., Maselli, O., Petit, J. R., Grente, B., and Steig, E. J.: A Horizontal Ice Core From Taylor Glacier, Its Implications for Antarctic Climate History, and an Improved Taylor Dome Ice Core Time Scale, Paleoceanogr. Paleoclimatology, 33, 778-794, 10.1029/2017pa003297, 2018.

Indermuhle, A., Monnin, E., Stauffer, B., Stocker, T. F., and Wahlen, M.: Atmospheric CO2 concentration from 60 to 20 kyr BP from the Taylor Dome ice core, Antarctica, Geophysical Research Letters, 27, 735-738, 10.1029/1999gl010960, 2000.

Mayewski, P. A., Twickler, M. S., Whitlow, S. I., Meeker, L. D., Yang, Q., Thomas, J., Kreutz, K., Grootes, P. M., Morse, D. L., Steig, E. J., Waddington, E. D., Saltzman, E. S., Whung, P. Y., and Taylor, K. C.: Climate change during the last deglaciation in Antarctica, Science, 272, 1636-1638, 10.1126/science.272.5268.1636, 1996.

Schilt, A., Baumgartner, M., Schwander, J., Buiron, D., Capron, E., Chappellaz, J., Loulergue, L., Schupbach, S., Spahni, R., Fischer, H., and Stocker, T. F.: Atmospheric nitrous oxide during the last 140,000 years, Earth Planet. Sci. Lett., 300, 33-43, 10.1016/j.epsl.2010.09.027, 2010.

---

## Author Response (AR1)

**Response to Referee #1**

General comments: I would strongly advise to reorganize the paper in separated sections for more clarity. The way it is now, you continuously go back and forth between the sites and methods. I would suggest the following organization: Introduction, Field sites and analytical methods (–> presentation of your sites and the measured data in the field and in the lab + analytical uncertainties), Age models (-> choice of tie-points and chronological uncertainty propagation for both TG and TD), Results -> Δage and Discussion. Your manuscript would gain in clarity and would guide the reader toward your results and interpretations. You should avoid the listing of sites and data in the text and instead propose tables summarizing the data/sites information you need. This is particularly true for the blue ice sites you cite in the text and the different measurements performed on your cores.

We reorganized the text of the paper following Referee 1's comment. The new manuscript is organized as follows:

Abstract

1 Introduction

2 Field site and methods
        2.1 Field site
        2.2 Core retrieval
        2.3 Analytical methods
        2.4 Analytical uncertainties

3 Age models
        3.1 Taylor Glacier MIS 5/4 blue ice drill cores
        3.2 Taylor Glacier -380 m Main Transect core
        3.3 Taylor Dome
        3.4 Age model uncertainties

4 Results
        4.1 Δage
        4.2 Δage uncertainties

5 Discussion

6 Conclusions

We added a table that summarizes all metadata concerning the measurements. It is now clearer which measurements were made on which cores, at which institutions, field versus laboratory, and continuous versus discrete.

Please find responses to specific comments below.

**Specific comments & Technical corrections:**

 ABSTRACT:

- line 24: "Taylor Glacier (Antarctica)"
We added "(Antarctica)."

- line 27: "low SNOW accumulation WITHIN the Taylor…"

We added "snow" and change "at the Taylor Glacier accumulation zone" to "within the Taylor Glacier accumulation zone."

We changed "Taylor Dome" to "this area."

INTRODUCTION:

We added references for "paleoarchive of the Earth's past atmospheric composition" - (Bauska et al., 2017; Petrenko et al., 2017; Schilt et al., 2014).

We changed "age of ice and air bubbles always increases with depth" to "age of ice and air bubbles generally increases with depth."

The reference to "distance" here simply refers to any generic reference point from which distance is measured in a blue ice area. In the case of Taylor Glacier, distance is measured from a flag that marks the location of the Main Transect. The flag was originally placed at an arbitrary location along the transect, and it ensures continuity between different sampling efforts during different seasons. E.g. -58 m is always 58 m south of the flag.

These details are described by (Baggenstos et al., 2017), and we added this reference here.

We removed "with fast access to age information."

We added "as precise as the aforementioned methods." We changed "have" to "present." We also deleted "providing" for better readability.

We do not think it is appropriate to add a table describing various blue ice areas because the paper is not a review of blue ice areas. There is already a published review of Antarctic blue ice areas that we cited in the original manuscript (Bintanja, 1999). We simply wished to point out that there are several blue ice areas that have been studied, however we will follow editorial guidance on this issue.

We replaced "expands" with "extends." We prefer not to replace "by developing ice and gas chronologies spanning the MIS 5/4 transition" with "back to" because "back to" implies that the archive is continuous back to the 5/4 transition, which it is not.

The relevant sentence in the original manuscript is, "In 2015 a new ice core was retrieved approximately 1 km down-glacier from the 'Main Transect,' the across-flow transect containing ice from Termination 1 through MIS 3 (Baggenstos et al., 2017) (Figure 1)."

We prefer not to remove "the across-flow transect" from this sentence because we think it is important to define what the Main Transect is and to note its orientation with respect to the glacier flow explicitly.

-Line 34-36: "paleoarchive FROM TAYLOR GLACIER, where it was previously thought to be absent". Remove "larger context of". Replace "into" by "within". Replace "at Taylor Dome" by "of this region"
We changed this to read "the description of a new MIS 4 paleoarchive from Taylor Glacier, where it was previously thought to be absent." We removed "larger context of." We replaced "into" with "within." We replaced "Taylor Dome" with "of this region."

FIELD SITE AND ANALYTICAL METHODS:

Page 3:
-line 5: if you are not citing an acronym, ice sheet is written without capital letters
We changed "Ice Sheet" to "ice sheet."

-line 6: "northERN"
We changed "north" to "northern."

-line 7: "ice EQUIVALENT accumulation"
We changed "ice accumulation" to "ice equivalent accumulation."

-line 15: "80 km LONG ablation zone", and you already said it in the previous paragraph
We removed "~ 80 km."

-line 16: need rewording. I suggest the following: "Water stable isotopes obtained from an along-flow transect just below the equilibrium line"… "revealed uncontinuous ice covering the last glacial period" –
We changed the sentence to read "Water stable isotope data obtained from an along-flow transect from just below the equilibrium line to the terminus revealed ice from the last glacial period outcropping at sporadic places along the transect."

line 20: "revealed continuous records of ice from the Holocene to the last ice age, with ice of the last interglacial and older found…" references for this statement?
We included references for this statement. They are (Schilt et al., 2014), (Bauska et al., 2016), (Baggenstos et al., 2017), and (Buizert et al., 2014).

-line 22: "the most COMMONLY USED archive" instead of utilized
We changed "utilized" to "commonly used."

-line 27: Reference for the previous ice core study. Where was taken this new ice core compared to the previous study? Need more precision. What was the sampling problem with the previous record?
There is not a previous ice core study per se. We have worked on Taylor Glacier for over 5 years, and prior to the 2014-2015 field season the MIS 5/4 transition was thought to be missing from the glacier archive. Then in 2014-2015 we found the MIS 5/4 transition in a new location that was previously not sampled. The new location is 1 km down glacier from the Main Transect, which we note on line 34 and show in Figure 2a.

We referenced relevant discussion in (Baggenstos, 2015) to clarify.

-lines 30-33: need a reference
It is unclear what the referee wants referenced. We added the reference Rhodes et al. 2015 in case it is the CH4 variability at DO16/17. If referee 1 meant the results from the -380 m PICO core there is not a reference because those data are unpublished until this manuscript.

-line 36: replace "in" by "of"…"CH4 variations similar to those ASSOCIATED WITH DO19" or "corresponding to"
We replaced "in" by "of." We replaced "similar to those at Dansgaard-Oeschger event 19" to "similar to those associated with Dansgaard-Oeschger event 19."

-line 37: "CH4 CONCENTRATION increase"
We added "concentration."

Page 4:
-line 3: replace "work" by "analysis", replace "spanning" by "section"
We replaced "work" with "analysis." We replaced "spanning" with "section."

-line 4: need rewording, proposition: "…CH4 and CO2 concentrations, which confirmed the MIS 4/5 transition record in the gas phase"
We changed line 4 to read, "sampled for laboratory analyses of CH4 and CO2 concentrations, which confirmed the MIS 5/4 transition record in the gas phase."

-line 6: spanning not properly used
We stated that the 0-9m and 17-19.8m sections were sampled, instead of using the word "spanning."

-lines 7-11: it would help to make a table for all the analyses performed on the different cores, with specification of the proxy measures, where, the time coverage of samples (or portion of core) and the method used for measurements, analytical uncertainty…
Other referees requested a similar table. We included a table that summarizes the metadata for the analyses discussed in the manuscript including where the samples were taken, which coring device was used (BID or PICO), and in which laboratory and what type of measurements were made.

-line 24: "resulted in a good agreement of our measurements with other…"
We changed the sentence to read, "resulted in a good agreement between our measurements and other Antarctic CH4 records."

Page 5:
-line 19: "… on archived Taylor Dome ICE CORE samples…"
We changed the sentence to read, "Discrete measurements of CH4 and CO2 were made at OSU on archived Taylor Dome ice core samples…"

-line 22: "(~10g OF ICE, …)"
We added "of ice" after "~ 10 g."

Page 6:
-line 10: 'CO2 CONCENTRATION decrease", again later
We changed the sentences to read, "CO2 concentration decrease" and "CO2 concentration increase."

-line 11: remove "and"
We removed "and."

-line 13: value of the offset?
We stated the value of the offset. It is ~ 13 ppm at 61.5 ka.

-line 15: need rewording, a proposition: "…younger ages. Therefore, we refrain from further align the CO2 rises together for better consistency."
We reworded the sentence to read, "…younger ages. However, $CO_2$ offsets between ice cores are observed (Luthi et al., 2008), and we cannot reject the possibility that the offsets are real. Therefore we refrain from value-matching the $CO_2$ rise.

-line 17: why not use the d18Oatm of Vostok or TALDICE instead of NGRIP? You would have a complete record over your period of interest on AICC2012, but potentially with a lower resolution.
Referee 4 had a similar comment. Deep TALDICE d18Oatm is unpublished, Vostok d18Oatm is low resolution, and the two records do not agree precisely in terms of when the light excursion begins at the MIS 4/3 transition. Siple Dome d18Oatm would be the best choice, but the new Seltzer timescale does not

extend beyond 50,000 years ago and the old timescale is not synced with AICC2012 (Seltzer et al., 2017). We think it is beyond the scope of this paper to sync Siple Dome to AICC2012, and we are aware of other work already in progress towards this goal. NGRIP is helpful because it provides variability to match where CH4 variations are small, and it is consistent with AICC2012 (which is the timescale that we use to tie to EDML CH4). We note that the EDML d18Oatm shows quite good agreement with NGRIP d18Oatm in terms of the onset of the MIS 5/4 excursion. Since EDML is lower resolution than NRIP, we still pick tie points using the NGRIP d18Oatm, however we show the EDML agreement in our revised Figure 3 below. (Capron et al., 2010; Landais et al., 2007)

-line 23: replace "has" by presents"
We changed "Taylor Glacier d18Oice has more variability…" to "Taylor Glacier d18Oice is more variable…"

-paragraph 3: I am not very much convinced by value matching for dating. We do not really understand the usefulness of the -380 core data until the idea of similar firn conditions. This and the following argument are important for your interpretation later. This paragraph needs rewording.
The reviewer points out that paragraph 3 on page 6 is poorly worded because the purpose of the -380 m core is not clear from the beginning. We think the -380 m core is useful because it shows similar trends in the gas data (CO2, d18Oatm, and CH4) as the MIS 5/4 cores from ~ 1 km down glacier. This suggests stratigraphic continuity between the Main Transect (where the -380 m core was drilled and where all previous work on Taylor Glacier has been conducted) and the new MIS 5/4 drill site. The d15N-N2 is similarly low in the -380 m core as in the 5/4 cores. The implication of this is that the archive of ice found at the Main Transect likely originated from the same accumulation zone as the 5/4 cores. In other words, the Taylor Glacier ablation zone is not a confounding mixture of ice that has flowed from different deposition areas at different times. Rather, the archive appears to be a stratigraphically continuous and intact record with a common source deposition zone.

We reworded the paragraph so that readers understand this point clearly and know the purpose of the -380 m core at the beginning of the paragraph.

The reviewer was also not convinced that our tie point choices for the -380 m core were robust. In the original manuscript we value-matched the -380 m CH$_4$ data because the data are sparse and we lack the context of a longer record to confidently match the beginning and ending of transitions and features like for the MIS 5/4 cores. However, we recognize that value-matching cannot provide unique ages for the -380 m core, especially before the MIS 4/3 transition where the variability in the gas records is small (i.e. one could assign different ages to a given depth). We intend to rewrite paragraph 3 on page 6 to de-emphasize the dating of the -380 m core, as it was not our intention to develop a robust chronology for that core. We think the important thing is that the -380 m core contains gas bubbles that span the MIS 4/3 transition and some of late MIS 4. We would like to emphasize that CO2, d18Oatm, and CH4 are all changing across the MIS 4/3 interval in the -380 m core, similar to in the new MIS 5/4 cores, and to find variability in all three of those parameters that is synchronous and of the right magnitude is unique. Thus we think assigning the age of the -380 m core broadly to the MIS 4/3 transition and late MIS 4 is robust, even if the exact chronology is uncertain.

In the revised text we de-emphasize the exact dating of the -380 m core, present the tie points we chose more clearly in a table, and display the -380 m data in a new figure so that it is not cluttered with the new MIS 5/4 BID data. We also emphasize the purpose of interpreting it – to show evidence for stratigraphic continuity between the MIS 5/4 drill site and the Main Transect, which implies that the source accumulation zone for the Taylor Glacier ice archive was the same through time. We think we are justified interpreting the -380 m core this way without necessarily improving the certainty of the -380 m chronology.

Page 7:
-paragraph 2: not useful, could be removed.
One puzzle that has emerged from our work at Taylor Glacier is why the MIS 4 dusty ice was so elusive to find, whereas the LGM dusty ice is clearly represented and is even visible at the surface. Paragraph 2

addresses this problem and offers an explanation - that the MIS 4 ice is quite thin. We prefer to keep this paragraph, but we moved it to the results section and emphasized the usefulness of the paragraph at the beginning.

**3.2 ANALYTICAL AND AGE MODEL UNCERTAINTIES**

Page 7:
-line 18: "is likely"
We changed this to read, "The mismatch between field and laboratory CH4 in the top 0-4 m of the core is likely due to…"

-lines 19-22: not clear. You say that you consider the 2015-2016 data as uncontaminated, but as the same record differ from the lab, in the end you do not interpret the data… but you did later in the text… Moreover, you did not discuss the reasons that could explain why the records are so different. I would possibly keep the tuning, but associate it with a much larger uncertainty than the other points due to the mismatch with the lab data. Then only use the CH4 data in grey area for dating purposes and no more.
Other referees had similar comments about the 0-4 m CH4 data and our choice to tie the field data to AICC 2012. We stated why we think the laboratory and field records are different – it is likely because resealed cracks in the glacier surface affected the CH4 in the lab samples but not the field samples. These kinds of cracks tend to penetrate the top 4 m of ice (line 19) at Taylor Glacier, and CH4 measurements in the 0-4 m surface ice have looked wrong in the past, so this is not a new observation (Baggenstos, 2015).

We assigned larger uncertainty in this section. We also emphasized how we do not interpret the top 4 m rigorously. We explained and rationalized in the text more clearly what we chose to do. We only presented the CH4 data from 0-4 m for completeness, and the delta age in the 0-4 m is not critical for our interpretations (the high delta age values occur at ~ 5.5 m). There are no CO2 or d18Oatm data from the 0-4 m section to interpret, and the delta age in 0-4 m section has very little bearing on the overall story we present. Thus we think it is justified to offer our best plausible gas age scale for 0-4 m, clearly show the discrepancy between the laboratory and the field data, and state that the 0-4 m section could be contaminated but that it will not be used in our interpretations that follow.

The discussion about the analytical uncertainty should be following the presentation of the analytical methods.

We reorganized the text so that the uncertainty discussion comes after the analytical methods, similar to comments from other referees.

–paragraph 4: I am not convinced about your argument for the confirmation of data. From the looks of the data presented on Figure 2, I would say that your choice of markers is not convincing, I would have chosen differently… From your Figure 3, I understand that your choices were made in order to align together the records you cite as confirming your alignment (e.g. nssCa). I would recommend to change the way you presented your figure 2 to make the reader see by himself why you choose these tie-points and not others. You should focus more on this aspect, which is the base of your discussion later, it would strengthen your work. Not necessarily in the main text, it could be an appendix.
Though referee 1 would have chosen tie points differently, he or she did not say exactly how. Thus it is difficult to defend our tie point choices specifically to this referee's criticism. Generally speaking, in the revision we provided stronger justification for the tie point choices we prefer. Other reviewers also asked for information like this.

Specifically we made figures clearer so that readers can see easily why we picked certain tie points. We added more tie point justification and moved it to the supplementary information. Here the original Figure 2 showing our tie point matches is expanded into Figures S1 and S2 so that readers can more easily see what features we matched.

We revised our final tie point choices. These are summarized in the preceding summary document, but the main changes from the original manuscript include: (1) 6 new nssCa tie points that match variability

between TG nssCa and EDC nssCa, (2) only 3 particle count tie points matching TG particle counts with EDC laser dust (instead of the original 9), and (3) 2 additional d18Oice tie points that match variability in TG water isotopes with EDC water isotopes. We opted to include more nssCa tie points instead of particle count tie points because the nssCa data are more quantitative, we can compare to EDC nssCa (a like-like comparison) instead of comparing insoluble particle counts to EDC laser dust (different measurements), and the nssCa record is less noisy than the particle count record. We hope that the addition of two more d18Oice tie points helps readers see the similarity in the d18Oice variability at TG and EDC for AIM 19 (72.5 ka) and AIM 20 (76 ka).

The gas tie points between CH4 have not changed substantially from the original manuscript. The two tie points that match TG d18Oatm to NGRIP d18Oatm have changed slightly based on feedback from reviewers. The oldest one linking 19.8 m to 74.65 ka now ties 19.27 m to 73.74 ka in order to tie the lowest measured d18Oatm to the local minimum in the NGRIP record. The other d18Oatm tie point was shifted to tie to the midpoint of the MIS 5/4 transition in NGRIP.

The tie points and the final match are shown in our revised Figure 3 (above).

Page 8:
-lines 3-4: "20 cm = 300 years", based on what? Which chronology?
The relevant part of the sentence in question is: "there is a 10 cm offset between the continuous field CH4 and the discrete laboratory CH4, and a 20 cm offset between the continuous laboratory CH4 and the discrete laboratory CH4. 20 cm depth uncertainty equates to, conservatively, 300 years on the gas age scale near the onset of Dansgaard-Oeschger event 19." Here we were estimating the age error associated with depth offsets between the cores, the largest of which was 20 cm at DO 19. We believe it is clear when we say "on the gas age scale" that we are using our chronology.

Referee 4 pointed out that our conservative estimate was not conservative enough. We think referee 4 was actually misreading the axes of Fig 3B, but we did realize upon closer inspection that the slope of the gas age-depth curve in its steepest segment is 20.8 yr/ cm. So our conservative estimate of the effect of a 20 cm depth offset is ~ 420 years for the gas age scale. We changed "300" to "420" in the text and propagate the uncertainty to the chronologies and the delta age calculations.

Lines 5-8: This is not a proper argument. If you say that both CH4 data from TG and EDML are similarly smoothed in the firn column, you are implying that they have similar firn conditions (i.e. accumulation rates, firn depth…). Is it the case?

Our statement is based on the observation that the magnitude of the changes in CH4 and CO2 concentration are similar in TG and EDML. It appears that the CH4 signal in TG is more smoothed than EDML at DO 18 (65 ka), but this is the only place in the record where the magnitude of the changes is substantially different. It makes sense that the amount of smoothing in the firn would be the increasing between 60-70 ka where delta age is increasing and accumulation is presumably decreasing. There are no abrupt events in the gases during this interval besides DO 18, so we must use this as our metric for estimating the smoothing. EDML CH4 increases to ~ 515 ppb while TG only reaches ~ 475 ppb. Of course the peak CH4 at DO 18 is only defined by one data point at EDML, but if we assume it is correct then the maximum CH4 concentration recorded during DO18 at TG is 40 ppb lower than that at EDML. We also note the differences between d18Oatm at TG versus NGRIP during the same interval. Therefore we think the firn smoothing must be different in the two cores during this interval, and we think it is likely due to increasing the height of the lock-in zone consistent with delta age increasing to extremely high values as accumulation decreased.

Our initial statement was meant to reflect how the cores generally agree in terms of the magnitude of smoothing across the whole record, but we neglected to explore the larger discrepancies near DO18 that are likely due to firn smoothing.

We changed what we wrote in the paper to more accurately reflect our assessment of the degree of smoothing at DO 18. We don't think the smoothing is significantly contributing to the uncertainty in our tie point selection.

-lines8-9: Analytical noise... why is that? What is the measurement uncertainty of your method?
We deleted this sentence when rewriting the analytical uncertainty section. We provided a value in Table 1.

-lines 9-12: Please, when using a chronology as reference, make sure of the uncertainty values you cite... What you wrote is not correct. The AICC2012 chronology uncertainty over your period of interest (i.e. ~65-74 ka) at EDML is ranging between 1500 years and 1400 years (cf. supplementary material of Veres et al., 2013). The values you have indicated correspond to the uncertainty of the ice and gas chronology at the orbital scale, prior to the last interglacial. -Following all this discussion of uncertainties, what are the uncertainties associated with your ice and gas chronologies for TG? You never gave a value and I do not see them on your figures. The same for your revised TD chronology.
The reviewer notes an error in the original manuscript – we cited the wrong absolute age uncertainty associated with the AICC2012, which we use as a reference scale for dating the gas and ice records in the new Taylor Glacier cores. The reviewer pointed out the 1-sigma uncertainty in EDML is actually 1400-1500 years between 74-65 ka and can be found in the supplementary material of (Veres et al., 2013), but the supplementary material only gives the uncertainty in the ice age chronology for 74-65 ka. The main text gives the uncertainty in the gas age chronology for EDML (figure 2 in Veres et al. 2013), which is also ~ 1500 years. For the ice chronological uncertainty, the uncertainty in EDC should be considered instead of EDML because we tie our dust data exclusively to EDC to obtain the ice age scale. The 1-sigma uncertainty for the EDC ice age scale is ~1800-2500 years for the time period 74-65 ka.

We corrected the uncertainty we cited for the AICC2012 reference age scale to 1500 years for the gas phase and 2500 years (taking the maximum) for the ice phase so that it is consistent with the information in (Veres et al., 2013).

In general we presented the uncertainty estimation in the revised manuscript similarly to the original manuscript – i.e., each tie point we picked for Taylor Glacier and Taylor Dome is assigned a maximum and minimum age to estimate the uncertainty of the match, and these estimated uncertainties are propagated through the chronology by interpolating between the maximum and minimum ages at each tie point. The age range at each tie point is assigned by considering (1) the resolution of the data for a given feature that we matched, (2) the analytical uncertainty of the data that we matched to, and (3) how robust (or possibly ambiguous) the matched feature was (i.e. could we be matching the wrong feature?).

In the revised text we explicitly display the errors along with the age models (shading in Figure 5A and Figure 5C below). The uncertainty range is also included in the delta age calculation (shading in Figure 5B and Figure 5D below).

3.3 ΔAGE AND COMPARISON TO TAYLOR DOME

Page 8:
-line 7: Temperature and accumulation are not the only factors influencing Δage. All factors acting on the firnification process do as they impact on the firn depth variability. What about insolation of wind stress affecting the snow metamorphism into ice?
We do not understand what referee 1 means by "insolation of wind stress." If he/she means wind stress, we did not include this because we think the effects on delta age are secondary. If he/she means insolation, then we also did not include this because insolation effects on delta age are also secondary. Temperature and accumulation are the primary controls on delta age. We are unaware of firn densification models that include wind stress or insolation with major influence on firn evolution. If insolation and wind stress affect delta age, we think they are of secondary importance to temperature and accumulation.

-line 19: remove "on the order of hundreds of years", it is given by the lower limits just before. Change "smaller" in "smallest"
We removed "on the order of hundreds of years," and we changed "smaller" to "smallest."

-line 21: replace "at" by "for"
We replaced "at" with "for."

-lines 26-27: ok for the two sources of uncertainties, but you forgot to take into account the absolute uncertainty of the ice and gas chronologies. You have ~1500 years uncertainty from the AICC2012 age scale, consequently the uncertainty of your new chronology should be around ~1600 years for the gas age, and ~1530 years for the ice age (I took one random range from your choice of tie-points). Then your maximum and minimum Δage should be obtained from the (ice age - 1sigma)-(gas age + 1sigma) and (ice age + 1sigma)-(gas age - 1sigma). You should give an approximate value of the Δage uncertainty for the reader to have an idea of the significance of your Δage values later.

We accounted for the uncertainty in delta age in the original manuscript by propagating our tie point uncertainty (described above) using the calculation that the reviewer describes here. We did not propagate the absolute uncertainty from the reference age scale, but we note that the actual uncertainty in delta age acquired from the reference age scale should be much less than the total propagated uncertainty from EDML (1500 years) and EDC (2500 years) because these uncertainties are correlated in depth. I.e., it is unlikely for one to be too old while the other is too young. Because the uncertainty estimates that we placed on our tie points are very generous, we think that we already estimate a reasonable uncertainty for delta age (between ~ 2000 years, Figure 5). This uncertainty range compares well with the uncertainty cited in (Baggenstos et al., 2018).

-line 30: "10 ka"+/- ??? uncertainty needed.
We stated our uncertainty more clearly in the text.

-line 33: then why is it so different? Replace "high" by "large"
We explain the difference in terms of accumulation gradients in the discussion section of the text. We replaced "high" with "large."

-line 34: now you talk of the influence of wind, but not before…
We talk about wind in terms of scouring, or removal of snow. Not in terms of influencing the snow grain metamorphism, which we think is of secondary importance in the firn densification. Wind scouring works to reduce net accumulation. Whenever we write "accumulation" in this manuscript, we are referencing the combined or net effect of addition of snow by precipitation AND removal of snow by wind scouring.

Page 9:
-paragraph 1: I do not think that the last sentence is necessary, you should delete it.
The sentence in question is, "Note that a **Δ** age of 0 or less is physically impossible, and minimum **Δ** age <= 0 in Figure 4 is merely an artifact of estimating the error for individual tie points generously and without the physical constraint that ice age > gas age." We deleted the sentence.

-paragraph 2: You should gather together in one section the chronology construction for your two sites, with the proper calculation of their respective uncertainties.
We reorganized the text so that there is an Age Models section describing the construction of the Taylor Glacier 5/4 BID cores and Taylor Dome chronologies (and the -380 m Main Transect core).

-line 13: "in the same manner AS described"
We added "as."

-lines 17-19: You should then directly give a 0 value. Note then the uncertainty associated to the Δage is then not guassian..
We simply give 0 values where the minimum error estimation causes the negative delta age artifact. No, the uncertainty is not Gaussian. It is not possible to assign a Gaussian error to our tie points given our methods.

-line 21: Δage of 2.5 ka, but p8 line 20 you cited an extrema value of 12 ka with reference to Baggentos et al., in review… why are the values so different?

This is explained in the Discussion section of the text. The accumulation gradient switches at the LGM relative to MIS4.

-lines 22-25: I disagree with this statement. It comes too soon. For TG, not located on a dome, ice thinning and ice flow are very important factors that could affect the depth-age relationship. For TG you cannot interpret directly your variations on Δage in terms of accumulation. To distinguish between the major influences of thinning and accumulation, you need an ice flow model. If your ice flow model indicate that there are no significant thinning variations, then and only then you can interpret it in terms of accumulation. Moreover, you give absolutely no justification for your favour toward accumulation changes, and you do not explain why you disregarded the thinning influence.

The statement in question is "The implication of the relatively 'normal' delta age is that accumulation at Taylor Dome did not dramatically change at the onset of the last glacial period or throughout MIS4 as Taylor Glacier did. Comparing the depth-age relationships in the new Taylor Glacier core versus the Taylor Dome ice core highlights the difference in accumulation between the two sites."

We are confused by referee 1's comment. If he/she means that thinning of the ice could affect delta age, then we disagree. Ice thinning can affect the slope of the depth-age relationship, but it cannot affect the stratigraphic order of bubbles and ice at depth, i.e. the Δage = ice age-gas age at any given depth will remain constant with any degree of thinning. In the revised manuscript we reference (Parrenin et al., 2012) and point out that delta depth (the difference in depth between ice and gas of the same age) can evolve with time due to thinning and glacier flow, but delta age is fixed when gas diffusion effectively ceases at the lock-in depth.

If referee 1 means thinning of *firn* at the original deposition site, then we agree that in an extreme case this could affect the delta age because the process occurs before bubble close off. However, in order to achieve a delta age of 10,000 years you would have to thin the firn such that 10,000 annual layers of snow were included in the firn pack before bubble close off. For example, if a typical delta age in east Antarctica is ~ 3000 years, then this means thinning firn to ~30% its thickness, which seems outside the realm of possibility even on the flank of a dome.

We do see how referee 1 takes issue with the second part of the relevant statement - interpreting the depth-age relationship strictly in terms of accumulation changes without considering thinning. We simply meant to state that the depth-age relationship supports our interpretation of the high delta age values. We changed the wording of this part of the paper and acknowledge that thinning due to glacier flow could be the cause of the observed depth-age relationship.

-last paragraph: you should give the modern values of accumulation measured at these two sites. It would give an idea of how much your prior assumption of all differences are due to accumulation changes is valid for modern times.
Modern accumulation rates at Taylor Dome were determined by (Morse et al., 1999), and a good illustration of a steep gradient in the modern accumulation across a 30 km north-south transect on Taylor dome is shown in (Morse et al., 2007). Accumulation changes along the gradient from 14 cm/yr to 2 cm/ yr.

(Kavanaugh and Cuffey, 2009; Kavanaugh et al., 2009a; Kavanaugh et al., 2009b) describe the modern accumulation rate in the Taylor Glacier catchment, which is informed by the accumulation gradient reported in (Morse et al., 2007). Taylor Glacier is estimated to receive 3-5 cm/yr. (Kavanaugh et al., 2009b) also reports the fact that Taylor Glacier is in a rain shadow and is much drier than the regional average, with references to (Morse et al., 2007; Morse et al., 1998).

We included this information and references in the discussion section where the accumulation gradient is discussed.

Page 10:

-lines 14-18: give values for the LGM reconstructed accumulation at both TD and the virtual sites. This gradient is reverse from yours. Why do you use it then? The useful result from this study to you is only "the opposite accumulation gradient (decreasing from south to north) for ice older than 60ka".
We include it because it is interesting to us that it shifted between the two time periods. It expands on a storyline in the literature that is related to the errors in the original TD age model. (Morse et al., 1998) first predicted the shift in storm gradients based on radar data, and we find it interesting that our delta age data support this.

-lines 18-26: bring nothing more, just show support for the LGM gradient that is different from yours. I would advise to remove these sentences.
This sentence becomes even more important given referee 1's prior comments about thinning. The authors of (Morse et al., 1998) rejected the notion of differential flow (i.e. thinning) because the layer thicknesses did not vary in the same way with depth.

-last paragraph: remove the first two sentences, you are only rewording your results.
We removed the first two sentences.

Page 11:
-line 4: need a reference for this statement.
We added a reference (Hall et al., 2015)

-paragraph 2: the MIS 4 gradient is similar to modern conditions. Are modern conditions in agreement with your proposed hypothesis?
Yes, though there are no data available to constrain the modern delta age at the probable deposition site for our samples.

FIGURES & TABLES:

Figure 1: I would advise to change the organization: a-Antarctica map, b-landsat imagery, simplified map of TG.
We changed the organization of figure 1 according to referee 1's suggestion.

Figure 2: The way the data are presented now, one can strongly argue your chosen tuning points. The scales are two small to see the consistency between the associated variability. I am not at all convinced about your tie-point between the d18Oice of EDC and TG, records present different variability. I would advise to remove from the legend the last two sentences. -Tables 1&2: You should add some indications on your figure 2, on the reference records, to directly make the link between the tables and your chosen points (e.g. DO19…). In Table 2 legend, remove the sentence "Ice phase…"
Unfortunately referee 1 did not state how he or she believes that one can strongly argue against the chosen tie points, which makes it difficult to rebut this point specifically. In the revision we further justify the d18Oice tie points along with other tie points used to construct the chronologies. We added labels for important features to ease comparison between the graphical display of tie points and the list of tie points in Table 1, e.g. "DO 19" or "DO 18". We also added figures graphically displaying the Taylor Dome tie points, analogous to the original Figure 2, which only showed Taylor Glacier tie points. Generally speaking, we justified our tie point choices more clearly in the supplementary information of the text.

Figure 3: I would say that there is absolutely no point in plotting together records that were tuned together, or if you really want to, it should be in an appendix. You already use some other untunned records to validate your chronologies. I would leave here only 1 gas, 1 ice records, and then the (b) part of the figure. You should extend the lines for the identification of MIS limits to the bottom of the figure for more clarity. In the legend your last sentence is not necessary, you could delete it.
We think displaying the tuned records helps the reader to see the variability we were matching in Figure 2 and shows how the data between the tie points agree. Showing the matched data in this way is common practice. The figure also puts the environmental records we are discussing in context.

Figure 4: Same comments as for Figure 3. Your should keep consistent the colours of curves from one figure to another. Why didn't you remove the three points in questions and simply state it in the measurement section?

Same responses for Figure 3. We kept the colors consistent between the two figures.

Why didn't you remove the three points in questions and simply state it in the measurement section?

We want to keep the three data points for completeness. It particularly aids readers who are using the same data set, or want to verify that the data set is similar to his/her own copy of the data.

**Response to Referee #2**

**1- SUMMARY AND GENERAL COMMENTS:**
The study by J. Menking and collaborators presents three new ice cores from the Taylor Glacier Blue ice area that they combine to provide the first "composite" ice core record from this location that covers the transition between Marine isotopic Stage (MIS) 5 an MIS 4 (~74 to 65 ka). The chronology for the air trapped in the ice is defined based on the analysis of the global atmospheric tracers CH4 and atmospheric d18O of O2 (d18Oatm) and their synchronisation with well-dated CH4 and d18Oatm records from other Antarctic ice cores. The ice age scale is defined mostly based on the ice dust content synchronisation, again with other well-dated Antarctic dust profiles. From these two ice and gas age scales, they infer the evolution of the age difference between ice and gas at the same depth – the so-called delta age – through this MIS5-MIS4 climatic transition. Substantial delta age changes are observed through time over this time interval i.e. with values from ~2000-3000 years at ~ 74 ka and approaching ~ 10 000 years at ~ 60 ka. The authors also provide a new evaluation of the delta age evolution throughout the same period in the Taylor Dome ice core (located south of the glacier), which suggests no significant delta age changes for this site. The authors attribute these contrasting delta age evolutions between the two sites to a steep accumulation gradient across Taylor Dome that intensified across the transition from MIS 5 to MIS 4.

This paper presents a study that will be of great interest for the ice core community and to the extended paleoclimate community. It is thus well within the scope of Climate of the Past. Overall the manuscript is well written and presents substantial new material and interesting interpretation of the results. However several aspects of the paper need improvements and clarifications and thus I believe that major revisions are needed before it can be considered for publication.

My first major comment is related to the fact that the authors interpret the differences in the delta age evolutions between the Taylor Glacier area and the Taylor Dome ice core site almost exclusively in term of a change in the accumulation gradient between the two areas. While this could be an acceptable interpretation, they absolutely need to build a much stronger case regarding why this is their favoured one (e.g. versus ice thinning) and thus provide a much more elaborated discussion of their new results. But also, they should discuss the other possible controlling factors; in particular, those are commonly identified as impacting the firnification processes e.g. the role of surface temperature vs accumulation rate vs ice impurity content have already been discussed over the past few years (e.g. Bréant et al. 2017, Capron et al. 2013; Hörhold et al. 2012). I believe that a summary of the current knowledge (and knowledge gaps) regarding the climate and environmental factors that impact changes in delta age would be useful. In particular, it would be of added value to further mention firn densification models that provide an alternative method to estimate delta age. At the moment the authors only acknowledge the Herron and Langway model (1980) although several other models building on this original work have been developed in the more recent years (e.g. Goujon et al. 2003) and more recent development in Bréant et al. 2017, dynamical version of Herron and Langway (1980) used in e.g. Buizert et al. 2015).

The role of surface temperature was discounted in our initial interpretation because the differences in delta age between Taylor Glacier and Taylor Dome are so large, but the accumulation sites are quite close to each other and likely to not differ in surface temperature history very much. Accumulation seems much more likely to vary between the two sites, particularly given the previous work by (Morse et al., 1998), cited in our manuscript, showing different layer thicknesses across the dome. This interpretation is consistent also with the notion that accumulation has a greater control on delta age than temperature does. We think ice impurity content likely has a secondary effect compared to accumulation. We have a measure of impurity content in the particle count data and Ca concentrations. Particle count and Ca begin to rise at 7.5 m depth (moving up core), but delta age has already begun rising in non dusty ice at 11.5 m depth – so impurities do not seem to be driving delta age to first order.

We do agree that a summary of the factors controlling delta age would be appropriate, and in revision we added more text that references other factors controlling delta age.

The reviewer also mentions thinning. We believe the reviewer is suggesting that thinning due to flow from the dome to the sample site would somehow impact the age difference between gas and ice. Referee 1 made

a similar point, to which we responded in detail. While thinning obviously could impact the depth difference between coeval points in the gas and ice phase, we do not see that it affects delta age because it does not disrupt the stratigraphic order of bubbles in relation to the ice matrix that encloses them. Our depth-age relationships are determined independently for the gas and ice phases, thus we make no assumption about accumulation to determine delta age. If we did assume accumulation rate to get delta age, and if we had assumed constant thinning for both Taylor Dome and Taylor Glacier, thinning could have been an issue.

My second major comment is related to the form of the paper. First I believe that some reorganizations of some sections are necessary and I detail this in the next section. Second, I think that the Figures 2, 3 and 4 need to be revised so that the readers are able to better visualized the different records that are being presented but also so they better support the results and the proposed interpretation. More details are provided in the next section of the review.

See detailed comments below where these issues arise.

Additional comments are also provided in the following and I would strongly advice the authors to consider them when preparing a revised version of their manuscript.

See detailed comments below.

**2- SPECIFIC COMMENTS:**
- Section 2 (Field site and analytical methods) is not always easy to follow, in particular regarding which type of measurements has been performed on which core and where (on site or in labs back in the USA). I would suggest the authors to propose a summary table in the revised manuscript that detail clearly this information.

We added a table (Table 1) that details the metadata for all measurements made – i.e. which core, which measurement, at which institution, and in the field or lab.

- The authors propose to treat the three ice cores covering the MIS5-4 transition as a single ice core record (unified depth and age scales). While I agree with them that it is justified, I believe that they should provide additional details on how they line up the different records together (and possibly provide a specific figure?) and discuss the attached uncertainties that arise from proceeding as such on the resulting "composite" record.

The cores are not "aligned" in depth, per se. They were drilled adjacent to one another, so we assume that, e.g., 10.0 m depth in the 2014-2015 core is the same as 10.0 m depth in the 2015-2016 core. There was no shifting or stretching the depth scales to make the records match better between different cores. The only problem that leads to errors in the depth scales is irregular angle breaks at the ends of individual blue ice drill cores that were not properly aligned in the field immediately after recovery. This could theoretically lead to depth offsets of no more than 20 cm between cores as most angle breaks are < 10 cm. Our view is that the effect of depth offsets is visible in the comparison of the discrete CH4 records from the 2014-2015 core versus the field CH4, where you see up to a 10 cm depth offset between records at DO 19. 10cm conservatively equates to 210 years on our age model where age changes the most with depth. The continuous CH4 measured at DRI versus in the field (same 2015-2016 core) actually exhibit larger offsets (up to 20 cm = 420 years on our age scale), likely from errors in the depth logging or again from angle breaks that cause depth offsets between sticks cut from the same core for field versus lab continuous flow analysis. Since this is the largest depth offset observed, we think this sufficiently estimates (and probably overestimates) the error due to depth offsets. Thus we propagated 20cm = 420 years error into our gas age calculations. A similar estimate was made for the ice age scale.

This is explained more clearly in the revised text. We added a paragraph in the age model uncertainties section that elaborates on the treatment of depth uncertainties. We also display the propagated error on the depth-age plots, not just the delta age (as in Figure 5 below).

- Section 3.1 is hard to follow, the authors should consider restructuring it such as 1) they present how the ice age scale has been defined and then 2) as the gas age scale has been defined. Regarding the definition of the tie points based on the alignment of the dust record, I find that some of them are quite ambiguous considering the number of spikes present in the TG records. For instance why would they assign the tie point at 73.6 ka to the spike at 12 m rather than the spike at 9 m? I believe that the authors have a good reason for doing so, however, it should be spelt out more explicitly. It is necessary that the figure be much enlarged to allow a detailed inspection of the records.

We appreciate Referee #2's suggestion to restructure section 3.1. We reorganized the text such that the 'Age Models' section comes before Results and Discussion. In this section ice age and the gas age models are explained in separate paragraphs. We also moved the explanation of the revised Taylor Dome age scales to the 'Age Models' section.

Regarding the tie points based on aligning the dust records – we feel these tie points are justified because they produce the best overall match between the Taylor Glacier dust and water isotope records with EDC. We explored a large number of alternate strategies, which did not perform as well. For example, the specific tie point questioned by the reviewer (12 m versus 9 m) is best justified with the d18Oice data. If the dust is matched at 9m, the correlation between Taylor Glacier and EDC d18Oice deteriorates substantially because of mismatches in the variability around AIM 19 and AIM 20. The uniqueness of the d18Oice and dust records together justifies the tie point.

We justify our tie point choices more clearly in the supplementary information of the main text. We also eliminated the tie point to dust completely and instead chose 2 new tie points from the d18Oice record so that readers clearly see which variability we are matching instead of potentially ambiguous variations in nssCa. We think matching directly to d18Oice instead of using d18Oice as justification for a possibly more ambiguous nssCa match makes a stronger case for the age model in this section of the core.

- I do not think that the analytical uncertainties should be discussed after the determination of the age model. The authors should consider adding a brief description of each dataset after the analytical method descriptions and there, add details regarding their specificity and limitations.

We restructured our discussion of analytical uncertainties such that they are discussed following the discussion of analytical methods. The total uncertainties in our age models are discussed after the age models are discussed.

- It is a little strange that the presentation of the new measurements on the Taylor Dome ice core and the definition of the new age scale and for Taylor Dome are currently presented as part of the discussion. Why not instead presenting the new age model of Taylor Dome as an additional sub-section in the age model section that is currently only dedicated to the dating of the Taylor Glacier ice? And similarly for the new measurements, they should be also included in the analytical description section and information should be also added in the table I propose to add in the revised manuscript. Also, I think it would be very useful that more background information is provided regarding the Taylor Dome site, in particular regarding the previous age scales available for this time interval.

We reorganized the text so that analytical methods and uncertainties come before the Age Models section. The age model section is divided into 3.1 Taylor Glacier MIS 5/4 cores, 3.2 Taylor Glacier -380 m Main Transect core, and 3.3 Taylor Dome so that each age model we developed is discussed thoroughly.

Metadata about the new Taylor Dome measurements were included in Table 1, and text was included in the analytical methods section about the methods used for Taylor Dome samples.

3- FIGURES
- I appreciate the effort of the authors to show how they defined the different tie points to link between the Taylor Glacier records on a depth scale the dated reference records. However, it should be bigger to allow a closer inspection of the different records and where the tie points have been chosen.

We split Figure 2 into Figures S1 and S2, which are now included in the supplementary information. The figures are larger so that the tie point picks are more clearly visible. We think this will make the picks more readily justified now that closer inspection is possible.

- Figures 3 and 4 should appear much bigger. Also, to facilitate the comparison of delta age evolutions between Taylor Glacier and Taylor Dome, I suggest to remove the panels b from each figure and combine these panels b into a single and additional figure. They can be presented in parallel, making sure that the scale used for the delta age evolution is the same for both sites.

We enlarged figures 3 and 4. We reorganized the panels so that the "b" panels are now plotted together in one separate figure for easier comparison (Figure 5 above).

**4- STYLISTIC, TYPOGRAPHICAL COMMENTS AND MINOR COMMENTS**

P2, L16: You should also mention the work that has been done in the Patriot Hills blue ice area e.g. Fogwill et al. (Scientific Reports 2017).

We included the Patriot Hills work in our list of blue ice areas.

P2, L34: I find the expression "MIS 4 paleoarchive" to be an awkward formulation; I would suggest to reformulate the sentence e.g. "(2) the description of a new climatic record from Taylor Glacier across MIS 4".

We changed the sentence to read, "(2) the description of a new climatic record from Taylor Glacier across MIS 4."

P4, L1: "second exploratory core": this is a bit confusion to say "secondary" since the PICO core was also referred to as a "secondary exploratory core". It should be rephrased e.g. "During the same 2014-2015, another exploratory core was obtained directly : : :.".

We changed the sentence to read, "… another core was obtained directly…"

P4, L5: Again the numbering of the core is confusing (as in total, as far as I understand, four cores were drilled with only the last three having MIS5/4 transition ice). Hence it would be could to reformulate such as e.g. "In the 2015-2016, an additional core was drilled: : :".

We listed the various cores and which measurements were made on which core (Table 1).

P5, L26: The authors should be more specific in the title of the section e.g. "Determination of the ice age and gas age scales".

We reorganized the text so there is an Age Models section, which we renamed "Determination of Age Models."

P6, L4: "minimal" please be more quantitative here and give a quantitative range at least.

We eliminated this text so the comment is no longer relevant.

P8, L11: Although you refer to the tables, the authors should also provide at least a quantitative range regarding the relative age uncertainties.

We reworded how we assessed the uncertainty of the age models. We plotted the uncertainties along with the depth-age curves, and we provided the mean uncertainty along the cores.

**Response to Referee #3**

The manuscript presents the initial multi-tracer dating of recent large size ice cores from Taylor Glacier (TG), covering a period of about 25 ka around the MIS 4/5 transition, as well as new data aiming at improving the gas chronology of the Taylor Dome (TD) ice core during the same period. Such characterization of a blue ice field providing large amounts of ancient ice is certainly of interest for the paleoclimate community and well within the scope of Climate of the Past. The results are discussed in terms of age difference between the gas and ice phases (delta age) and related varying accumulation rates. This interpretation involves some assumptions and simplifications that are not enough described in my view. For example, a number of age synchronization tie points appear ambiguous to me and the remaining discrepancies between records are not sufficiently commented. The inferred very low accumulations are likely to imply erosion periods, and the impacts of the ice-flow (thinning, hiatuses, possible folding etc.) should be better considered. Even if firn modeling with somewhat empirical models well outside the calibration range of their parameters is not compulsory, the physical processes controlling delta age and $d^{15}N$ fractionation should be better described.

Overall I think that major revisions are needed in order to better discuss the approximations made (e.g. ignored firn and ice physics), describe the consequences of alternative assumptions on ambiguous chronological tie points for multi-species consistency, age scales and delta age. I think that the paper should be more focused on an in depth discussion of the ice cores dating and dating issues, and less focused on somewhat spectacular but uncertain conclusions on delta age and accumulation. A number of suggestions are provided below.

We thank Referee #3 for helpful comments.

As discussed in the response to other reviewers, we addressed the perceived ambiguity of tie point selection by justifying them with more extensive discussion in a supplement. We enlarged the figures so that it is easier to see why we chose to match variations the way we did. We made it more clear why alternative tie point selections produce poorer matches with EDC records by picking more tie points from other datasets (e.g., d18Oice). A specific example of a possibly ambiguous tie point choice was brought to our attention by reviewer 2 and reviewer 4 – why assign the peak in dust at 73.6 ka to 12 m instead of 9 m? The reason is that if we assign the 9 m peak to 73.6 ka then the d18Oice is shifted such that the minimum between AIM 19 and AIM 20 no longer aligns with the EDC d18O record. The correlation between d18Oice EDC and d18Oice TG gets worse due to stretching the TG AIM 19 peak by several thousand years. This way we also would not align the nssCa peak at 73.6 ka (there is no nssCa variability in our record at 9 m).

A second possible ambiguity is the dust peak at 15 m. In our set of tie points we do not align this peak, so we tried two alternatives to align it to variations in EDC nssCa. If we align it with the EDC nssCa peak at ~77 ka (1) this stretches the d18Oice record out such that the signal no longer matches EDC d18Oice at AIM 20, (2) the nssCa variability in TG doesn't really match the variability seen at 77 ka in EDC, and (3) the delta age gets unreasonably high (we expect accumulation to be higher in stage 5 versus stage 4 due to warmer average temperatures and thus delta age to be relatively lower than during stage 4). We also explored aligning the 15 m dust peak with the EDC variability at ~73.6 ka, but again this causes a mismatch the d18Oice in EDC at AIM 19/20.

A third possible ambiguity is in the dust peaks between 0-1 m. We could align the large dust peak at 0.3 m to the nssCa peak in EDC at ~ 64 ka. This shifts other aligned dust peaks back in time - i.e. the peak at 1.25 m aligns with a very small dust peak at 65 ka and seems out of place, and the three particle count peaks between 1-3 m depth do not have corresponding 3 laser dust peaks to align with. Instead one peak has to be skipped. We prefer to align the particle count peak at 0.3 m with the smaller EDC peak at 61.5 ka because we observe that background particle count appears to be decreasing toward shallower depths (see minima in the particle count record between peaks) similar to how EDC nssCa and laser dust decrease between from 64 ka to about 60 ka. However, we recognize this interpretation puts the 0.3 m nssCa peak at a place on the AICC 2012 age scale where EDC nss Ca has no corresponding peak (see figure below). We have contacted the original authors of the data in question, and no logs of contamination or processing errors exist for these depths in EDC. The existence of the laser dust peak without a corresponding nssCa peak is

as of yet unexplained (Fischer, H. and Lambert, F. personal communication). It is possible the dust captured in EDC at 61.5 ka had very little Ca while the dust at TG did.

The nssCa mismatch we are describing above is shown in this figure (black arrows show particle count peak and corresponding laser dust peak (red) on AICC2012, the blue trace has a nssCa peak without a corresponding peak in EDC (brown)).

[Figure]

One plausible alternative for the 0.3 m tie point is to shift it to older ages, which causes a mismatch in the rest of the data and increases the delta age estimate by 2.5 ka.

Another plausible alternative is that the "stray" nssCa peak in the Taylor Glacier record is from local, wind-blown $Ca^{2+}$ dust and is not representative of a larger-scale Antarctic dust event. The peak occurs in the top 30 cm of the ice core where dust data have been rejected previously ((Baggenstos et al., 2018) rejected top 40 cm) due to contamination of vertical cracks by local wind-blown dust.

We discussed our justification for tying the 0.3 m dust peak, but we also emphasize that we do not interpret the age scale in the top 40 cm rigorously, similar to (Baggenstos et al., 2018).

Because we are discussing the shallow part of the core here, we think it is appropriate to inform the editor about a mistake we made in the presentation of data in the original manuscript. We cut off the top meter of the TG records in the original Figure 3. This is why there are tie points for ice as young as 61.5 ka but no data that young in the original Figure 3. We did this for the gas data because there is clearly CH4 contamination up to 1200 ppb in the 0-1m section of our cores (which appeared in all measurements, no disagreement between DRI and field CH4). We suspect this is due to snow machine oil/ exhaust at the drill site.

We show all data in the revised manuscript and revised figures for completeness, but the ice records shallower than 40 cm and the gas records shallower than 4 m are not be interpreted rigorously. This is described and justified clearly in the text.

Regarding interpretations of high values of delta age – we noted in our response to reviewer 2 that while differential ice thinning would affect the depth-age relationships, it would have no effect on delta age because thinning does not disrupt the stratigraphic relationship between ice and gas bubbles at depth. The reviewer also referred to hiatuses in accumulation. We think an accumulation hiatus is in line with (if an extreme example of) how we are currently interpreting the high delta age values – i.e. high delta ages correspond to low accumulation rates. We did not explicitly discuss what our records would look like if a

complete cessation of accumulation occurred. We included text that discusses how the records might look if accumulation hiatuses occurred. A hiatus, if it did occur, is most likely in the section 60-64 ka where CH4 is flat, d18Oatm variability is small, and full MIS 4 conditions are underway with extremely cold temperatures and low accumulation at the TG catchment. Because our record does contain the complete CO2 rise for the MIS 4/3 transition, we think there is good reason to believe there is no significant hiatus in bubble trapping. 60-64 ka on the gas age scale corresponds to 68-71 ka on the ice age scale, where there is still clear variability in our particle count and nssCa records. We think this is further proof that there is not a hiatus in accumulation. We will include these justifications in the text. Regarding folding, there is no evidence of folding in the records we developed, which would show up as reversals in gas and ice phase records as compared to known trends from other ice cores. We see no reversals in our gas records and ice phase records. For the sake of demonstrating our thinking - one might for example question whether the CO2 variability at AIM 19 is in fact two limbs of a fold with its center at the CO2 peak. Looking at CH4, d18Oatm, nssCa, and insoluble particle count tracers on depth axes rules out the possibility that the ice is folded because the records are not identical on both sides of the hypothesized fold axis. The same can be said even where the gas records are relatively flat – e.g. between 8.5-10.5 m depth when gas concentrations are relatively low, d18Oatm is relatively enriched, and there is little variability. Here there is also little variability in the nssCa and insoluble particle count records to resolve the problem. We note in this ambiguous section that the d18Oatm is steadily becoming more enriched and d15N is becoming steadily more depleted with no evidence that the trends reverse, as you would expect if the ice were folded there.

We included more information in our discussion on the physical parameters controlling delta age and d15N.

**Specific comments**

p2 l34-35 and p3 l26-28: Missing MIS 4 and MIS 4/5 transition in previous TG records. The authors should provide references and introduce more the possibility of having different hiatuses in different TG ice cores. The ice flow in the area should be better illustrated, for example Figure 1 (a) could be further zoomed on the drill sites and some flow line directions could be provided.

The missing MIS 4 is explicitly discussed in (Baggenstos et al., 2017), which we added as a reference for relevant discussion.

It is unlikely that there are different accumulation hiatuses in the different ice cores presented in this work, if that is what the referee means. The 5/4 BID cores as well as the PICO auger exploratory core were drilled within ~ 1 m of one another and so must have traveled down glacier as a unit from the same accumulation area. Even the -380 m core on the Main Transect, which is < 1 km from the location where the 5/4 BID cores and PICO core were drilled, came from the same accumulation zone as the 5/4 BID cores (and all other TG stratigraphic units) without experiencing a hiatus or any sort of prolonged difference in accumulation relative to the 5/4 cores. Our point here is that the whole accumulation zone sourcing the TG ice archive would have experienced accumulation hiatuses at the same time, broadly speaking. We recognize it is possible for a glacier accumulation zone to have small-scale heterogeneity in accumulation rate either due to differences in precipitation rate or due to different magnitudes of wind scouring. We think this kind of variability would not affect the 5/4 BID cores or the PICO exploratory core because they were obtained so close to one another, but it is conceivable that prolonged heterogeneity in the accumulation zone caused discrepancies between the -380 m Main Transect core and the 5/4 cores. However we observe that the -380 m core d15N values are quite comparable to those measured in the 2015-2016 5/4 BID core. We interpret this as evidence that the two cores came from firn columns with similar characteristics, implying that the accumulation zone was more or less the same for both the Main Transect and the 5/4 drill site. In other words, the stratigraphy is continuous between the two drill sites. In case this is unclear or seems weak due to the arguable dating of the -380 m core, we are basically saying that the d15N/CH4, d15N/CO2, and d15N/d18Oatm ratios are the same in the -380m core as in the 5/4 BID core, supporting the conclusion that the stratigraphy on Taylor Glacier is continuous and that different hiatuses in accumulation, or even different accumulation zone sources altogether, were unlikely.

We added general direction of flow lines to the simplified map in Figure 1c.

p2 l37: a reference should be provided for the previous TD chronology

The original chronology st9810 (Steig et al., 1998) was based on CH4 matching to GISP2 and inferring delta age to get the ice chronology. But the ice chronology was incorrect because it assumed accumulation could not be exceptionally low (and thus delta age could not be exceptionally high). The error was pointed out by aligning the TD Ca record to EDC (Mulvaney et al., 2000). The TD gas chronology was updated by synchronization to the Vostok GT4 timescale (Barnola et al., 1991; Petit et al., 1999) that extends back to ~ 68 ka (Indermuhle et al., 2000). A full chronology (gas and ice) was most recently updated by (Baggenstos et al., 2018) back to 60 ka.

We adopt tie points from (Baggenstos et al., 2018) where our age scales overlap. We include the aforementioned references and a summary of the previous TD chronology in the revised manuscript.

p3 l14-16: a reference should be provided for these site characteristics

We added references for the sublimation and flow rates in lines 14-16 on page 3 – (Kavanaugh et al., 2009a; Kavanaugh et al., 2009b).

p3 l25-28: a reference should be provided for the ice flow structure of the "main transect"

We will added references to lines 25-28 on page 3 for the vertical dip of layers on the Main Transect (Baggenstos et al., 2017; Bauska et al., 2016; Petrenko et al., 2017; Petrenko et al., 2016; Schilt et al., 2014).

p3 l30-31: the exact location of the "-380m" drill site (coordinates) should be provided. More site information could be provided (e.g. altitude, mean annual and summer temperatures etc.)

The mean altitude, mean annual, and mean summer temperatures are not different from any other site on Taylor Glacier discussed in the paper. It is a drill site on the Main Transect (Figure 1), 380 m from a flag that marks the center "0 m" on the Main Transect. This information is in the revised paper.

p5 l8-11 and p8 l1-4: the depth offsets, uncertainties and unification method between the different "TG 5/4" cores should be better described.

Referee 2 had a similar question and we repeat our response here. There is not a unification method, per se. Each core was drilled adjacent (within 1 m) to the original borehole drilled with the PICO auger. Each core has a depth scale determined by summing the lengths of individual, meter-long BID cores. We assume, for example, that 15.0 m in one core = 15.0 m in another core. The cores are not "aligned" in the sense that we did not stretch or alter the depth scales to match the data precisely. When you view all measurements on depth there are very small offsets between the records, indicating slight depth offsets, likely due to short angle breaks at core ends that affect the depth summation along the core. We conservatively estimated the effect of these offsets on our age models and propagated them through the delta age calculations. This is discussed in the revised text in the age model uncertainties section.

p5 l17: "The interpretations that follow do not depend on data taken from 0-4 m", and similar statement p7 l22. In Figure 2, the 3 TG CH4 data series are not consistent above 5m depth, and in Figure 3 the CO2 consistency with the composite in the upper part of the TG record mostly rely on the 2 upper points. What would be the consequence of matching the TG CO2 record below 4 or 5 m depth to the composite CO2 record instead of using the CH4 record which is nearly flat between ~4.5 and 7 m depth for multi-species consistency and delta age? In Table 2, two CH4 tie points and half of the ice phase tie points are located well above 4 m depth.

We see why Referee #3 would be suspicious about the data 5m and shallower – the CH4 records depart from one another substantially above 4m with smaller differences between 4-5m depth, and the CO2 appears to date too young relative to the composite data. But what Referee 3 says here is not entirely

correct. The shallowest CO2 measurement is at 4m depth, where the CH4 differences are much smaller, and the CH4 rise evident in both datasets (associated with DO17) is one of the most robust features. The discrepancy in the CO2 depends highly on the tie point at 5.4 m – the "low point before DO 16/17" in table 1. We think this is the most robust gas tie point of the entire set. If we shifted this to younger ages, it would smear the CH4 rise out such that TG CH4 would lead EDML. The next tie point is at DO18. We did not choose other tie points from the CH4 record because the CH4 variability between DO18 and DO17 is minimal, thus any tie points chosen there would be ambiguous. We could choose tie points deliberately from the CO2 record such that the slopes of the CO2 increases are more similar, but we refrained from doing this given that CO2 offsets between different ice cores are a known but relatively poorly understood phenomenon (Luthi et al., 2008). We addressed this in lines 13-16 on page 6. In fact, as an example, there are CO2 offsets between the TG CO2 and the composite record from (Bereiter et al., 2015) of even larger magnitude than at the 4/3 transition during the middle of stage 4 (Figure 3).

The consequence of matching the CO2 would be that the records would be more consistent (value-matched), and delta age would be lower by ~1.5 ka at the most. The uncertainty we estimated for the delta age calculation is already larger than this.

We would like to stress that the parameters in the ice phase (i.e. d18Oice and dust) are only affected by the surface cracks in the top 40 cm, not the entire top 4 m. This is stated in the Table 1 caption, but we will state it more clearly in the main text too. So the ice phase tie points are not an issue except potentially in the top 40 cm.

Referee #3 is correct that two tie points for the gases are chosen above 4m, which is why we shaded those tie points gray in Table 1. Our intention was that those points be interpreted cautiously. The CH4 record shows variability that looks very much like the CH4 variability associated with DO 16/17, hence the temptation to choose tie points and extend the gas chronology to depths shallower than 4m. But the mismatch in CH4 between the DRI and field data sets leaves us unable to reject the possibility that both data sets are wrong < 4m. This wouldn't change the conclusions of the paper because delta age begins to rise at 11.5 m depth in our core, with maximum delta age occurring at ~ 5.5 m. We would like to reemphasize that we do not interpret the gas data shallower than 4 m rigorously and that those data do not inform our interpretations of the high delta age values.

p5 l27-30: In Figure 2, the TG CH4 records look a lot smoother than the EDML record. The dissimilarity of the two signals limits the possibilities of unambiguously synchronizing them. This could be due to different processes such as analytical smoothing (Stowasser et al., 2012), longer gas trapping duration in firn at very low accumulation rates (Spahni et al., 2003; Köhler et al., 2011; Fourteau et al., 2017), gas diffusion through ice (Bereiter et al., 2014 and references therein). This should be discussed, possibly smoothing the EDML record to try to simulate the TG record, comparing with the lower accumulation EDC record etc.

We think analytical noise and firn smoothing in the EDML record are the main reasons for the dissimilarity between the EDML and TG CH4. We would prefer to plot the error bars on the EDML data, which visually help the readers see the smooth atmospheric signal, rather than smooth the data set directly. EDC CH4 looks quite similar in resolution and smoothness to EDML. The relative amplitudes of abrupt CH4 features can be an indication of relative smoothing. EDC, EDML, and TG all have the same magnitude CH4 feature at DO 19. At DO 18 EDML CH4 is higher, followed by EDC CH4, followed by TG CH4. The CH4 rise at DO 16/17 (near the MIS 4/3 transition) is largest in EDML, only slightly smaller in TG, and lowest in EDC. Using this as an indication of smoothing, then the effect in TG is largest at DO 18 and negligible at other times in the record.

We don't think the CFA system is smoothing beyond what the firn has already done to the gas record. The main justification for this is that the discrete CH4 measured in the lab (green dots at DO19 in Fig 2) and CFA CH4 (purple and red lines in Fig 2) agree well.

We included discussion of smoothing in the text including justification of why we don't think smoothing effects are significantly impacting our tie point choices.

p5 l34-35: I did not understand why the _18Oatm record is tied to NGRIP only: a North Hemisphere discontinuous record covering only parts of the studied period. Could other data also be used? (e.g. Petit et al., 1999; Kawamura et al., 2007; Buiron et al., 2011)

The TALDICE and Dome Fuji datasets are unpublished and/or unavailable publicly) to our knowledge, though they appear in figures in the referee's citations. Both are low resolution through the time period of interest, and Vostok d18Oatm is also quite low resolution. To our knowledge the Dome Fuji DFO 2006 age scale is not synchronized to AICC 2012, though Vostok and TALDICE are. We do not think synchronizing to any of the three records helps eliminate ambiguity that CH4 doesn't already solve. Where d18Oatm is helpful is syncing TG to NGRIP in the older part of the gas record where CH4 variability is comparatively smaller but d18Oatm variability is large. Also worth noting here is that NGRIP d18Oatm is relatively high resolution across the 71-76 ka section. The match to NGRIP is further justified by the close agreement with EDML d18Oatm, which we plot in the revised Figure 3 (below).

We justified the synchronization to NGRIP in the text.

p5 l38: some tie points look ambiguous to me and the tie points assignment should be further discussed. For example, the EDC and TG _18Oice records look quite different in Figure 2, thus the _18Oice tie point does not look robust to me. On the dust plot in Figure 2, I do not understand why the small EDC peak at 75.75 ka was tied to the TG particles peak at _12m rather than the one at _9m depth.

We refer back to our response at the beginning of this document following the general comments. The referee here likely made a typo because our 75.75 ka age is aligned with the d18Oice peak at 16.62 m. Thus we assume the referee means the 12.05 m dust tie point that we aligned with 73.58 ka (now updated to 12.20 m and 73.62 ka). We specifically addressed this tie point in the response above, as well as two other ambiguous tie points.

We provided further justification of our tie point selections in the text as already described.

p6 l9-27: Due to the dissimilarities between the records in Figure 2, I believe that it is impossible to unambiguously assign the tie points. Thus I doubt that the choices were made without taking into account the constraints discussed in this section. An overall discussion of the constraints, what led to the current best guess dating and how other assumptions could be (or not) discarded would be most useful.

We agree and will include more discussion of the rationale that led to our tie point choices.

We rewrote the text so that it more accurately reflects how we reasoned through the tie point choices, especially now that the tie point choices have been revised.

p6 l31-32 and Figure 3: I do not understand how the CH4 record from the "-380m" core could be unambiguously tied to AICC2012. On the other hand the CO2 records seem easier to match and matched. The overall dating constraints should be better described.

We edited the text to explain in more detail how we aligned the -380 m core to AICC 2012, including presenting the tie points in a table and discussing our tie point choices in the text. The main revision here is that we deemphasize the exact -380 m core dating, presenting our tie points as a plausible chronology. We explain more clearly why we think it is robust that the -380 m core is roughly late MIS 4 and MIS 4/3 age.

p6 l31 - p7 l14: I did not understand this discussion of the differences between the TG records. The dating of the "-380m" core is presented in one line and the CO2 mismatch with "TG 5/4" not discussed, nor the d18Oatm mismatch with NGRIP at ~66 ka. The lack of information on flow line directions make the direct comparison between TG records difficult to understand, and few references are provided. I suggest to focus more this section on gas scales consistency between the "-380m" and "TG 5/4" cores, and how the CO2 mismatch between the two TG cores in the 60-64ka age range could be explained. Is the ice phase of the "-380m" ice core also dated? Are large _age values also inferred?

We addressed more completely the dating of the -380 m core in the text as well as include a table with tie points. We also moved the -380 m data to a separate figure where we compare the -380 m gas data with those from Taylor Glacier MIS 5/4 cores as well as the reference records on AICC 2012. The CO2 mismatch with TG MIS 5/4 was not discussed in the text, but we now address it explicitly in the age model section. The d18Oatm mismatch with NGRIP was also not discussed, but now it is with reference to offsets at DO 18.

We think the -380 m core implies that there is stratigraphic continuity between the Main Transect and the drill site of the new MIS 5/4 BID cores. We think the exact dating of the -380 m core is unimportant; rather the important part is that the gases appear to be late stage MIS 4, the d15N is similarly low, and the age-depth relationship is similarly steep. This supports the idea that different accumulation zones are not sourcing the Taylor Glacier blue ice area at different times. Instead, Taylor Glacier ice has likely come from the same deposition zone throughout the last ice age. We deemphasize the exact dating of the -380 m gas age scale in the revised paper and instead argue that the methane and CO2 rises and the d18Oatm depletion are roughly what we would expect if the gas age was ~ last MIS 4 and MIS 4/3, implying that there is continuity between the Main Transect records and the new MIS 5/4 records.

Unfortunately there is no ice phase data for -380 m, so we cannot infer delta age.

p7 l8-9: As this paragraph comes just after the section comparing the "-380m" and aggregated "TG 5/4" cores, readers may wonder which one is the new ice core.

We clarify in the text that the "new" ice core is the TG MIS 5/4 core.

p7 l8-13 and p9 l25-31: providing and discussing plots of annual layer thicknesses (based on depth - ice age, depth - gas age relationships at TG and TD) would help understanding the interpretations related to accumulation and thinning variations.

We calculated annual layer thickness, but we do not think it adds any information that is not already visible in the depth-age plots. The annual layer thickness is smallest where the age changes the most with depth.

p7 l24 - p8 l12: This discussion of uncertainties should appear earlier in the article and be more detailed (see also above comments on p5 l27-30, p5 l38, p6 l9-27).

Other referees also suggested this. We moved the analytical uncertainty discussion to the Field Site and Analytical Methods section. We more thoroughly discussed the relevant uncertainties.

p8 l1-4: this is not consistent with p5 l11. Due to the strongly varying depth-age gradients on Figure 3 (b), the overall largest age bias related to depth offset/uncertainty should be mentioned.

Correct, we updated the text to say 20 cm depth uncertainty instead of 10 cm.

p8 l5-8: the smoothing due to gas trapping duration most likely dominates the diffusive smoothing in the open pores of the firn. It is accumulation rate dependent (e.g. Spahni et al., 2003; Köhler et al., 2011; Fourteau et al., 2017) and thus likely different at EDML and TG. In Figure 2, the TG CH4 record looks much smoother than the EDML record. It would thus be interesting to discuss the gas trapping duration consistent with the firn sinking speed due to the estimated accumulation rates (time needed by the firn to sink by a few meters).

We agree that the smoothing is probably somewhat different between EDML and TG, but not so different that it affects our tie point choices or age model significantly. See our comment above about smoothing in EDML, EDC, and TG, particularly the part about the amplitude of CH4 variability at DO 18. The smoothness of the TG CH4 record versus EDML is due less to the gas trapping process and more to (1) the different analytical methods employed – continuous measurements in TG (and thus some degree of smoothing in TG, though small compared to smoothing in firn), versus discrete measurements in EDML, and (2) higher analytical noise in EDML (the TG continuous CH4 is within the EDML CH4 error bars at all

parts of the records).

We agree it would be interesting to estimate gas-trapping duration, but we do not have a robust estimate of accumulation rate given that the d15N and delta ages are well outside of the calibration range of firn models. We might estimate accumulation rate given the depth-age plots if we knew the thinning function, but we do not know the thinning function and think it is unwise to trust fundamental thinning approximations given the archive's unconventional path to the drill site.

We more clearly demonstrate the analytical noise in EDML by plotting the error bars on Figures 2 and 4 (and S1 and S3).

p8 l16-18 and l35-36: a much more in depth presentation of firn processes influencing delta age, _depth and the physics of _15N should be provided. The consistency between a very large _age and a very shallow firn (_15N indication) should be commented.

In response to this as well as other referees' comments, we provided more references to delta age studies, and we acknowledge the other processes influencing delta age as well as the fractionation of 15N in the firn column. We more clearly elucidated the correlation between large delta age and shallow firn with quantitative estimates of accumulation rate, close-off depth, and diffusive column height.

p8 l24-30: the example of the successive datings of the Taylor Dome ice core, well discussed in Baggenstos et al. (2018) could be used as a base for a more realistic uncertainty discussion.
We are unsure what the reviewer is suggesting here. If the reviewer is referring to the evolution of the Taylor Dome ice core chronology, which was described at length in (Baggenstos et al., 2018), then it is unclear to us how this would be the basis for the discussion of the uncertainty on our delta age calculations discussed on p8 l24-20. Our uncertainty (for both Taylor Glacier and Taylor Dome cores) is based on independently estimating the uncertainty of individual tie points and interpolating the max/min possible chronologies. The evolution of the Taylor Dome timescale was the result of errors in determining the delta age in the Taylor Dome timescale during the LGM. The chronology was revised by dust synchronization (Mulvaney et al., 2000) to obtain a correct ice age scale, and later refined and extended further back in time by (Baggenstos et al., 2018). The successive datings of the Taylor Dome core are useful in understanding the history of the timescale, but they do not provide much useful information about how to estimate more realistic uncertainty for our time period (57-77 ka).

We added discussion of the history/ evolution of the Taylor Dome chronology insofar as it puts our work in context, but we will refer to the discussion in (Baggenstos et al., 2018) in lieu of re-summarizing everything.

p8 l35 - p9 l4: the fact that the physics of _15N (thermal and convection effects) is much more complicated than a pure gravitational effect can't be ignored (e.g. Severinghaus et al., 2001; Severinghaus et al., 2010). The very low _15N values measured in TG ice suggest that either the firn is very thin (an estimate should be provided) or nongravitational effects are important.

We provided an estimate of firn thickness in the text. Based on gravitational effects alone the firn thickness is estimated ~ 15 m, though the height of the convective zone is a major uncertainty in this. A deep convective zone would drive d15N to lower values.

p9 l6-19: the new Taylor Dome age scales presentation repeats methodological information already provided for TG cores but does not discuss the remaining inconsistencies between records and ambiguous tie points. A more in depth discussion of the Taylor Dome age scales should be provided.

We cleaned up the text with respect to repeated methodological information. We discussed Taylor Dome dating in more detail including rationale for the tie points we chose. We discussed in more detail the inconsistencies between the records.

p9 l21-31: this section is unclear to me. If TG and TD ice cores have strongly different _age in the study

period (assuming that the tie points sufficiently constrain the age difference between the gases and ice in a single ice sample), TD can't be the origin site of TG ice even considering differential thinning.

The TG accumulation site is to the north of the TD ice core site, which we showed in Figure 1 and also stated in line 33 page 9. The point we are trying to argue in the paper is that over a small distance, accumulation varied significantly. At the LGM this trend is reversed. This is the interesting implication of the delta age histories.

We described this better in the text so that the point comes across more clearly.

p9 l2-4 and p9 l33 - p10 l26: no accumulation values were derived from the Taylor Glacier record and the discussion is focused on different time periods (present and LGM), thus it could be shortened.

We rewrote this section.

**Technical corrections**

p8 l18-20: smaller _age values were obtained at very high accumulation rate sites such as DE08-2 (40 years, Etheridge et al., 1996)

p11 l24-25: twice "spanning the MIS 5/4 transition"

Here we are referencing the gas age scale separately from the ice age scale. We rewrote this to seem less redundant.

p13 l15: Baggenstos, 2015 (PhD) a web link could be provided.

p13 l23 and in article text: Update reference to Baggenstos et al. (2018), now available as a preprint.

We updated the reference to the published version of the manuscript.

p15 l49: suppress QUATERNARY

We changed QUATERNARY to Quaternary.

p16 l56: uppercase/lowercase issue Figure 2, dust panel: some grey lines are not consistent with the tie points in Table 2 (chronology inversions in some grey lines) Figure 3: the top part of the TG particles count record, including the tie point at 0.31 m depth, is not shown.

We are unsure what the uppercase/lowercase issue is that referee 3 refers to here. The chronology does not actually invert in the dust panel, though we see where the referee is talking about – it appears to invert in the gray lines where the dust begins to rise. We expanded Figure 2 by splitting it into Figures S1 and S2 so generally the tie points are easier to see/ read. This should help readers not only understand why we chose tie points, but also make it clear that the chronology is not inverted.

**Response to Referee #4**

**General comments:** The authors collected and analyzed a set of new ice cores from the Taylor Glacier blue ice area covering the MIS 5 to 4 transition and whole MIS4, and present a suite of data (d18Oice, dust, Ca ion, d18Oatm, CH4, CO2, d15N2). Through age synchronization of gas and ice with other dated ice cores, they find extremely large delta age (ice age - gas age difference), which suggests much reduced accumulation rate at the snow accumulation area for the analyzed core and, by comparing the results with the Taylor Dome ice core data with their revised chronology, give climatic implications of the accumulation contrasts between the Taylor Glacier accumulation area and Taylor Dome.

Regional reconstructions of glaciological and climatological conditions in the glacial period in Antarctica are important for better understanding of the climate system in the Antarctic and its relation with wider areas, and thus the topic of this manuscript is well suited for the Climate of the Past, and the data presented in general seems to be of high quality. First I would like to respect and congratulate the authors for finding the ice from entire MIS4 after the years of fieldworks and high-quality investigations.

I review it mainly in terms of whether the ages, delta age and resulting accumulation rate reduction are reasonably estimated, because they are the basis for the climatic interpretation and conclusions, and also because they have the highest scientific value in this study in my opinion. In doing so I find that the manuscript needs a major revision to make much stronger cases for the extremely increased delta age and reduced accumulation rate (including its timing) at the Taylor Glacier accumulation area, which in turn are based on age synchronization and interpretation of the resulting delta age. In particular, I find it difficult to evaluate the robustness of their choice of the age tie points for some cases from the given text and figures/tables. There are also several tie points, which I did not understand how they could match with the existing ice core records. The authors made poor use of the data (especially d15N) for the discussion of the accumulation rate. While I agree with the authors that the delta age increased and accumulation rate probably decreased in MIS 4, it should be based on much more rigorous considerations. Also, some parts of the manuscript need to be reorganized to better present the field works, ice core samples, measurements and methods, results and discussion. I strongly encourage the authors to improve the article by deeper analyses, interpretation and better presentation of their excellent data.

We thank referee 4 for helpful comments. In general we added stronger justification for the tie points that we chose, including adding supplementary text that explains our reasoning and improving the figures to facilitate the reader being able to see clearly why we chose the tie points the way we did. We strengthened our discussion of the low accumulation rate interpretation and included quantitative estimates. We still think it is not possible to make robust, quantitative estimates of accumulation rate from firn models given that the models are not built or calibrated to describe firn columns where delta age is this high. Nevertheless for the purpose of strengthening our claims/ developing the discussion further, we (1) referenced other controls on delta age, (2) discussed more thoroughly/ quantitatively the controls on d15N including more detailed discussion of why d15N is low at Taylor Glacier, and (3) referenced the Megadunes, Antarctica site as a point of comparison to the Taylor Glacier accumulation zone, especially with respect to the influence of deep air convection on d15N. We reorganized the text following comments from referee 4 as well as referees 1-3. Please see specific answers to the referee's comments below for more details.

**Specific comments:**

**Abstract:** Add description that there are different ice cores, and that how the delta age was estimated ("Dating the ice and air bubbles" is too short even for the abstract). Similarly, "A revised chronology for the Taylor Dome ice core" needs some more explanation. Also, give numbers and error ranges for delta age and accumulation rate ("very low accumulation" is too vague; later in the text it is stated as virtually zero accumulation rate).

We changed "A new ice core" to "New ice cores…" We changed "Dating the ice and air bubbles in the new ice core" to read "We determine chronologies for the ice and air bubbles in the new ice cores by visually matching variations in gas and ice phase tracers to preexisting ice core records. The chronologies

reveal…"

We stated in the text that we are cautious about estimating accumulation rate quantitatively because we recognized that our large delta age value would require accumulation rates that are well below the empirical calibration range of the Herron-Langway firn densification model (the lowest accumulation site in the Herron & Langway paper is Vostok at 2.4 cm/ yr ice equiv.) (Herron and Langway, 1980). To our knowledge there is not a more appropriate model that accurately predicts firn densification under conditions of extremely low accumulation. We maintain the view that extrapolating beyond the empirical range of firn densification models may lead to errors that cast any determined accumulation rate into considerable doubt. Nevertheless we proceeded with caution to determine a conservative maximum accumulation rate for the Taylor Glacier accumulation zone given delta age = 10 ka as determined from our new ice core records. We used Herron and Langway to compute a matrix of density profiles for different temperatures. Assuming the density of snow is 0.36 g/mL and the close-off density is 0.83 g/mL we computed the age of the firn (the delta age) at the close-off depth using Herron and Langway's equation 11 (Herron and Langway, 1980), which we use as an estimate of delta age. A contour plot of delta age on temperature and accumulation axes allowed us to examine the range of temperatures and accumulation rates expected given our independently determined delta age. We then computed the d15N due to gravitational enrichment for a matrix of diffusive zone heights using the barometric equation (Craig et al., 1988). Knowing the estimated close-off depth from the firn model allows us to estimate the height of the convective zone that must bring d15N into agreement with measured values. See supplementary for more details.

We opted not to include the accumulation estimates in the abstract because we don't want to emphasize them as a robust result.

**Introduction:**

P2, L30: "glacial inception" is used differently (it is often used for MIS5e to 5d transition), so perhaps replace it with "major sea level fall" or "major ice sheet growth in the Northern Hemisphere".

We changed "glacial inception" to "ice sheet expansion."

P3, L1-2: "the differences in the ice age-gas age difference". Delete one of the "difference". Field site and analytical methods: The title of the chapter should reflect the fact that it also describes ice cores drilled in different seasons.

We replaced "ice age-gas age differences" with "delta age."

P3, L33 and P4, L1: The phrase "a second exploratory core" appears twice for different cores (PICO and BID cores).

We removed the redundancy.

P3, L35-37: Please clarify if this measurement was done in the field (brown markers in Fig 2), and when and how did you conduct the whole CH4 measurements. It is unclear to me because you mention D/O 19 but not the larger increase at D/O 17 (D/O 17 is only mentioned earlier for the "-380 m" core). Did you obtain all data before you drilled the BID core? Didn't you take the D/O 17 CH4 transition in the 2014-15 PICO core into account for the preliminary age estimate in the field?

We clarified where measurements were done with a table (Table 1). We rewrote this section so that the comment is not relevant anymore.

P4, L3-5: Please clarify which data you mean (Fig. 2, green markers). Also, ice sampling for d18Oatm and d15N is not mentioned here (2014-15 BID core) but there are data points in Fig. 2 that says the measurements were done in two years (2016 and 2017). Please give full explanation about the cores, sampling, measurements and periods for all data you present in a better way (not only about this core; using

We give a more complete explanation of the sampling and measurements by reporting a table (Table 1) that lists each core, whether the core was drilled with the BID or the PICO, which measurements were made including where and when, and the analytical uncertainty of the measurement.

P5, L6-8: Didn't you take any samples to have overlaps with the previous cores? Did you make the sample cuttings for OSU and SIO in the field?

We think here the referee means P4, not P5. No, unfortunately we did not take overlapping samples. The BID cores were cut into quarter cores for SIO and OSU in the field, but the samples were cut at the respective laboratories.

P5, L9-10: The description of the sampling of the 2015-16 core in the laboratory is better placed after describing the core transportation. Overall, the descriptions of field and lab samplings and analyses are scattered so they should be better organized.

We reorganized the paper so there is a core retrieval section that comes before analytical methods, which comes before the results. We think the paper is better organized now.

P5, second paragraph: This part is about field measurement methods so it should come earlier before the first presentation of the relevant data. And, for which core this paragraph's description applies (2015-16 core only)?

See previous comment about paper reorganization. Also see previous comments about Table 1. It is now clear which measurements were made on which cores.

P5, third paragraph: Please clarify which kinds of measurements were made for the different cores (maybe use a table). The measurement methods should come earlier than the first description of the data, or tell the readers that the methods are described later if you introduce the data first (like in the current manuscript).

We now describe all measurements before we present the data. We included a table that details the measurement metadata.

P5, L16: The CH4 field data from 4 - 5 m in the 2014-15 core disagree with the CFA data of 2015-16 core by several tens of ppb, which is much more than your precision and should be discussed as well.

It is not uncommon to have an outlier of several tens of ppb in discrete field measurements, as the precision is much worse than the laboratory analyses due to environmental conditions in the field laboratory and the use of a small, portable instrument. There is not a blank correction applied to these measurements, and there may also be depth offsets from the MIS 5/4 BID cores. These factors likely cause the offsets seen in the top 4-5 m that referee 4 mentions here.

**Results and discussion:**

P5, L36-37: The oldest tie point between the TG and EDC using d18Oice seems unacceptable given the different shapes of the isotopic curves of TG, EDC and EDML cores for this and other periods presented in Fig. 2.

The AIM events we recognize in the Taylor Glacier core are large features that exist in EDC and EDML. We state in the text that we recognize Taylor Glacier d18O is noisier than the records we match to. Nevertheless, the AIM features are unmistakable changes of up to ~ 3‰ that we think represent robust features for tie point selection. Our smoothing of the d18O noise helps identify the peaks and troughs of the features more clearly. These features are important for our record because they provide tie points that importantly resolve ambiguities in the nssCa (because nssCa varies little in the deeper part of the record).

We expanded the axes so that the shapes of the isotope curves are more visible to readers. We added supplementary text clarifying what we see as robust features in the d18O ice record. We also picked tie

points directly from the d18Oice records so that readers clearly see the variability at the AIM events that we are matching in the d18Oice records.

P5, L38 - P6, L1: Some of the dust tie points seem unacceptable or maybe you didn't explain the details of the manual matching. You put one at 12 m but why did you choose that particular one and not other peaks?

Referees 1-3 made similar comments concerning ambiguous tie points, including the tie point at 12 m. We assign the dust peak at 12 m (rather than at 9 m or at 15 m) to the dust peak in EDC at 73.6 ka because this way the AIM 19 and AIM 20 that we identify in the d18Oice line up with the AIM events in EDC. If the peaks at 9 m or 15 m are fit to the dust peak at 73.6 ka instead, the d18O no longer matches.

We justify this in the text in the revised manuscript. We also picked tie points directly from d18Oice to avoid the ambiguity.

Another one at about 6.5 m is described as low point in dust, but the 2015-16 field data (purple in Fig. 2) actually show a peak there (I guessed that orange plot in Fig. 3 is the same as purple in Fig. 2, showing high values at the tie point). The grey lines for the dust (Fig. 2) are drawn between the EDC and TG purple data (orange in Fig.3), but is this particular one connects EDC and TG DRI data instead? Why did you choose the point where the two dust records from the same core disagree?

The peak that appears as a thin purple line in Figure 2 that exceeds the axis limits is either a measurement artifact in the raw data or too small of an event to match to EDC. The raw data shown in Figure 2 are not filtered for outliers, and we prefer to show the full raw data set to demonstrate the data quality. In the ~ 40 cm above this noise there is a real peak (smaller amplitude up to 0.4 ug/g, 6.1 m) that appears to lead the dust rise in EDC in Figure 3.

We understand the confusion because we did not describe our criteria for matching dust peaks. We only fit features that span a range of depths on the order of at least 10's of centimeters and show structure (more than one data point comprising the peak). The peak that exceeds the axis limits is an example of noise because the high dust concentrations span less than 2 mm of ice. The smaller peak centered at 6.1 m that spans ~ 30 cm of ice is a real dust event.

In any case, the new tie point for the period before the MIS 4 onset is 7.75 m, 71.95 ka, chosen from the d18Oice variability. This way the dust ambiguity is avoided altogether.

We expanded the original Figure 2 into Figures S1 and S2 so it is easier for reviewers to see the variability we are matching in the dust records. We also describe in the text what we consider a true feature in the dust versus noise. We also plotted smoothed versions of the d18Oice and particle count records (the two noisiest records from Taylor Glacier) so that the large-scale variations are seen clearly.

Around the one at 70.11 ka, the TG dust peak is offset compared to EDC dust peak (isn't it better not to match the highest point in the peak which have certain width?). Similar examples are at _65.6 and 63.9 ka.

The small adjustments to center the peaks perfectly are not important considering that those differences are well within the errors we place on the ice age scale. Originally we left some of these peaks off center because we did not want to "over fit" the data. However, we understand the value of matching the peaks more perfectly in terms of communicating what we did and convincing readers that our matches are good.

We adjusted the tie points so that the peaks are more centered.

Overall, the lack of details on the matching force me to suspect that you chose the dust tie points while actually checking the resulting chronology by comparing Ca ion data from the two cores (I see that Ca ion data between TG and smoothed EDC compare much better than between the dust records), meaning that Ca is not just used for the validation of blind test (looking only dust) but effectively involved in the tuning. Otherwise, how could you choose the dust tie point at 12 m? In fig. 3, dust data look like bar graph (vertical grey and orange bars) but they should actually be line plots. It is hard to evaluate the match in this figure so

please improve the plots.

We thought the way that this was described in the original text was sufficient, but we see how the reader could be misled by the very good fits between, for example, nssCa but not particle count. We revised the tie point scheme so that more tie points are chosen for all of the records (d18Oice, nssCa, and particle counts) to be more transparent about our tie point choosing process. For the gas tie points we do not see good candidate tie points in the d18Oatm or CO2 that would improve upon the tie points already picked from CH4. Also see the response to other referees above about our hesitation to value-matching CO2.

We revised the ice age scale tie points including more tie points for d18O ice and nssCa in addition to particle counts. We present them clearly in the tables and justify our choices clearly in the supplement text.

From the text (linear interpolation), I think the TG depth-age plot (Fig. 3b) should be straight lines between the tie points, but they don't look like so. A clear example is at an inflection point in the ice chronology at about 72.3 ka, for which there is no ice tie point. There might be my misunderstanding and if so please give full explanation for the interpolation. Please also plot markers at the tie points on the depth-age curves (Fig. 3b). The depth-age and delta-age lines in the figure are too low in resolution (the lines consist of tiny segments of horizontal and vertical lines, like aliasing in low resolution digital images).

Curvature in the age scales in Fig. 2 and Fig. 3 is an illusion because of tie points that are close to one another in depth or because of the low graphic resolution of the figures. We moved the "B" panels in Figure 3 and Figure 4 to their own figure that is easier to see (Figure 5).

You should reject two youngest CH4 tie points. Cracking and contamination should increase the measured CH4 concentration, so those two tie points are probably put on contamination peaks (note large disagreements between purple line, red line and brown markers). Perhaps you can use the peak at 3 m in brown data (if you take only low values in the two of the CFA data you see the same peak, which is uncertain but this could be a true atmospheric peak concentration). The match of d18Oatm records looks somewhat uncertain especially for the older one.

We do not want to match the brown data in Fig 2 because it was measured on a system in the field that is lower precision. That tool is used as a rough guide for determining ages in the field, and those data must always be verified in the lab. Other referees also had issues with tie points/ data in the top 4 meters of our cores.

We emphasize more strongly in the text that we are NOT rigorously interpreting 0-4 m, merely providing a plausible gas chronology for the 0-4 m section based on our view that the CH4 field data are showing the true atmospheric signal.

Why did you connect the oldest d18Oatm data point to the beginning of the d18Oatm enrichment in NGRIP data (why not the second oldest data point in TG d18Oatm which is the highest)? I think the measurement precision is high for the TG dataset, but then I wonder what is the gap at 17m between the 2014-15 and 2015-16 cores. It might suggest depth offset between the two cores. Please discuss.

We changed the tie point to match the second (and most depleted d18Oatm) data point to the NGRIP data. Yes, there is an offset between the 2015-2016 core and the 2014-2015 core d18Oatm (~0.05‰) at 17m. This could imply a depth offset - that the cores are in fact not supposed to overlap there because there is a depth logging error. The measurement precision, which is quite good, seems to suggest this is more likely the case. Unfortunately it is not possible to deduce what the depth error actually is here, but it is worth noting that shifting the red data 20 cm deeper (our estimated depth uncertainty, stated in the paper) would result in a plausible scenario where d18Oatm is decreasing monotonically with depth.

For matching d18Oatm records, why did you only use the NGRIP data as the reference? There are clear discrepancies in the values (probably regardless of the matching quality) for some periods (_73 and 64-49 ka). I think Siple Dome d18Oatm data (Severinghaus et al., 2009) is of higher precision (not only measurement precision but also smaller and smoother thermal fractionation) and resolution, and Siple

Dome and TG were measured in the same lab. Siple Dome also has the data younger than _63 ka where NGRIP data is lacking. So there seem good reasons that you should try using it as well (of course you have to match SD to AICC2012 using CH4). There may be a hope to match around 60-65 using small fluctuations in d18Oatm.

We agree that Siple Dome would be ideal, but the recently published age scale (Seltzer et al., 2017) only extends to 50 ka. We would need to first sync the rest of the age scale to AICC 2012 to be consistent with our record, which we think is outside the scope of this work. We are also aware of other efforts to sync the Siple Dome age scale (Buizert, personal communication) and would thus prefer not to do it. If you look at other d18Oatm records for this time period (TALDICE, Vostok) you will see that they are very low resolution and offer no clear alternative for tying the d18Oatm more robustly than what we have done. You will also see that the offsets between TALDICE and Vostok are of the same magnitude as the offsets between TG and NGRIP. We note deep TALDICE d18Oatm is unpublished.

P6, L1-3: You should take into account the potential age error due to linear interpolation between tie points. The comparison between linear and cubic spline interpolation is insufficient as the demonstration of the age uncertainty between tie points. You should consider using available gas records as much as possible (CO2, d18Oatm; see comments above and below).

We interpret this comment to mean that we should value-match in between tie points where our Taylor Glacier data show differences from the reference records. There is only one section where our records depart significantly from the reference record (CO2 mismatch between 64-60 ka), and we deliberately chose not to value-match the CO2 data given that CO2 offsets between different ice cores are a known and as of yet unresolved issue (Luthi et al., 2008). It is unclear to us where else we might be introducing large errors due to linear interpolation. We think that the uncertainty we estimate with our method (interpolating between maximum and minimum ages at each tie point to generate an oldest and youngest age model) reasonably estimates the uncertainties between tie points (Figure 5 above).

P6, L9-16: Agreement of TG CO2 with existing records is overall very good. However, I think the decision not to use CO2 for the synchronization between 60 and 65 ka is not satisfactory especially because there is no other tie points. You should at least try matching CO2 and look how the resulting chronology look reasonable or not.

If we match the CO2 to the (Bereiter et al., 2015) composite where referee 4 describes (around 60.5 ka), the gas record shifts to older ages by ~ 650 years. The error we stated (in Table 1) is already larger than this, so the case where CO2 is matched is essentially already accounted for if you consider our error range. In this section we would have to value-match the CO2 because there are no robust or obvious inflection points. We prefer not to value-match using CO2 because of the unresolved issue of CO2 offsets between different ice cores, described in the text.

We now draw the errors in the age scale on our plot of ice age and gas age versus depth so that the errors in the chronology are more visible to readers.

P6, L20: See the comment above about the dust and Ca.
P7, L19-22: See comment above about CH4 for 0-4 m.

P7, L31-34: Explanation is insufficient. What do you mean by "surveying the value matched or correlated data"? How exactly did you consider resolution and analytical errors.
In the original manuscript we assigned maximum/ minimum ages to each tie point that estimated the range of possible ages. Our choice of age range for each tie point was based on consideration of (1) the resolution of the data for a given feature that we matched (i.e. do we know the age of a true peak or trough in the data, or is it masked by low resolution?), (2) the analytical uncertainty of the data that we matched to, and (3) how robust (or possibly ambiguous) the matched feature was (i.e. could we be matching the wrong feature?). If any of the three criteria were poor or ambiguous then we enlarged the age uncertainty range to reflect a worse quality match. We then propagated the uncertainties by interpolating through the maximum and minimum age at each tie point, which resulted in an oldest and youngest possible chronology (and also

a maximum and minimum delta age by calculation). We considered calculating a fit index for each tie point and a probability distribution for each match, but this method is more suited for value-matching data whereas we are matching features where multiple parameters are changing at the same time (i.e. peaks and troughs in $d^{18}O_{atm}$ and $CH_4$, or in $nssCa^{2+}$ and particle count). We think that an algorithm will not necessarily do this better than we can do by eye, or at least the difference will be negligible for the delta age interpretations we are making in this manuscript.

We think the uncertainties estimated by the methods described above are justified because (1) even with assigning very generous uncertainty to each tie point, the uncertainty does not affect our interpretations about delta age (i.e., the delta age that we calculate after propagating the uncertainties to our chronologies is still large during MIS 4 and supports the notion of the development of a steep accumulation gradient between the Taylor Dome coring site and the Taylor Glacier accumulation zone), (2) the uncertainty we estimate for delta age is realistic and is similar in magnitude to the uncertainty in delta age from other Antarctic ice cores, including the delta age uncertainties in Baggenstos et al. 2018, and (3) the $CH_4$ record on our new gas age scale matches Hulu speleothem $\delta^{18}O$ very closely at the onset of DO 16/17 and DO 19 (Figure 6 below). The last point supports our choice of tie points for synchronizing to the AICC2012 gas age scale because the Hulu data are independently dated.

In the revised text we explain more clearly how we assigned uncertainty to each tie point, and we justify more clearly why we think the uncertainty is reasonable.

P8, L1-4: The offset of 20 cm is quite large when comparing laboratory measurements by CFA and discrete samples. Please explain the possible causes for this. Why don't you correct the CFA depth assignment by matching the depths of the sharp CH4 features? Why is the 300 yr estimate a conservative one? You have other CH4 tie points where you have much steeper depth-age slope, so it does not sound conservative at all.

The 20 cm depth offset does not have to do with CFA versus discrete samples. We explained the cause of the depth offset in the text – it is because of angle breaks in BID cores that were not aligned and accounted for during drilling. It is possible that there are smaller depth errors due to mistakes in depth logging, but these must be smaller than the offsets introduced due to angle breaks.

The place in the ice core where age is changing the most with depth is where the slope is the shallowest on Figure 3 (age axis is on the bottom). If you compute the age change for 20 cm along this slope, you get 416 years. So the referee is right that we did not estimate the value high enough. We will change the conservative estimate to 420 years and propagate it accordingly.

P8, L9-10: Absolute age uncertainty attached to AICC2012 for this age range is probably incorrectly cited. Please check. Also, it is useful to refer to Chinese speleothem ages (using CH4 and d18Oatm) for the possible range of absolute age error for the studied period.

We cited the age uncertainty from AICC2012 incorrectly. We corrected the absolute uncertainty that we cite to 1 σ = 1500 years for the EDML gas age scale and 1 σ = 2500 years for the EDC ice age scale. A comparison of Taylor Glacier CH4 to Hulu d18O shows very good agreement in terms of the onsets of DO 19 and DO 16/17 (Figure 6, above). Though we necessarily acquire the aforementioned uncertainties when using AICC2012 as our reference age scale, we think that the absolute age uncertainty in our gas age scale is probably less than this given the close match to Hulu. We also note that the relative errors in our ice cores will be less than the total propagated EDC and EDML 1 σ uncertainties because the uncertainties in gas age and ice age are correlated with depth.

P8, L14: It is common to use small 'a' for the term Delta-age (not Delta-Age).

We changed Delta-Age to Delta-age.

P8, L16-18: A better explanation would be that delta-age depends on firn thickness (ice or water equivalent) and accumulation rate, and the firn thickness depends primarily on temperature and accumulation rate.

We think the reviewers will find our revised treatment of delta age more thorough.

P8, L24: I think a weakness of the delta age estimation and whole discussion based on it is that there is no ice and gas age estimates for the same depth, so the uncertainty of delta-age depends on the uncertainties of ice and gas ages between tie points, which is not evaluated well. You should try to have more constraints on the gas age (with CO2 or d18Oatm) between 60 and 69 ka where you have the very large delta-age (which is the basis for your argument of "virtually zero accumulation rate").

We are hesitant to value-match the CO2 data (there is not another way to tie the CO2 given the nature of the variability – inflection points are somewhat unclear/ poorly defined, a ramp-fit algorithm or some other statistical tool would be needed to define them. We also chose not to fit the d18Oatm in this interval given the offsets between TG and NGRIP d18Oatm between 62-68 ka. The most convincing variability in d18Oatm occurs during DO 18 where we already have a more robust CH4 tie point. If you look at other d18Oatm records (e.g. Vostok, TALDICE, or EDC) now plotted in Figure S1 the resolution is too low to match in this range. You will also see that the offsets in d18Oatm between those cores are larger than the offsets between TG and NGRIP/ EDML.

We agree with the referee that it would be better to have ice and gas age tie points for the same depths, but the reality is there are not robust features in the gas and ice phases at all depths.

P8, last paragraph: Discussion here is too qualitative (with the words like "near zero", "where ice accumulates very slowly"). Please be more quantitative by giving possible range for the surface mass balance in the TG accumulation zone in MIS 4 from your data.

See other comments above regarding estimating the accumulation rate quantitatively.

We provided a quantitative estimate in the revised text.

P8, L35-36: d15N is not only controlled by gravitational fractionation. You should introduce it appropriately.

We summarized the d15N controls more completely.

P8, L37 - P9, L1: I agree that d15N likely reflect firn thinning during MIS4, but the change is not linear with respect to delta-age. About half of the d15N change actually occurs at around 71 ka when delta-age is still stable at _4 kyr, and it is in fact before MIS4 (so this should also be discussed in your climatic discussion part). You should definitely discuss this large change while delta-age is small and stable, and in doing so, also run firn models for the high and low d15N around 70-73 ka. You might get some idea on how much non-gravitational signal could be contained in d15N data, or how much accumulation reduction is needed to explain d15N at that change (assuming that the change is purely gravitational). Another exercise is to use the presumed ratio (firn thickness in ice-equivalent) / (real firn thickness), which seems stable over wide range of temperature and accumulation rate (_0.7 from Parrenin et al., 2012, CP) and use the d15N and delta age to estimate accumulation rate, for example at 72, 70 and 61 ka and some times in between. I think you obtain a mm or so for the 10000-yr delta-age with d15N around 61 ka.

We discuss the change in d15N but we did not try to pick apart the gravitational versus non-gravitational effect because we think the uncertainties in accumulation/ firn thickness/ convective zone height, etc. are too large to draw meaningful conclusions.

P9, L18-20: You should actually put the constraint ice age > gas age.

We added the constraint by removing delta age data where it is < 0. This does bring up the issue of what the minimum delta age should be… delta age = 0 is equally impossible, for example. Rather than force delta age data to be an arbitrary minimum when < 0, we simply removed data that is < 0.

P9, L22: "onset of the last glacial period" is confusing as it is often mean the MIS5e-5d transition.

We changed the sentence to read "onset of full MIS 4 glaciation…"

P9, L30-31: Suggesting the accumulation control on the depth-age curve instead of thinning based on the delta-age and d15N while avoiding deeply discussing delta-age uncertainty and d15N is not acceptable (see above).

We developed our discussion of delta age more deeply by summarizing more completely the controls on delta age, by summarizing the meaning of d15N including reference to the barometric equation for computing the height of the diffusive zone, and including more references to previous work on delta age. We still think that accumulation and temperature have the greatest first-order control on delta age, and that an extremely high delta age as found here is very unlikely to occur without extremely low accumulation. When discussing controls on delta age we distinguished which controls we think are secondary (e.g. impurities, wind stress, thinning of firn) versus primary (e.g. temperature and accumulation).

Conclusions P11, L26: The statement "virtually zero net accumulation" needs more solid basis and quantification as commented above. The rest of the discussion (about atmospheric circulation and ice sheet) may change after the revision with deeper look into your data, so I would not review it (and it is not my speciality in any case).

See above comments about accumulation rate estimates. We included our estimate of accumulation rate, and we clearly stated that it is a conservative estimate that is limited by the fact that we lack a firn model to accurately describe very low accumulation conditions.

James Menking 2/18/2019 9:07 AM

James Menking 2/18/2019 9:07 AM

James Menking 2/18/2019 9:07 AM

James Menking 2/18/2019 9:07 AM

James Menking 2/18/2019 9:07 AM

James Menking 2/18/2019 9:07 AM

James Menking 2/18/2019 9:07 AM

[revised manuscript text omitted]

James Menking 2/18/2019 9:07 AM

James Menking 2/18/2019 9:07 AM

James Menking 2/18/2019 9:07 AM

James Menking 2/18/2019 9:07 AM

James Menking 2/18/2019 9:07 AM

James Menking 2/18/2019 9:07 AM

James Menking 2/18/2019 9:07 AM

James Menking 2/18/2019 9:07 AM
Deleted: The spatial pattern of accumulation on Taylor Dome during the last glacial maximum was confirmed by independent age determinations on Taylor Glacier and Taylor Dome ice with Taylor Glacier last glacial maximum Δage = ~ 3000 years and Taylor Dome last glacial maximum Δage = ~12,000 years (Baggenstos et al., 2018, in prep.; Baggenstos, 2015). Therefore two independent lines of evidence support the notion that accumulation practically ceased at Taylor Dome during the extreme aridity of the last glacial maximum. It is thought tha ... [34]

James Menking 2/18/2019 9:07 AM
Moved (insertion) [5]

James Menking 2/18/2019 9:07 AM

James Menking 2/18/2019 9:07 AM

James Menking 2/18/2019 9:07 AM

[revised manuscript text omitted]

Formatted [42]
James Menking 2/18/2019 9:07 AM
James Menking 2/18/2019 9:07 AM
James Menking 2/18/2019 9:07 AM
Formatted Table [44]
James Menking 2/18/2019 9:07 AM
James Menking 2/18/2019 9:07 AM
Inserted Cells [47]
James Menking 2/18/2019 9:07 AM
Formatted [45]
James Menking 2/18/2019 9:07 AM
Deleted Cells [46]
James Menking 2/18/2019 9:07 AM
Inserted Cells [48]
James Menking 2/18/2019 9:07 AM
Inserted Cells [49]
James Menking 2/18/2019 9:07 AM
Inserted Cells [50]
James Menking 2/18/2019 9:07 AM
Inserted Cells [51]
James Menking 2/18/2019 9:07 AM
James Menking 2/18/2019 9:07 AM
James Menking 2/18/2019 9:07 AM
James Menking 2/18/2019 9:07 AM
Deleted Cells [52]
James Menking 2/18/2019 9:07 AM
James Menking 2/18/2019 9:07 AM
Inserted Cells [54]
James Menking 2/18/2019 9:07 AM
Inserted Cells [55]
James Menking 2/18/2019 9:07 AM
James Menking 2/18/2019 9:07 AM
Formatted Table [56]
James Menking 2/18/2019 9:07 AM
James Menking 2/18/2019 9:07 AM
James Menking 2/18/2019 9:07 AM
James Menking 2/18/2019 9:07 AM
James Menking 2/18/2019 9:07 AM
James Menking 2/18/2019 9:07 AM
James Menking 2/18/2019 9:07 AM
James Menking 2/18/2019 9:07 AM
James Menking 2/18/2019 9:07 AM
James Menking 2/18/2019 9:07 AM

[revised manuscript text omitted]